# HSSBench: Benchmarking Humanities and Social Sciences Ability for Multimodal Large Language Models

**Zhaolu Kang**[1,2,*], **Junhao Gong**[1,*], **Jiaxu Yan**[2,4,*], **Wanke Xia**[3], **Yian Wang**[4], **Ziwen Wang**[2],
**Huaxuan Ding**[2], **Zhuo Cheng**[5], **Wenhao Cao**[6], **Zhiyuan Feng**[3], **Siqi He**[2], **Shannan Yan**[3],
**Junzhe Chen**[3], **Xiaomin He**[1], **Chaoya Jiang**[1], **Wei Ye**[1,†], **Kaidong Yu**[2,†], **Xuelong Li**[2,†]
[1]National Engineering Research Center for Software Engineering, Peking University
[2]Institute of Artificial Intelligence, China Telecom (TeleAI)
[3]Tsinghua University  [4]Chinese Academy of Sciences
[5]University of British Columbia  [6]Renmin University of China
zlkang25@stu.pku.edu.cn

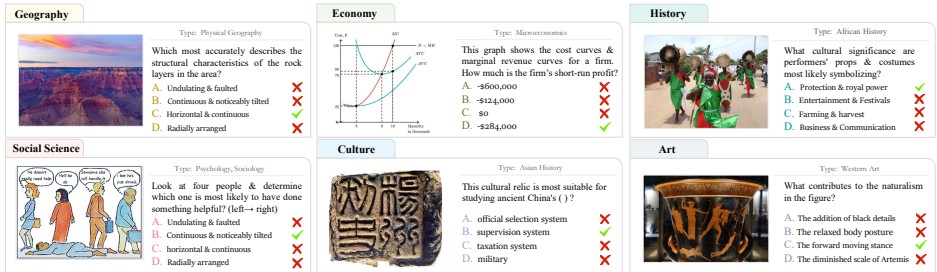

Figure 1: We propose HSSBench, a large-scale benchmark spanning six diverse categories and 45 types, comprising 13,152 samples collected in the six official languages of the United Nations.

## Abstract

Multimodal Large Language Models (MLLMs) have demonstrated significant potential to advance a broad range of domains. However, current benchmarks for evaluating MLLMs primarily emphasize general knowledge and vertical step-by-step reasoning typical of STEM disciplines, while overlooking the distinct needs and potential of the Humanities and Social Sciences (HSS). Tasks in the HSS domain require more horizontal, interdisciplinary thinking and a deep integration of knowledge across related fields, which presents unique challenges for MLLMs, particularly in linking abstract concepts with corresponding visual representations. Addressing this gap, we present HSSBench, a dedicated benchmark designed to assess the capabilities of MLLMs on HSS tasks in multiple languages, including the six official languages of the United Nations. We also introduce a novel data generation pipeline tailored for HSS scenarios, in which multiple domain experts and automated agents collaborate to generate and iteratively refine each sample. HSSBench contains over 13,000 meticulously designed samples, covering six key categories. We benchmark more than 20 mainstream MLLMs on HSSBench and demonstrate that it poses significant challenges even for state-of-the-art models. We hope that this benchmark will inspire further research into enhancing the cross-disciplinary reasoning abilities of MLLMs, especially their capacity to internalize and connect knowledge across fields.

## 1 Introduction

Multimodal large language models (MLLMs) Achiam et al. (2023); Team et al. (2024) have demonstrated remarkable performance across a wide range of tasks, recently achieving or even exceeding

---

[*]Equal Contribution. [†]Corresponding Authors.
[1]HSSBench is publicly available at: https://github.com/Zhaolu-K/HSSBench.

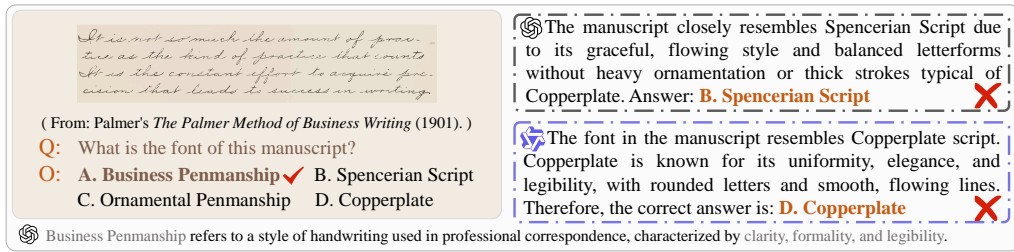

Figure 2: An example of cross-modal knowledge transfer issues in MLLMs within the HSS domain. They struggle to associate "Business Penmanship" font knowledge with relevant images or recognize fonts in image text.

human-level capabilities in many of them. As the performance of MLLMs continues to improve, conducting a comprehensive evaluation of their capabilities has become increasingly essential. In recent times, numerous benchmarks for evaluating MLLMs have emerged Hendrycks et al. (2020); Yue et al. (2024); Liu et al. (2024b); Saikh et al. (2022); Zhang et al. (2024). These tasks are designed to assess the models' ability to jointly understand and reason across multiple modalities, such as images and text, from various perspectives.

Specifically, most multimodal benchmarks are designed either from a general perspective Liu et al. (2024b; 2023); Meng et al. (2024); Han et al. (2024); Qian et al. (2024) or with a focus on scientific disciplines such as mathematics Wang et al. (2024a); Lu et al. (2024) science Li & Tajbakhsh (2023); Liang et al. (2024), and programming Song et al. (2025). Unlike STEM fields that employ "vertical reasoning": a focused sequential process using logical deduction and experimental analysis to arrive at singular correct answers, the **H**umanities and **S**ocial **S**ciences (HSS) emphasize "horizontal reasoning," requiring connections across different contexts and generating multiple valid interpretations rather than single solutions. This fundamental difference stems from the inherent attributes of these disciplines: While STEM fields utilize relatively fixed symbolic systems with standardized reading sequences, HSS disciplines feature symbol systems deeply rooted in regional cultures with meanings that require historical-cultural context interpretation. Furthermore, STEM knowledge can be iteratively developed through logical deduction and experimental analysis, whereas HSS knowledge verification relies on more complex pathways involving cross-referencing literature and expert consensus, with strong dependencies on real-world information.

Although some efforts have been made to explore aspects of the HSS, these attempts lack depth and do not provide a comprehensive and thoughtful examination of MLLMs within the context of these fields. In the study of HSS-related problems, achieving cross-modal knowledge transfer is crucial. Take the scenario illustrated in Figure 2 as an example: humans with basic knowledge can accurately infer "Business Penmanship" from the image content. When directly asked about knowledge points related to "Business Penmanship", the MLLM provides correct answers. However, when queried indirectly through an image, the model fails to recognize the font features in the text, preventing it from associating the relevant knowledge points with the image. This reveals a problem: most models struggle to establish meaningful mapping relationships between HSS-related images and the abstract concepts they represent. Although these models may recognize abstract concepts in isolation, they fail to effectively connect these concepts with the corresponding visual content. From a long-term perspective, a model skilled at solving mathematical problems but unable to interpret historical contexts or understand ethical principles offers an incomplete and potentially harmful form of intelligence.

To address this challenge, we introduce HSSBench, an innovative and comprehensive multilingual benchmark specifically crafted to thoroughly assess the performance of MLLMs in the HSS domain. Comprising around 13k carefully curated test items, HSSBench is structured into 45 types and covers 6 key categories within the field. HSSBench utilizes the visual question answering (VQA) format for this purpose, as shown in Figure 1. Given the involvement of numerous key domains, our work requires interdisciplinary collaboration. To this end, we have engaged experts from various fields to design the data framework and ensure quality control, thereby maximizing the representativeness and credibility of HSS-related issues. In addition to domain experts contributing data, we leveraged the

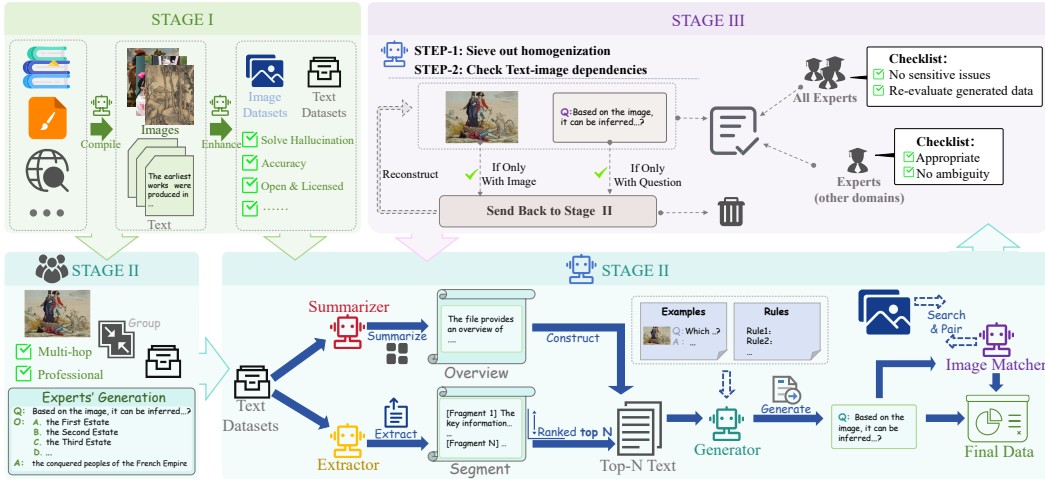

Figure 3: Our pipeline to build HSSBench.

expertise of both specialists and MLLMs to develop a VQA generation pipeline, which produced a portion of the high-quality data. Finally, through evaluations supporting English, Chinese, French, Russian, Spanish, and Arabic, HSSBench enables the assessment of the capabilities of MLLMs in addressing HSS challenges across a wide range of linguistic contexts.

In this study, we evaluate the performance of HSSBench across a range of MLLMs and find that it presents a significant challenge for these models, as their accuracy often falls below 60%. We conducted several comparative experiments to analyze the performance of the models. Our contributions can be summarized as follows:

- First, we introduced HSSBench, a novel dataset specifically designed for the HSS domain, which encompasses 6 distinct categories and 45 major types of HSS tasks.
- Second, we offered a practical data construction method. It utilizes a multi-agent framework tailored for the HSS domain, allowing batch generation of high-quality, novel datasets.
- Finally, we conducted detailed evaluations of over 20 MLLMs on HSSBench across six languages, verifying that HSS tasks still pose significant challenges for MLLMs. This work establishes a foundation for future MLLM research focusing on HSS and serves as a benchmark for future studies in this field.

## 2 HSSBench

The worldview within the HSS is expansive and lacks unified definitions. Our dataset focuses on six high-interest categories within the HSS domain: geography, art, culture, social sciences, history, and economics. Textual data remain the primary medium for disseminating knowledge in the HSS domain, while pictorial data, though less abundant and more challenging to collect, provide valuable complementary information. This presents significant challenges for experts in the construction of datasets. For this purpose, we designed and developed a VQA Generation Pipeline (VGP) to generate the dataset. After engaging experts for annotation, we designed a data construction agent based on their annotation logic, enabling us to generate a sufficient volume of data.

In this section, we will outline the details of the VGP. We have divided this process into three main stages: Dataset Preparation, Dataset Construction, and Validation. The three stages are summarized in Figure 3. 1) During the Dataset Preparation stage, both experts and a networked agent participate in obtaining the raw materials necessary for data construction. Either source is sufficient to provide the foundational content for the subsequent stage. 2) In the Dataset Construction stage, data can be constructed using two methods: expert-constructed and agents-constructed approach. 3) Within the Validation stage, both experts and agent filtration work jointly to identify high-quality data. The screened data passes through the second stage again until it meets the required standards. Data that do not conform to the construction logic are removed.

## 2.1 DATASET PREPARATION, STAGE I

At this stage, we focus on collecting data on target disciplines, covering texts and illustrations. Through systematic organization and strict screening, we lay a solid data foundation for the subsequent development of high-quality questions and answers.

**Data Selection and Collection with Experts.**   During the data preparation for HSSBench, ensuring data diversity is crucial. For image and text selection, we adopt a combined approach involving experts and a multi-agent framework.

For images, to avoid data leakage, we initially encourage experts from various fields to use their private images. When acquisition is difficult, the dataset also incorporates images drawn by experts and licensed ones from open-source communities obtained by both experts and the multi-agent framework. Experts and agents collaborate to screen image content, choosing those related to subject areas and targeted knowledge points for constructing high-quality questions.

Regarding text, as experts may face challenges like unfamiliarity with domain-specific knowledge, we recommend they acquire high-quality textual materials from multiple sources, such as academic resource repositories of universities and open-source communities. This enriches information sources and mitigates bias. In the early stages, domain experts obtain credible resources like textbooks, past papers, and digital course materials from various disciplines. Since these resources are proofread during compilation, they possess high reliability and dense knowledge content, making them trustworthy sources for constructing questions. Each expert reviews materials in their field, extracting relevant text passages and images, eliminating redundant information and standardizing the data format.

**Networked Information-Aggregation Agent.**   Inspired by the expert-led data collection process, we designed the Networked Information-Aggregation Agent to mimic their workflow. This agent broadens data sources by filtering Internet data while maintaining quality. For each discipline, it first compiles relevant keywords as knowledge point indices. Then, it retrieves online data, classifies it into text and images, and matches data against the keywords to assess relevance. For text, it evaluates aspects like professionalism, uniqueness, logical structure, and the need for cross-validation with images. If the text meets criteria, related images are extracted. Through this process, we obtain high-quality multimodal data at a controlled cost. Finally, domain experts review the curated data to ensure professionalism and reliability.

## 2.2 DATASET CONSTRUCTION, STAGE II

At this stage, we developed a multidisciplinary visual-question generation pipeline comprising two key phases: Experts Construction and Multi-Agent Construction. These phases are designed to ensure that the questions produced offer a comprehensive assessment of the model's effectiveness.

**Dataset Construction with experts.**   At this stage, experts have two main responsibilities: revising existing questions and creating entirely new ones. 1) When revising existing questions, experts optimize multiple-choice items, encompassing both the original ones provided in the materials and those requiring revision after failing validation tests. Experts carefully review the relevant textual and visual content, adjusting the question stems and answer choices by integrating real-world knowledge. The goal is to strengthen the connection between the questions and the accompanying images while improving the plausibility of the incorrect options. 2) For creating new questions, experts generate items based on the given high-quality texts and images. These new questions are expected to demand identification of knowledge points from the images and complex reasoning based on real-world understanding to arrive at the correct answers. Following these principles, experts produce a set of high-quality questions that fully utilize the provided multimodal materials to meet the demands of model performance evaluation.

**Dataset Construction with Multi-Agent.**   Inspired by the expert annotation process and the high-quality questions it produces, we designed a multi-agent automated construction framework aimed at improving data generation efficiency and reducing human labor. This agent architecture comprises several roles, including summarizer, extractor, question generator, and image matcher.

Initially, the summarizer and extractor independently analyze the document text to produce a comprehensive summary and a set of high-quality text segments, respectively. The summary provides an overview of the key knowledge points within the document, while the extracted segments offer detailed information suitable for direct question formulation. LLM then scores these text segments based on information density, uniqueness, and logical coherence, selecting the top N segments according to their scores.

The question generator then formulates N questions by leveraging both the full summary and the selected text segments. It operates under detailed guidelines and is supported by multiple examples of human-authored questions, including stems, options, answers, and explanations, to ensure adherence to question design standards. Finally, the image matcher pairs images with questions by leveraging either direct image-question matching or matching based on image descriptions, depending on the available data. This Multi-Agent approach enables large-scale generation of high-quality question datasets efficiently and with minimal human intervention.

## 2.3 VALIDATION, STAGE III

This subsection outlines the stringent validation procedures implemented to ensure the quality and relevance of the data used in model evaluation. Two critical types of validation are performed: Agent Validation and Expert Validation.

**Agent Validation.** This stage aims to eliminate duplicate questions and verify the strength of the correlation between the visual content and the corresponding text. First, the Validation Agent calculates the textual similarity between questions and filters out highly redundant ones based on predefined criteria to ensure the diversity of the dataset. Next, for all constructed data, it is necessary to ensure that 1) without providing an image, the question cannot be correctly answered based solely on the text, and 2) without providing a question, the image alone cannot lead to the correct answer. This requirement stems from our aim to scrutinize the model's capabilities from a multimodal perspective. If a single modality, text or images, suffices to answer a question, it compromises the quality of the dataset and its ability to comprehensively evaluate the performance of MLLMs. To meet these two requirements, the Validation Agent assesses the extent to which the question depends on the image. If the image is deemed unnecessary, the question is sent back to the Stage II for revision; if it still fails to meet the requirements after multiple iterations, the question is discarded.

**Expert Validation.** This stage focuses on expert evaluation of the data. For data annotated by experts, each entry must be 1) validated by other domain experts to confirm its appropriateness and lack of ambiguity, and 2) confirmed by all experts to ensure that it is free from sensitive issues. In the case of data generated by models, a rigorous evaluation by experts specializing in data generation is required to verify its accuracy and absence of ambiguity.

Furthermore, it requires the collective consent of all experts to confirm that the data are free of sensitive issues.

Figure 4: Overview of HSS-Bench.

## 2.4 IMPLEMENTATION DETAILS

In our VGP, we leverage GPT-4o and GPT-4.1 due to their outstanding capabilities. Throughout all stages of LLM, we employed the Chain-of-Thought(COT) prompting strategy. For more detailed information on VGP, please refer to Appendix A.

## 2.5 DATA STATISTICS

HSSBench consists of 13,152 multiple-choice questions distributed across six major categories—Economy, Art, Culture, Social Sciences, History, and Geography—and further divided into 45 specialized subtypes.

We require each question to be presented in a multiple-choice format with one correct answer and several distractors that are plausible but ultimately incorrect. Although some original questions in the HSS domain naturally have multiple correct answers, these multi-answer questions have been reformulated into single-answer questions for consistency. This was achieved by combining the

multiple correct options into a single choice. This approach preserves the original content and complexity of the questions while conforming to a uniform single-answer multiple-choice format.

It supports evaluation in six different languages. Our initial data came from multiple countries and languages, with questions originally created by domain experts in their source languages (examples in Appendix D). We then used LLM-based translation models to translate the questions into five other languages. All translations were carefully reviewed and validated by bilingual experts to ensure linguistic accuracy and cultural appropriateness, maintaining semantic consistency while respecting cultural nuances.

The statistical data of HSSBench is illustrated in Figure 4, with a detailed description of the dataset provided in Appendix B. Appendix B.1 reports human expert accuracy across all categories. Appendix B.2 provides an in-depth taxonomy of categories and subtypes. Appendix B.3 also includes an analysis of the most frequent content words per category. Appendix B.4 details the composition of the dataset in terms of contributions of human experts versus automated agents, together with quality validation through model accuracy comparisons.

## 3 EXPERIMENT

### 3.1 EXPERIMENTAL SETUP

**Models.** For open-source models, we selected Qwen2.5-VL-3B/7B/32B/72B Bai et al. (2025), Qwen2-VL-72B Wang et al. (2024c), QVQ-72B-Preview Wang et al. (2024b), Deepseek-VL2-Tiny Wu et al. (2024c), MiniCPM-o-2.6 Yao et al. (2024), mPLUG-Owl3-2B/7B Ye et al. (2024), Llava-onevision-7B Li et al. (2024b), Llama3-llava-next-8b Li et al. (2024a), InternVL3-8B Zhu et al. (2025), InternVL2.5-8B-MPO Chen et al. (2024b), Phi-3.5-Vision-Instruct Abdin et al. (2024), Janus-Pro Chen et al. (2025), Llava1.5 Liu et al. (2024a). For closed-source commercial models, we utilized GPT-4o/4.1/4.1-mini/4.1-nano OpenAI et al. (2024).

**Evaluation Settings.** 1) We conducted evaluations using two types of prompts to examine the MLLMs performance under different prompting strategies. One approach involved prompting the model to directly output the answer without any intermediate reasoning, while the other required the model to generate a COT before providing the final answer. 2) Additionally, we tested each question in six different language versions to investigate how the MLLMs' performance varies across languages for the same question. 3) Furthermore, we optimized the question formats by designing two types of prompts for each item: multiple-choice and open-ended questions. This allowed us to assess whether the model can correctly answer questions without any explicit hints. Detailed prompt settings are provided in Appendix C.8. 4) For the HSSBench evaluation, we employed two assessment methods and selected GPT-4o as a representative closed-source model and Qwen2.5-7B as a representative open-source model. A detailed comparative analysis of both models is provided in Appendix C.9.

### 3.2 RESULTS

**Overall Results.** Table 1 presents the overall results of various open-source and closed-source MLLMs evaluated on HSSBench in the context of English language tests. Ct. and Dr. represent two types of prompts: COT prompts and direct response prompts. C. and O. denote question types: multiple-choice and open-ended questions. "Human" refers to the average final scores achieved by experts in various fields. Detailed experimental results for Dr. and are presented in Appendix C.1.

All of our initial generated data consisted of multiple-choice questions. Since some questions cannot be reasonably converted to open-ended formats, changing all of them into open-ended questions would lead to poor-quality data. This approach would be unfair for evaluating the model and would fail to accurately reflect differences in model performance. Therefore, we chose to modify only those questions that can still be answered meaningfully as open-ended questions to serve as the evaluation data for open-ended question performance.

The experimental results highlight the performance differences between the various models. Of the open-source models, Qwen2.5-VL-72B-Instruct delivers the highest performance, although it still falls short of surpassing closed-source models in open-ended question tests. The GPT-4.1 series

| Model | Geography | | Economy | | Culture | | Social Sciences | | History | | Art | | All | | | |
|---|---|---|---|---|---|---|---|---|---|---|---|---|---|---|---|---|
| | Ct.C. | Ct.O. | Ct.C. | Ct.O. | Ct.C. | Ct.O. | Ct.C. | Ct.O. | Ct.C. | Ct.O. | Ct.C. | Ct.O. | Dr.C. | Dr.O. | Ct.C. | Ct.O. |
| Random | 24.93 | 0.00 | 21.92 | 0.00 | 25.00 | 0.00 | 24.90 | 0.00 | 24.91 | 0.00 | 25.00 | 0.00 | 24.62 | 0.00 | 24.62 | 0.00 |
| Human | 94.14 | - | 93.06 | - | 92.99 | - | 94.44 | - | 93.84 | - | 95.53 | - | 93.83 | - | 93.83 | - |
| *Open-source LLM (Scale < 10B)* | | | | | | | | | | | | | | | | |
| Qwen2.5-VL-3B-Instruct | 35.25 | 11.74 | 28.56 | 16.36 | 34.61 | 1.85 | 39.46 | 11.50 | 36.23 | 11.07 | 34.55 | 3.62 | 29.01 | 9.94 | 34.99 | 9.33 |
| Qwen2.5-VL-7B-Instruct | 40.69 | 21.60 | 41.19 | 31.31 | 30.19 | 4.63 | 42.86 | 19.00 | 40.62 | 19.67 | 33.42 | 11.31 | 37.88 | 15.21 | 38.19 | 17.89 |
| Llava-onevision-7b | 32.05 | 7.04 | 32.23 | 11.68 | 31.02 | 1.85 | 36.63 | 3.00 | 27.07 | 4.51 | 32.92 | 3.17 | 36.20 | 5.73 | 31.56 | 5.20 |
| Llama3-llava-next-8b | 27.59 | 4.23 | 19.82 | 6.54 | 30.53 | 3.24 | 32.89 | 6.50 | 26.87 | 8.61 | 29.26 | 5.43 | 31.20 | 6.50 | 27.93 | 5.81 |
| InternVL3-8B | 42.12 | 10.80 | 33.70 | 16.36 | 38.69 | 7.41 | 48.30 | 13.00 | 45.09 | 12.70 | 38.61 | 13.57 | 42.14 | 12.27 | 41.42 | 12.31 |
| InternVL2.5-8B-MPO | 37.24 | 17.37 | 34.07 | 20.09 | 35.00 | 8.33 | 43.35 | 15.50 | 40.54 | 16.39 | 35.99 | 13.57 | 39.30 | 11.77 | 37.68 | 15.21 |
| Phi-3.5-Vision-Instruct | 25.55 | 9.39 | 26.80 | 13.55 | 29.01 | 3.70 | 28.78 | 3.00 | 20.31 | 7.79 | 28.81 | 4.07 | 35.89 | 10.32 | 26.04 | 6.96 |
| Janus-Pro | 29.11 | 8.45 | 22.54 | 6.07 | 41.00 | 6.02 | 34.65 | 12.00 | 29.75 | 10.25 | 33.22 | 7.69 | 30.03 | 8.49 | 31.66 | 8.41 |
| Deepseek-VL2-Tiny | 5.78 | 3.29 | 3.77 | 4.95 | 14.12 | 0.00 | 6.35 | 3.00 | 6.37 | 2.46 | 13.03 | 16.74 | 29.86 | 3.42 | 8.23 | 5.09 |
| mPLUG-Owl3-2B | 25.83 | 4.69 | 24.63 | 1.35 | 31.83 | 3.69 | 29.99 | 3.00 | 25.96 | 4.51 | 29.43 | 4.07 | 28.73 | 4.02 | 27.71 | 3.57 |
| mPLUG-Owl3-7B | 30.60 | 7.04 | 33.15 | 9.91 | 13.73 | 2.30 | 36.40 | 7.00 | 27.64 | 4.10 | 25.69 | 2.30 | 33.01 | 6.68 | 27.52 | 6.23 |
| MiniCPM-o-2.6 | 26.02 | 5.98 | 19.08 | 7.22 | 22.22 | 3.57 | 26.83 | 8.59 | 25.96 | 4.03 | 22.25 | 5.38 | 3.70 | 5.71 | 24.11 | 5.71 |
| Llava1.5 | 12.75 | 4.03 | 7.38 | 2.68 | 10.74 | 3.36 | 7.38 | 5.37 | 10.74 | 4.03 | 10.74 | 6.04 | 8.06 | 4.03 | 9.96 | 4.25 |
| *Open-source LLM (Scale > 10B)* | | | | | | | | | | | | | | | | |
| Qwen2.5-VL-32B-Instruct | **52.48** | 21.33 | 52.79 | 6.67 | 38.94 | 8.00 | **53.87** | 24.00 | **57.03** | 26.67 | 39.20 | 3.33 | 48.38 | 15.89 | **50.75** | 15.00 |
| Qwen2-VL-72B-Instruct | 50.74 | 17.86 | 52.55 | 32.65 | 45.91 | 8.62 | 50.19 | 16.07 | 55.53 | 16.36 | 41.34 | 16.36 | 54.22 | 20.43 | 49.39 | 17.21 |
| Qwen2.5-VL-72B-Instruct | 55.59 | 13.33 | **53.83** | 37.33 | 41.49 | 7.33 | **57.77** | 17.57 | **60.30** | 28.19 | 40.84 | 14.67 | 54.17 | 18.17 | **51.87** | 19.73 |
| QVQ-72B-Preview | 19.93 | 3.33 | 21.87 | 17.33 | 29.67 | 3.33 | 26.54 | 7.43 | 28.86 | 18.79 | 23.80 | 9.33 | 25.60 | 10.37 | 24.69 | 9.92 |
| *Closed-source LLM* | | | | | | | | | | | | | | | | |
| GPT-4o | 46.88 | 22.07 | 52.97 | 35.14 | 45.61 | 14.29 | 45.26 | 16.00 | 48.36 | 15.98 | 43.42 | 12.67 | 46.09 | 20.05 | 46.88 | 19.36 |
| GPT-4.1 | 39.81 | **40.38** | 48.08 | **52.70** | **48.95** | **24.88** | 35.36 | **49.25** | 41.91 | **48.77** | **43.51** | **23.53** | 45.02 | **25.38** | 42.66 | **39.97** |
| GPT-4.1-mini | 47.67 | **34.27** | **58.27** | 49.10 | 48.26 | 20.74 | 45.05 | **36.68** | 47.84 | **36.89** | **43.88** | **23.53** | 45.75 | **24.32** | 48.03 | **33.59** |
| GPT-4.1-nano | 33.21 | 30.74 | 39.71 | 41.44 | 38.70 | 7.83 | 37.52 | 28.14 | 34.01 | 30.74 | 35.83 | 20.36 | 36.33 | 21.12 | 35.83 | 26.22 |

Table 1: Scores (%) of MLLMs on HSSBench (EN-I). The highest and second highest scores are marked in blue and green, respectively.

achieves state-of-the-art performance in most tasks, with a particularly impressive accuracy of 39.97% in open questions, almost double that of other closed-source models. In contrast, some open-source models show considerably lower performance. We also observe that model performance varies by language. Multilingual evaluation results are provided in the Appendix C.2.

**Model performance on different categories.** Among the six categories evaluated on HSSBench (EN-I), the economic-related tasks consistently emerge as the most challenging for the models. The average score of all models in this category is the lowest, indicating that addressing economic problems requires a deep understanding of various economic theories and the ability to apply them in complex reasoning. However, current open-source MLLMs still exhibit significant deficiencies in these aspects. In contrast, closed-source models perform exceptionally well in economically related tasks, and this advantage may stem from their exposure to large amounts of high-quality training data in the economic domain.

On the other hand, the Geography category appears to be the easiest task for the models, with the highest average scores observed across the board. This trend implies that geographic knowledge, which is often more factual and less abstract compared to other humanities and social sciences domains, is better captured by the training data and reasoning capabilities of the models.

Interestingly, in certain categories of multiple-choice tasks, such as Culture and Social Sciences, some open-source models outperform their closed-source counterparts. For example, larger open-source models like Qwen2.5-VL-32B and Qwen2.5-VL-72B demonstrate competitive or even superior results compared to closed-source models like GPT-4o in these domains. This phenomenon may be partially attributed to the fact that most of the data experts we invited are Chinese. Although the evaluation was conducted in English, it is possible that Qwen benefits from a training data advantage related to Chinese content, which could have influenced its superior performance in these tasks. It also indicates that the gap between open-source and closed-source models is narrowing in specific HSS tasks.

### 3.3 QUALITATIVE ANALYSIS

**Comparison between Direct Answer and COT Prompting.** Table 1 shows subtle differences in MLLMs' performance on HSSBench when using direct answer versus COT prompting. Notably, COT does not always help; some models perform better with direct answers, indicating that longer reasoning can mislead them.

Specifically, COT prompts exacerbate hallucination issues in certain models, where reasoning flaws in textual analysis and misinterpretation of visual inputs lead to the generation of incorrect background knowledge during step-by-step analysis. This causes reasoning to deviate from the correct answer. For example, when tackling geographic questions, models often struggle to accurately interpret spatial elements such as location markers and contour lines—visual features primarily composed of points and lines—resulting in analytical outcomes that contradict the actual image content.

Even when notable errors do not occur in intermediate reasoning steps, the final summarization phase can suffer due to excessive information generation, which exceeds the model's ability to effectively weigh the importance of answer options, leading to prediction mistakes. This highlights the importance of integrating reasoning steps cohesively. Further detailed analysis in Appendix D reveals that many models fail to internalize visual knowledge during the divergent thinking process of HSS tasks.

**Comparison between Multiple-choices and Open-ended.**  We reformulated some questions as open-ended to test models without answer options. Results reveal that HSS tasks remain very challenging: only a few models exceed 15% accuracy. This matches expectations, as even experts find these questions difficult without options or background cues.

Answer choices provide prior knowledge that narrows the answer space. Without them, models' reasoning becomes highly divergent, often drifting far from correct answers. This suggests models mainly rely on shallow visual features (e.g., size, type, motion) but miss deeper symbolic information like cultural context or spatiotemporal cues.

These findings highlight two key limitations: insufficient real-world knowledge and weak integration of visual and textual modalities. The performance drop with COT further reflects models' struggles to internalize and retrieve complex knowledge. Improving model capabilities on HSS tasks remains a critical challenge.

**The role of accurate answers.**  To test if models truly understand questions rather than exploiting options, we added confusing choices like "None of the above" while keeping the correct answer unchanged, sampling 150 questions per category (results in Table 11, Appendix C.3).

Lower-performing models' accuracy dropped noticeably. Analysis revealed two patterns: some models were misled to select the confusing option; others, though not choosing it, showed incoherent reasoning, failing to filter out wrong choices and sometimes excluding the correct answer altogether. This indicates models struggle to assess the relative credibility of options and make reliable judgments, exposing their vulnerability to distractors and raising concerns about robustness.

**Visual Information Extraction.**  To investigate the impact of visual information loss on the performance of MLLMs, we designed two complementary experiments. In the first experiment, we used GPT-4.1 to generate detailed textual descriptions of the images. In the second experiment, we invited domain experts to produce comprehensive and precise annotations for each image (results in Table 12, Appendix C.4).

Most models' accuracy dropped when images were replaced by GPT-generated texts, showing that direct visual input contains critical details lost in conversion. However, with expert annotations, accuracy improved noticeably despite no image access. Some models even surpassed previous performance ceilings in Culture and Social Sciences, as expert texts helped focus on crucial visual cues and better link to domain knowledge.

These results confirm earlier observations that current MLLMs have inherent limitations in retrieving and understanding visual information fully. They underscore the importance of improving models' abilities to extract and integrate key visual features for HSS tasks.

**Comparative Analysis of HSSBench with Related Benchmarks.**  We compared HSSBench with other benchmarks that include some HSS data, such as CMMMU, MME, and MMMU, as well as with STEM benchmarks (details in Appendix C.5 and C.6).

Across HSS benchmarks, relative model performance trends are consistent, but HSSBench yields lower accuracy, indicating it is more challenging and better captures HSS complexity. Compared

to STEM benchmarks, models perform much better on STEM tasks, highlighting the unique difficulties of HSS domains that require nuanced cultural and social understanding. These comparisons demonstrate HSSBench's value in driving progress on underexplored HSS challenges.

**Evaluation of Retrieval-Augmented Generation on HSSBench.** We tested Retrieval-Augmented Generation (RAG) by integrating external knowledge from Wikipedia and HSS documents with several smaller MLLMs under direct and COT prompting (results in Tables 15 and 16, Appendix C.7).

Contrary to expectations, RAG did not consistently improve accuracy. Some modest gains appeared in specific domains or models, but overall performance was often similar or worse than direct prompting alone. This suggests that general retrieval corpora and simple integration methods are insufficient for the nuanced knowledge HSS tasks demand.

The limited effectiveness of RAG in supporting MLLMs for complex multi-hop HSS tasks stems from misalignment between general-purpose corpora and HSS knowledge, as well as deficiencies in current retrieval and integration methods. Moreover, MLLMs struggle to internalize and transfer newly retrieved domain knowledge. These challenges highlight the need for specialized HSS retrieval resources, RAG techniques tailored to HSS, and enhanced mechanisms for MLLMs to acquire, transfer, and apply HSS-specific knowledge.

## 4 RELATED WORK

In recent years, significant progress has been made in the development of multimodal benchmarks and methodologies. Numerous datasets have emerged that assess models from various perspectives, which can be broadly categorized into Generation Benchmarks Liu et al. (2024b; 2023); Meng et al. (2024); Han et al. (2024); Qian et al. (2024); Xu et al. (2023); Yin et al. (2023); Zeng et al. (2023); Wu et al. (2024a); Luo et al. (2024); Feng et al. (2025); Wang et al. (2025e), Reasoning Benchmarks Wang et al. (2024a); Lu et al. (2024); Li & Tajbakhsh (2023); Liang et al. (2024); Song et al. (2025), and Application Benchmarks Fan et al. (2022); Rawles et al. (2023); Chen et al. (2023); Sermanet et al. (2023); Chen et al. (2024a); Koh et al. (2024); You et al. (2024); Gao et al. (2025); Kang et al. (2026); Zheng et al. (2026); He et al. (2026). Generation benchmarks cover a wide range of tasks, often including not only reasoning challenges but also content related to the humanities and social sciences. Their main objective is to provide a comprehensive assessment of the performance of MLLMs in various dimensions. In contrast, reasoning benchmarks are designed primarily around mathematical and scientific problems, seeking to rigorously assess the capabilities of MLLM in logical and analytical reasoning. Among the various evaluation formats, multiple-choice datasets are the most prevalent, owing to their simplicity in evaluation and ease of comparison Zeng et al. (2023).

The proliferation of these data sets has significantly accelerated research progress, serving both as training resources and as tools for assessing the multifaceted competencies of MLLMs Wang et al. (2025f); Liu et al. (2025b); Luo et al. (2026); Guan et al. (2026); Fu et al. (2026); Wu et al. (2025; 2024b); Feng et al. (2026); Li et al. (2026; 2025d;e); Hu et al.; Zhang et al. (2025c;b;a); Ding et al. (2025); Li et al. (2025c); Wang et al. (2026); Ma et al. (2026); Liu et al. (2025a); Bi et al. (2026); Wang et al. (2025d;c;b;a); Liang et al. (2025); Ma et al. (2025a;b); Li et al. (2025a); Yu et al. (2025); Feng et al. (2024; 2023); Li et al. (2021; 2024c; 2025b). Advances in data set construction are particularly exciting, ranging from human-annotated high-quality datasets Romero et al. (2024) to those generated through pipelines built in LLM Chandrasegaran et al. (2024). Despite these achievements, several critical limitations remain. 1) Most multimodal datasets cover a wide range of categories, but suffer from issues such as limited data sources, relatively simple questions, and insufficient image information. 2) Many large-scale datasets are collected through web crawling without thorough manual annotation and verification, which can introduce biases in evaluation results. 3) Current reasoning datasets predominantly focus on STEM tasks, relying heavily on scientific and mathematical data to assess reasoning capabilities. In contrast, the benchmark we propose is created through interactions among expert agents, targeting the HSS domain. In contrast, the benchmark we propose focuses on the HSS domain, emphasizing tasks that analyze and understand the abstract concepts embedded in images. This approach fills a notable gap in the field and lays the foundation for advancing model performance in this underexplored area.

## 5 CONCLUSION

We present HSSBench, a novel benchmark dataset constructed through a multi-agent pipeline involving experts from diverse fields. HSSBench is designed to rigorously evaluate the true mastery of tasks by models within the HSS domain. The dataset comprises six categories, each derived from raw data collected from repositories in the six official languages of the United Nations and subsequently processed to generate task-specific data.

We then carried out comprehensive benchmarking of various MLLMs using HSSBench. Our results reveal that HSSBench poses significant challenges to all tested models, which exhibit poor performance on reasoning tasks in the HSS domain. In particular, the accuracy of the model decreases substantially when answer choices are not provided as prompts. We hope that releasing HSSBench will encourage the AI community to place greater emphasis on reasoning over non-STEM data, thereby advancing research on MLLMs from this important perspective.

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

## A MORE VGP DETAILS

### A.1 ANNOTATION PLATFORM

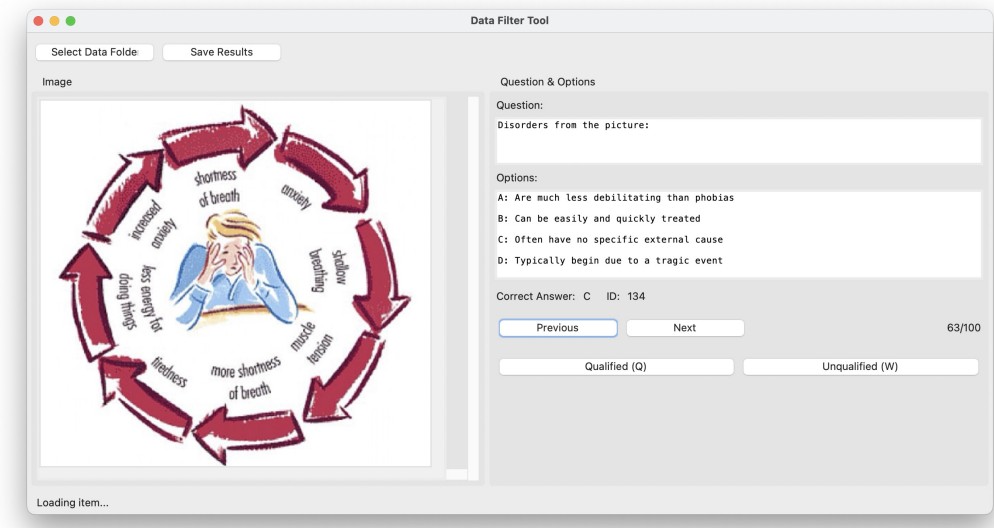

Figure 5: The interface for data construction and validation, allowing experts to use a visual interface to assist their work.

We use a simple visual interface as our annotation platform. For question input, experts can upload or write questions in the form. The interface is shown in Figure 5. During the validation process, experts can see all the data submitted by other experts. They can select entries to view detailed previews of the entries.

### A.2 INFORMATION ABOUT EXPERTS

| Name | Languages | Professional Background |
|------|-----------|-------------------------|
| Expert 1 | Chinese, English | Computer Science, Art, Economy |
| Expert 2 | Chinese, Japanese, English | Computer Science, Culture |
| Expert 3 | English, Chinese | Computer Science, Geography |
| Expert 4 | Chinese, English | Economy, History |
| Expert 5 | Chinese, English, Japanese | Language and Linguistics, History, Culture |
| Expert 6 | Chinese, English, Russian | Language and Linguistics, Social Science |
| Expert 7 | Chinese, English, Arabic | Language and Linguistics, Social Science |
| Expert 8 | Chinese, English, Arabic | Economy, Social Science |
| Expert 9 | Chinese, English | Geography |
| Expert 10 | Chinese, English, French | Economy, Art |
| Expert 11 | English, Chinese, Japanese | Art, Culture |
| Expert 12 | Chinese, English | Computer Science, Geography |
| Expert 13 | Chinese, English | Geography, History |
| Expert 14 | Chinese, English, Spanish | Computer Science, Geography |

Table 2: Information about the experts involved in dataset construction.

The team of experts who contributed to the creation of HSSBench comes from a variety of cultural and linguistic backgrounds, as well as interdisciplinary academic fields spanning the humanities and social sciences. This diversity is reflected in their linguistic fluency, professional expertise, and

international academic experiences. Table 2 summarizes the profiles of the experts involved in the dataset construction.

Many of the experts have international academic experiences and interdisciplinary training across various humanities and social science domains. While individual cultural perspectives may naturally influence the emphasis or framing of certain questions, we consider this diversity a strength rather than a limitation. It enriches the dataset by incorporating multiple viewpoints and insights, which is essential for a benchmark designed to be multilingual and cross-cultural.

To mitigate potential cultural bias, we carefully curated the source materials provided to the experts during the initial stage of dataset construction. These reference materials originate from multiple countries and languages, ensuring that the constructed questions and answers reflect widely accepted, global knowledge rather than culturally subjective viewpoints. Furthermore, during the data validation and filtering stages, we rigorously excluded content that could be culturally biased or inconsistent with universal values.

In summary, our approach balances the preservation of valuable cultural diversity with the need to maintain fairness and universality in the dataset. This careful design ensures that HSSBench serves as a robust and inclusive benchmark for evaluating multilingual large language models (MLLMs) across different languages and cultural contexts.

## B    DATASET DETAILS

This section provides detailed information about our benchmark designed to evaluate MLLMs' visual comprehension abilities through multiple-choice questions. The benchmark spans six major categories: Economy, Art, Culture, Social sciences, History, and Geography, each containing various specialized subtypes.

### B.1    HUMAN PERFORMANCE

To establish a performance baseline, we asked three relevant experts in each domain to spend a significant amount of time answering the entire dataset. Table 3 summarizes their performance across the six categories.

| Category | Expert 1 | Expert 2 | Expert 3 | Overall |
|----------|----------|----------|----------|---------|
| Economy | 96.02 | 91.70 | 94.70 | 94.14 |
| Art | 93.12 | 92.61 | 93.44 | 93.06 |
| Culture | 92.50 | 90.83 | 95.64 | 92.99 |
| Social sciences | 94.49 | 91.77 | 97.07 | 94.44 |
| History | 95.91 | 91.86 | 93.75 | 93.84 |
| Geography | 94.99 | 92.78 | 95.81 | 94.53 |
| Average | 94.84 | 91.76 | 95.40 | 93.83 |

Table 3: Scores (%) of Experts.

Human experts demonstrated high proficiency across all categories, with overall accuracy ranging from 92.99% to 94.53%. The highest individual performance was observed in Social sciences by Expert 3 (97.07%), while the lowest was in Culture by Expert 2 (90.83%). This high level of human performance establishes a challenging benchmark for evaluating large language models.

### B.2    THE DETAILS OF TYPES

Our dataset is organized into six major categories with various subtypes in each category. Table 4 presents the detailed breakdown of the dataset structure and the count of questions in each type. We allow each data entry to have multiple types because the intersection of knowledge across disciplines is essential.

The dataset exhibits varying distributions across categories, with Geography containing the largest number of questions and Economy the smallest. Within categories, there are also significant variations in subtype representation, reflecting the natural distribution of content within these domains.

### B.3 MOST-FREQUENT WORDS IN THE QUESTIONS

We analyzed the most frequent content words in the questions in all categories to understand the linguistic characteristics and tasksur dataset. Figure 6 shows the word clouds for the most frequent words in HSSBench per category, excluding common stop words. The word frequency analysis reveals distinct patterns across categories:

1) Visual observation terms dominate in Art and Culture categories ("observe", "picture", "shown", "scene"), indicating a focus on visual analysis tasks.

2) Economy questions frequently use technical terms ("price", "firm", "cost", "demand", "market", "marginal"), reflecting domain-specific concepts.

3) Geography questions heavily employ spatial and diverse visual comprehension capabilities ("diagram", "map", "area", "distribution"), emphasizing spatial reasoning.

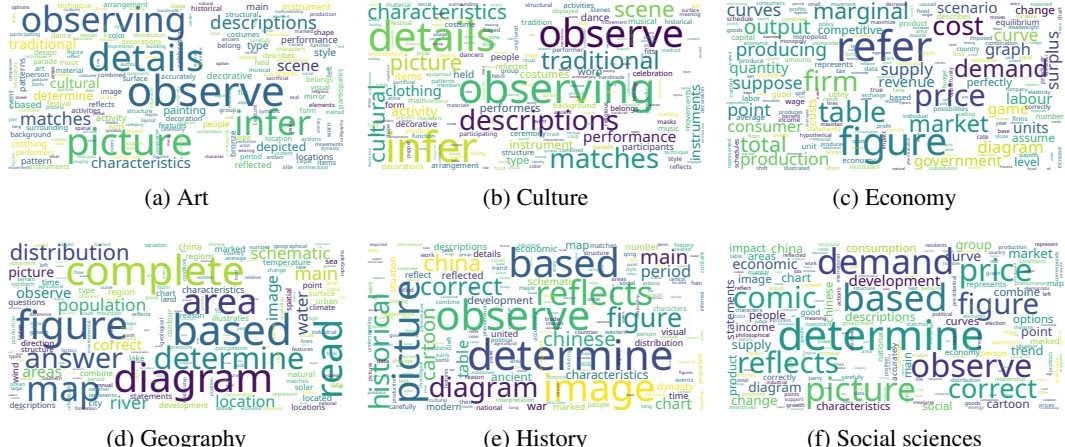

| (a) Art | (b) Culture | (c) Economy |
| (d) Geography | (e) History | (f) Social sciences |

Figure 6: Word Cloud in HSSBench per category.

### B.4 COMPOSITION AND QUALITY COMPARISON BETWEEN HUMAN AND AUTOMATED AGENTS

We provide a detailed breakdown of the data sources across the six domains in HSSBench, as shown in Table 5.

To further validate the quality of data generated by both human experts and automated agents, we evaluated the performance of GPT4.1-mini on the respective subsets. The accuracy results are summarized in Table 6.

The comparable accuracy results indicate that data generated by automated agents, after undergoing multi-round expert verification, maintain a quality level consistent with that of human expert contributions. These clarifications and statistics have been incorporated into the revised manuscript to enhance transparency regarding dataset construction and quality assurance.

## C EXPERIMENTAL DETAILS

### C.1 DETAILS OF RESULTS

Table 7 provides additional details on the experimental results presented in Table 1. The table below reports the overall accuracy of the models under the DIRECT prompt and presents a detailed breakdown of the performance for each category under the same prompt.

| Category | Type | Count |
|---|---|---|
| Economy | Microeconomics | 1,193 |
| | Macroeconomics | 163 |
| | Labor Economics | 105 |
| | Environmental Economics | 63 |
| | International Trade | 63 |
| | Resource Economics | 44 |
| | International Finance | 32 |
| Art | Folk Art | 1,051 |
| | Eastern Art | 1,041 |
| | Western Art | 672 |
| | Spiritual Art | 630 |
| | Applied Art | 452 |
| | Entertainment Art | 298 |
| | Elite Art | 303 |
| Culture | Regional Culture | 1,790 |
| | Material Culture | 1,382 |
| | Intangible Culture | 1,218 |
| | Indigenous Culture | 733 |
| | Institutional Culture | 266 |
| | World Culture | 228 |
| Social sciences | Sociology | 950 |
| | Political Science | 550 |
| | Psychology | 339 |
| | Anthropology | 190 |
| | Economics | 147 |
| | Education | 135 |
| | Philosophy | 127 |
| | Law | 99 |
| | Ethics | 61 |
| | Journalism | 24 |
| | History | 18 |
| History | Asian History | 1,586 |
| | World Modern History | 885 |
| | World Ancient History | 754 |
| | European History | 602 |
| | World Contemporary History | 346 |
| | American History | 295 |
| | African History | 147 |
| | Oceanian History | 82 |
| Geography | Physical Geography | 2,361 |
| | Regional Geography | 1,694 |
| | Descriptive Geography | 1,682 |
| | Explanatory Geography | 1,348 |
| | Human Geography | 1,319 |
| | Predictive Geography | 75 |

Table 4: Dataset Taxonomy and Question Distribution.

| Category | Human Expert | Automated Agent | Total Samples |
|---|---|---|---|
| Art | 984 | 1,204 | 2,188 |
| Geography | 2,671 | 891 | 3,562 |
| History | 1,946 | 550 | 2,496 |
| Economy | 974 | 459 | 1,433 |
| Social Science | 487 | 947 | 1,434 |
| Culture | 652 | 1,387 | 2,039 |

Table 5: Distribution of samples constructed by human experts and automated agents across different domains in HSSBench.

| Category | Human Expert Acc. | Automated Agent Acc. |
|---|---|---|
| Art | 42.32 | 45.43 |
| Geography | 48.82 | 46.52 |
| History | 45.59 | 49.12 |
| Economy | 58.87 | 57.67 |
| Social Science | 43.15 | 46.95 |
| Culture | 49.11 | 47.41 |

Table 6: Model accuracy (%) of GPT4.1-mini on human-expert-generated and automated-agent-generated subsets across different domains.

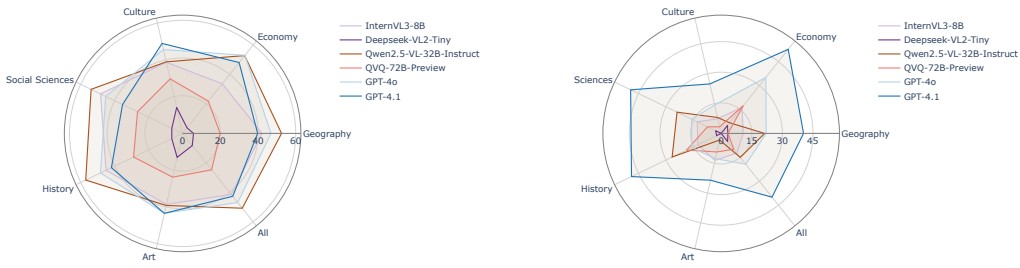

(a) Effect under multiple-choice question setting      (b) Effect under open-ended question setting

Figure 7: Partial MLLMs' results under CoT prompt settings.

| Model | Geography | | Economics | | Culture | | Social Sciences | | History | | Art | | All | | | |
|---|---|---|---|---|---|---|---|---|---|---|---|---|---|---|---|---|
| | Dr.C. | Dr.O. | Dr.C. | Dr.O. | Dr.C. | Dr.O. | Dr.C. | Dr.O. | Dr.C. | Dr.O. | Dr.C. | Dr.O. | Dr.C. | Dr.O. | Ct.C. | Ct.O. |
| Random | 24.93 | 0.00 | 21.92 | 0.00 | 25.00 | 0.00 | 24.90 | 0.00 | 24.91 | 0.00 | 25.00 | 0.00 | 24.62 | 0.00 | 24.62 | 0.00 |
| Human | 94.14 | - | 93.06 | - | 92.99 | - | 94.44 | - | 93.84 | - | 95.53 | - | 93.83 | - | 93.83 | - |
| *Open-source LLM (Scale < 10B)* | | | | | | | | | | | | | | | | |
| Qwen2.5-VL-3B | 29.84 | 11.27 | 29.44 | 17.76 | 27.83 | 5.09 | 35.22 | 10.00 | 26.91 | 8.61 | 26.78 | 7.24 | 29.01 | 9.94 | 34.99 | 9.33 |
| Qwen2.5-VL-7B | 40.07 | 19.25 | 38.18 | 30.37 | 30.29 | 7.87 | 44.91 | 11.00 | 41.20 | 14.75 | 32.33 | 8.14 | 37.88 | 15.21 | 38.19 | 17.89 |
| Llava-onevision-7b | 37.46 | 6.10 | 31.20 | 10.75 | 38.64 | 1.39 | 40.95 | 7.00 | 31.65 | 4.51 | 37.33 | 4.98 | 36.20 | 5.73 | 31.56 | 5.20 |
| Llama3-llava-next-8b | 29.98 | 6.10 | 21.81 | 5.61 | 35.69 | 6.48 | 35.36 | 6.00 | 29.48 | 9.02 | 34.46 | 5.43 | 31.20 | 6.50 | 27.93 | 5.81 |
| InternVL3-8B | 42.88 | 13.62 | 37.08 | 17.97 | 37.71 | 6.91 | 48.94 | 12.00 | 47.22 | 14.34 | 37.48 | 8.60 | 42.14 | 12.27 | 41.42 | 12.31 |
| InternVL2.5-8B-MPO | 38.08 | 14.08 | 34.73 | 16.36 | 39.13 | 6.94 | 45.69 | 10.50 | 40.82 | 11.07 | 38.27 | 11.76 | 39.30 | 11.77 | 37.68 | 15.21 |
| Phi-3.5-Vision-Instruct | 29.84 | 10.33 | 29.44 | 11.21 | 27.83 | 10.65 | 35.22 | 7.00 | 26.91 | 11.48 | 26.78 | 10.86 | 29.01 | 10.32 | 26.04 | 6.96 |
| Janus-Pro | 27.57 | 8.92 | 16.01 | 7.01 | 42.38 | 6.02 | 31.26 | 9.00 | 28.62 | 12.30 | 32.33 | 7.24 | 30.03 | 8.49 | 31.66 | 8.41 |
| Deepseek-VL2-tiny | 25.44 | 5.16 | 17.38 | 3.15 | 40.17 | 0.46 | 30.33 | 2.50 | 29.97 | 4.51 | 35.19 | 4.52 | 29.86 | 3.42 | 8.23 | 5.09 |
| mPLUG-Owl3-2B | 25.27 | 4.23 | 30.48 | 2.70 | 32.56 | 2.30 | 34.17 | 4.50 | 26.88 | 6.15 | 30.48 | 4.07 | 28.73 | 4.02 | 27.71 | 3.57 |
| mPLUG-Owl3-7B | 32.71 | 7.04 | 33.08 | 6.31 | 33.45 | 4.61 | 37.31 | 5.50 | 29.97 | 8.20 | 33.73 | 8.14 | 33.01 | 6.68 | 27.52 | 6.23 |
| MiniCPM-o-2.6 | 3.48 | 1.67 | 3.90 | 14.16 | 2.06 | 3.70 | 5.75 | 3.03 | 4.72 | 2.27 | 3.02 | 8.70 | 3.70 | 5.71 | 24.11 | 5.71 |
| Llava1.5 | 3.36 | 2.68 | 10.07 | 5.37 | 3.36 | 2.01 | 8.05 | 4.70 | 8.05 | 1.34 | 15.44 | 8.05 | 8.06 | 4.03 | 9.96 | 4.25 |
| *Open-source LLM (Scale > 10B)* | | | | | | | | | | | | | | | | |
| Qwen2.5-VL-32B | 52.54 | 16.00 | 39.38 | 8.00 | 39.87 | 10.00 | 53.09 | 26.67 | 57.52 | 24.67 | 44.92 | 10.00 | 48.38 | 15.89 | 50.75 | 15.00 |
| Qwen2.5-VL-72B | 56.71 | 16.00 | 51.67 | 32.00 | 57.77 | 5.33 | 47.43 | 16.89 | 59.26 | 24.16 | 63.12 | 14.67 | 54.17 | 18.17 | 51.87 | 19.73 |
| QVQ-72B-Preview | 21.53 | 3.33 | 18.38 | 22.00 | 29.57 | 2.67 | 26.26 | 10.14 | 29.58 | 12.75 | 23.85 | 11.33 | 25.60 | 10.37 | 24.69 | 9.92 |
| Qwen2-VL-72B | 54.22 | 19.30 | 41.06 | 24.07 | 55.33 | 5.66 | 58.85 | 12.96 | 62.17 | 32.08 | 50.00 | 18.87 | 54.22 | 17.79 | 49.39 | 18.82 |
| *Closed-source LLM* | | | | | | | | | | | | | | | | |
| GPT-4o | 45.68 | 23.47 | 48.92 | 35.14 | 48.16 | 15.67 | 46.72 | 18.00 | 47.24 | 14.75 | 46.66 | 13.57 | 46.99 | 20.05 | 46.88 | 19.36 |
| GPT-4.1 | 43.80 | 26.29 | 52.34 | 42.34 | 52.77 | 18.89 | 40.86 | 26.13 | 38.86 | 22.13 | 44.74 | 16.74 | 45.02 | 25.38 | 42.66 | 39.97 |
| GPT-4.1-mini | 45.14 | 26.29 | 54.92 | 45.95 | 52.08 | 15.21 | 41.84 | 22.11 | 39.34 | 18.85 | 44.70 | 17.65 | 45.75 | 24.32 | 48.03 | 33.59 |
| GPT-4.1-nano | 33.18 | 20.66 | 34.96 | 36.04 | 45.61 | 14.75 | 38.42 | 15.58 | 32.69 | 20.49 | 36.47 | 18.55 | 36.33 | 21.12 | 35.83 | 26.22 |

Table 7: Scores (%) of MLLMs on HSSBench (EN-II).

## C.2 RESULTS IN DIFFERENT LANGUAGES

Table 1 and Table 7 present the test results on English-language data. To analyze the impact of different languages on the model, we also conducted large-scale experiments using Chinese-language data. The final experimental results are shown in Table 8 and Table 9 below.

| Model | Geography | | Economics | | Culture | | Social Sciences | | History | | Art | | All | | | |
|---|---|---|---|---|---|---|---|---|---|---|---|---|---|---|---|---|
| | Ct.C. | Ct.O. | Ct.C. | Ct.O. | Ct.C. | Ct.O. | Ct.C. | Ct.O. | Ct.C. | Ct.O. | Ct.C. | Ct.O. | Dr.C. | Dr.O. | Ct.C. | Ct.O. |
| Random | 24.93 | 0.00 | 21.92 | 0.00 | 25.00 | 0.00 | 24.90 | 0.00 | 24.91 | 0.00 | 25.00 | 0.00 | 24.62 | 0.00 | 24.62 | 0.00 |
| Human | 94.14 | - | 93.06 | - | 92.99 | - | 94.44 | - | 93.84 | - | 95.53 | - | 93.83 | - | 93.83 | - |
| *Open-source LLM (Scale < 10B)* | | | | | | | | | | | | | | | | |
| Qwen2.5-VL-3B | 40.49 | 10.80 | 28.89 | 19.81 | 36.30 | 0.93 | 42.43 | 9.50 | 43.91 | 11.89 | 33.85 | 4.98 | 35.45 | 11.79 | 35.45 | 9.65 |
| Qwen2.5-VL-7B | 48.43 | 15.96 | 35.00 | 25.00 | 27.60 | 2.78 | 49.36 | 19.50 | 50.54 | 15.98 | 32.90 | 6.79 | 43.12 | 13.17 | 41.86 | 14.24 |
| Llava-onevision-7b | 39.68 | 4.69 | 30.80 | 10.38 | 32.96 | 1.85 | 39.46 | 7.00 | 32.97 | 4.51 | 35.03 | 2.71 | 37.89 | 4.98 | 35.62 | 5.13 |
| Llama3-llava-next-8b | 27.29 | 3.29 | 19.82 | 3.77 | 33.55 | 0.93 | 32.89 | 2.50 | 26.84 | 3.28 | 29.48 | 2.26 | 31.18 | 5.21 | 28.35 | 2.68 |
| InternVL3-8B | 47.45 | 11.74 | 35.37 | 17.92 | 38.21 | 9.26 | 53.47 | 14.00 | 50.97 | 18.44 | 39.94 | 10.41 | 44.81 | 14.70 | 44.92 | 13.71 |
| InternVL2.5-8B-MPO | 41.14 | 11.74 | 36.11 | 23.58 | 36.20 | 6.94 | 48.09 | 16.50 | 47.63 | 6.94 | 38.35 | 10.86 | 42.72 | 14.01 | 41.45 | 14.55 |
| Phi-3.5-vision-instruct | 22.29 | 6.10 | 25.64 | 4.25 | 30.16 | 2.31 | 23.55 | 1.00 | 23.35 | 4.92 | 28.30 | 1.81 | 34.08 | 6.74 | 25.35 | 3.45 |
| Janus-Pro | 28.10 | 11.27 | 21.67 | 8.49 | 44.16 | 6.48 | 34.79 | 8.50 | 29.13 | 14.34 | 40.49 | 14.48 | 33.50 | 11.94 | 32.81 | 10.72 |
| mPLUG-Owl3-2B | 27.20 | 1.88 | 29.43 | 0.90 | 33.74 | 2.30 | 29.78 | 3.00 | 26.76 | 2.87 | 29.43 | 4.52 | 32.25 | 1.90 | 28.25 | 2.58 |
| mPLUG-Owl3-7B | 35.01 | 5.63 | 34.23 | 3.15 | 26.39 | 1.84 | 39.26 | 2.50 | 26.08 | 2.46 | 34.23 | 5.43 | 34.22 | 4.71 | 31.77 | 3.49 |
| MiniCPM-o-2.6 | 26.97 | 7.69 | 19.50 | 4.88 | 35.21 | 6.25 | 31.23 | 3.90 | 30.27 | 6.76 | 30.20 | 5.98 | 1.29 | 6.83 | 29.04 | 5.94 |
| *Open-source LLM (Scale > 10B)* | | | | | | | | | | | | | | | | |
| Qwen2.5-VL-32B | 58.45 | 19.33 | 49.86 | 7.33 | 40.90 | 5.33 | 56.49 | 22.67 | 64.33 | 24.67 | 41.62 | 14.67 | 51.60 | 14.22 | 52.86 | 15.67 |
| Qwen2.5-VL-72B | 61.65 | 12.67 | 52.02 | 33.33 | 43.60 | 10.67 | 61.04 | 14.19 | 68.08 | 18.12 | 43.76 | 16.00 | 60.04 | 21.96 | 55.94 | 17.51 |
| QVQ-72B-Preview | 19.82 | 8.00 | 21.10 | 18.00 | 30.80 | 3.33 | 27.79 | 6.76 | 27.85 | 10.74 | 25.90 | 6.67 | 22.95 | 9.14 | 24.82 | 8.92 |
| Qwen2-VL-72B | 21.28 | 19.30 | 38.25 | 24.07 | 21.25 | 5.66 | 27.79 | 12.96 | 25.69 | 32.08 | 27.65 | 18.87 | 59.11 | 17.79 | 25.59 | 18.82 |
| *Closed-source LLM* | | | | | | | | | | | | | | | | |
| GPT-4.1mini | 47.53 | 35.21 | 55.55 | 55.41 | 44.43 | 18.43 | 44.07 | 34.17 | 47.92 | 43.03 | 42.50 | 30.32 | 44.19 | 24.54 | 46.78 | 36.32 |

Table 8: Scores (%) of MLLMs on HSSBench (ZH-I).

Figure 8 illustrates the performance of various models under four different prompt configurations.

Table 10 and Figure 9 shows the performance of the model on datasets in different languages. Our dataset includes data organized in six languages. The table below presents the experimental results on a stratified sample of 900 instances.

| Model | Geography | | Economics | | Culture | | Social Sciences | | History | | Art | | All | | | |
|---|---|---|---|---|---|---|---|---|---|---|---|---|---|---|---|---|
| | Dr.C.g | Dr.O.g | Dr.C.g | Dr.O.g | Dr.C.g | Dr.O. | Dr.C.g | Dr.O.g | Dr.C.g | Dr.O.g | Dr.C.g | Dr.O.g | Dr.C.g | Dr.O.g | Ct.C.g | Ct.O.g |
| Random | 24.93 | 0.00 | 21.92 | 0.00 | 25.00 | 0.00 | 24.90 | 0.00 | 24.91 | 0.00 | 25.00 | 0.00 | 24.62 | 0.00 | 24.62 | 0.00 |
| Human | 94.14 | - | 93.06 | - | 92.99 | - | 94.44 | - | 93.84 | - | 95.53 | - | 93.83 | - | 93.83 | - |
| *Open-source LLM (Scale < 10B)* | | | | | | | | | | | | | | | | |
| Qwen2.5-VL-3B | 39.51 | 12.21 | 29.26 | 13.21 | 30.94 | 7.87 | 40.03 | 14.50 | 38.36 | 11.48 | 30.03 | 11.76 | 35.45 | 11.79 | 35.45 | 9.65 |
| Qwen2.5-VL-7B | 52.52 | 17.84 | 33.97 | 20.75 | 27.60 | 4.63 | 51.13 | 15.00 | 52.13 | 11.48 | 31.17 | 9.95 | 43.12 | 13.17 | 41.86 | 14.24 |
| Llava-onevision-7b | 42.20 | 6.57 | 30.21 | 7.55 | 36.59 | 1.39 | 42.93 | 5.00 | 35.61 | 4.10 | 36.12 | 5.43 | 37.89 | 4.98 | 35.62 | 5.13 |
| Llama3-llava-next-8b | 29.61 | 4.69 | 19.97 | 6.60 | 38.80 | 5.09 | 34.37 | 5.00 | 28.86 | 3.28 | 34.54 | 6.33 | 31.18 | 5.21 | 28.35 | 2.68 |
| InternVL3-8B | 46.44 | 18.78 | 34.27 | 20.75 | 40.08 | 6.94 | 52.90 | 18.00 | 51.05 | 14.34 | 40.19 | 9.95 | 44.81 | 14.70 | 44.92 | 13.71 |
| InternVL2.5-8B-MPO | 42.26 | 18.78 | 31.32 | 14.15 | 40.72 | 7.87 | 48.87 | 17.50 | 49.11 | 12.30 | 40.78 | 14.03 | 42.72 | 14.01 | 41.45 | 14.55 |
| Phi-3.5-vision-instruct | 29.92 | 6.57 | 27.49 | 7.08 | 45.58 | 7.87 | 32.89 | 4.50 | 29.71 | 6.56 | 40.68 | 7.69 | 34.08 | 6.74 | 25.35 | 3.45 |
| Janus-Pro | 27.29 | 9.86 | 18.50 | 12.74 | 50.83 | 14.35 | 32.18 | 8.50 | 29.44 | 10.25 | 43.21 | 15.84 | 33.50 | 11.94 | 32.81 | 10.72 |
| mPLUG-Owl3-2B | 31.25 | 0.94 | 23.73 | 2.71 | 35.95 | 0.46 | 37.17 | 3.00 | 31.89 | 2.46 | 33.18 | 2.71 | 32.25 | 1.90 | 28.25 | 2.58 |
| mPLUG-Owl3-7B | 35.85 | 7.51 | 35.65 | 6.79 | 35.70 | 1.38 | 40.59 | 3.00 | 28.17 | 4.10 | 35.65 | 6.79 | 34.22 | 4.71 | 31.77 | 3.49 |
| MiniCPM-o-2.6 | 0.93 | 7.55 | 0.77 | 8.06 | 1.42 | 1.61 | 1.99 | 5.41 | 0.60 | 20.00 | 2.42 | 2.27 | 1.29 | 6.83 | 29.04 | 5.94 |
| *Open-source LLM (Scale > 10B)* | | | | | | | | | | | | | | | | |
| Qwen2.5-VL-32B | 56.29 | 20.67 | 42.83 | 6.00 | 41.83 | 4.00 | 57.42 | 18.00 | 63.00 | 25.33 | 42.21 | 11.33 | 51.60 | 14.22 | 52.86 | 15.67 |
| Qwen2.5-VL-72B | 68.15 | 22.67 | 48.40 | 30.00 | 50.07 | 12.00 | 63.73 | 20.27 | 71.02 | 31.54 | 49.02 | 15.33 | 60.04 | 21.96 | 55.94 | 17.51 |
| QVQ-72B-Preview | 19.40 | 6.67 | 17.90 | 20.00 | 29.72 | 4.00 | 24.34 | 11.49 | 28.13 | 8.05 | 23.44 | 4.67 | 22.95 | 9.14 | 24.82 | 8.92 |
| Qwen2-VL-72B | 64.42 | 28.07 | 41.83 | 16.67 | 55.98 | 7.55 | 62.00 | 14.81 | 68.77 | 22.64 | 51.49 | 16.98 | 59.11 | 17.79 | 25.59 | 18.82 |
| *Closed-source LLM* | | | | | | | | | | | | | | | | |
| GPT-4.1mini | 45.93 | 26.76 | 53.31 | 43.69 | 46.25 | 14.29 | 39.61 | 20.10 | 39.26 | 24.18 | 42.09 | 17.65 | 44.19 | 24.54 | 46.78 | 36.32 |

Table 9: Scores (%) of MLLMs on HSSBench (ZH-II).

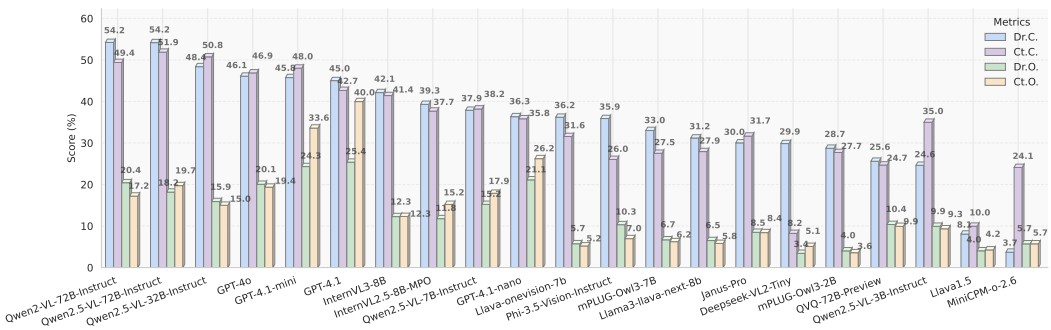

Figure 8: Comparison of Model Performances Across Four Prompt Settings.

| Model | Arabic | Chinese | English | French | Russian | Spanish |
|---|---|---|---|---|---|---|
| InternVL3-8B | 35.20 | 49.37 | 42.38 | 39.37 | 38.70 | 39.21 |
| Qwen2.5-VL-7B-Instruct | 33.74 | 40.21 | 38.37 | 35.55 | 34.83 | 34.83 |
| Qwen2.5-VL-32B-Instruct | 46.94 | 54.79 | 50.24 | 47.57 | 48.51 | 50.24 |
| Qwen2.5-VL-72B-Instruct | 49.72 | 55.41 | 51.28 | 51.28 | 48.27 | 48.83 |
| QVQ-72B-Preview | 33.98 | 36.89 | 39.80 | 37.86 | 40.77 | 38.83 |
| GPT-4.1-mini | 41.33 | 44.89 | 41.67 | 41.78 | 41.44 | 46.89 |
| Average | 40.82 | 46.93 | 43.96 | 41.99 | 42.87 | 43.47 |

Table 10: Contrast Scores (%) of MLLMs on HSSBench for six UN languages and six models.

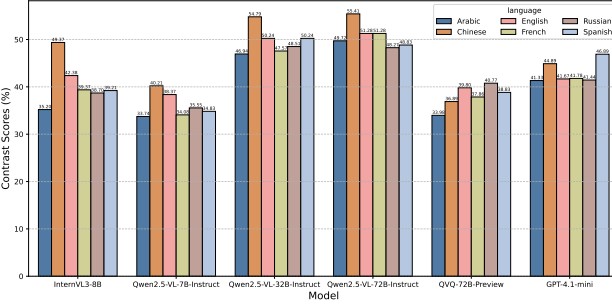

Figure 9: Contrast Scores (%) of MLLMs on HSSBench for six UN languages and six models.

| Model | Geography | | Art | | Culture | | Social Sciences | | History | | Economy | | Total | |
|---|---|---|---|---|---|---|---|---|---|---|---|---|---|---|
| | Ct.C. | Ct.C.Conf. | Ct.C. | Ct.C.Conf. | Ct.C. | Ct.C.Conf. | Ct.C. | Ct.C.Conf. | Ct.C. | Ct.C.Conf. | Ct.C. | Ct.C.Conf. | Ct.C. | Ct.C.Conf. |
| Qwen2.5-VL-7B | 48.00 | 44.67 | 32.12 | 31.69 | 29.33 | 26.67 | 39.86 | 34.00 | 55.33 | 49.33 | 35.51 | 36.05 | 40.21 | 37.12 |
| InternVL3-8B | 50.67 | 46.00 | 41.55 | 34.01 | 45.33 | 49.33 | 53.38 | 48.67 | 60.00 | 56.00 | 39.86 | 45.99 | 49.37 | 46.01 |
| MiniCPM-o | 27.51 | 21.47 | 39.33 | 34.67 | 49.33 | 50.67 | 35.81 | 29.73 | 38.26 | 26.85 | 27.33 | 23.33 | 36.34 | 31.22 |
| Qwen2.5-VL-32 | 50.83 | 49.83 | 46.10 | 40.90 | 41.55 | 30.99 | 59.61 | 51.92 | 61.98 | 56.77 | 50.94 | 53.77 | 51.85 | 47.74 |
| QvQ-72B-Preview | 16.67 | 16.67 | 29.33 | 28.00 | 35.78 | 29.67 | 26.35 | 18.91 | 28.18 | 27.51 | 25.33 | 24.66 | 26.98 | 24.41 |
| Qwen2.5-VL-72 | 60.67 | 55.33 | 45.33 | 42.05 | 49.33 | 44.67 | 56.76 | 55.40 | 65.77 | 67.11 | 54.67 | 55.33 | 55.41 | 53.29 |
| GPT-4.1-mini | 46.67 | 47.33 | 56.67 | 54.67 | 52.67 | 47.33 | 40.00 | 42.67 | 54.67 | 44.00 | 45.33 | 42.00 | 49.33 | 46.33 |

Table 11: Contrast Scores (%) with confounding option of MLLMs on HSSBench.

## C.3 MULTIPLE-CHOICE CONFOUNDING OPTION EXPERIMENT DETAILS

Table 11 and Figure 10 provides a detailed presentation of the experimental results after adding a confounding option. We sampled 900 data points. "Conf." indicates that, in addition to the given options, an extra option—"None of the above answers is correct"—was added. The model's output performance was then compared under these conditions.

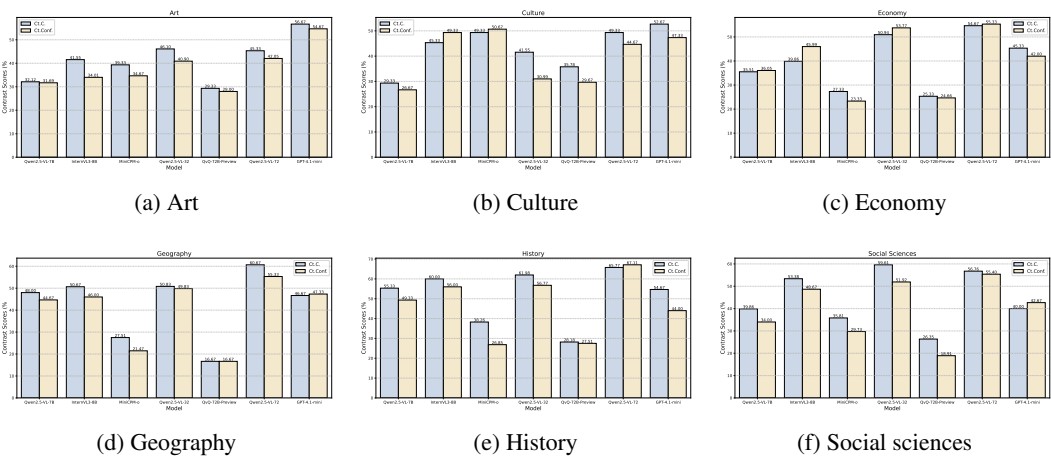

(a) Art     (b) Culture     (c) Economy

(d) Geography     (e) History     (f) Social sciences

Figure 10: Contrast Scores (%) with confounding option of MLLMs on HSSBench.

## C.4 VISUAL INFORMATION EXTRACTION EXPERIMENT DETAILS

Table 12 and Figure 11 presents the extraction of image information into text (where "De." indicates a detailed description of the image generated directly by GPT-4.1, and "De-H." indicates a detailed explanation provided by a domain expert for each image). The table below shows the detailed results of comparative experiments in which only the extracted image information, rather than the images themselves, was provided to the model.

| Model | Geography | | | Art | | | Culture | | | Social Sciences | | | History | | | Economy | | |
|---|---|---|---|---|---|---|---|---|---|---|---|---|---|---|---|---|---|---|
| | Ct.C. | De.C. | De-H.C. | Ct.C. | De.C. | De-H.C. | Ct.C. | De.C. | De-H.C. | Ct.C. | De.C. | De-H.C. | Ct.C. | De.C. | De-H.C. | Ct.C. | De.C. | De-H.C. |
| Qwen2.5-VL-7B | 39.33 | 40.67 | 41.33 | 43.48 | 37.41 | 40.14 | 30.67 | 30.00 | 36.00 | 37.16 | 39.33 | 42.67 | 44.00 | 46.00 | 51.33 | 35.77 | 35.21 | 38.73 |
| InternVL3-8B | 42.00 | 43.33 | 43.33 | 33.33 | 34.69 | 36.05 | 42.00 | 32.00 | 42.00 | 47.30 | 41.33 | 45.33 | 48.00 | 48.00 | 50.00 | 40.88 | 35.92 | 39.44 |
| Qwen2.5-VL-32B-Instruct | 46.00 | 44.67 | 52.00 | 35.33 | 44.00 | 43.33 | 42.67 | 34.00 | 47.33 | 48.67 | 44.67 | 52.67 | 44.00 | 52.67 | 62.67 | 26.67 | 52.00 | 55.33 |
| QvQ-72B-Preview | 31.33 | 36.67 | 42.00 | 26.67 | 30.67 | 38.00 | 28.67 | 27.33 | 36.00 | 34.67 | 34.00 | 36.00 | 29.33 | 40.67 | 38.67 | 19.33 | 37.33 | 33.33 |
| Qwen2.5-VL-72B-Instruct | 54.67 | 54.67 | 58.67 | 42.00 | 48.67 | 41.33 | 34.67 | 32.67 | 42.67 | 47.33 | 50.00 | 47.33 | 56.67 | 64.67 | 66.67 | 35.33 | 56.00 | 57.33 |
| GPT-4.1-mini | 52.00 | 50.00 | 56.00 | 43.33 | 42.67 | 52.67 | 50.00 | 40.67 | 54.00 | 42.67 | 40.00 | 48.67 | 48.67 | 46.67 | 54.67 | 55.33 | 53.33 | 60.00 |

Table 12: Contrast Scores (%) about Visual Information Extraction of MLLMs on HSSBench.

## C.5 COMPARISON WITH HSS-RELATED BENCHMARKS

We acknowledge that some existing benchmarks, such as MME, include HSS-related test data, particularly within the Art domain. To better understand the relationship between these datasets and our proposed HSSBench, we conducted a detailed comparative analysis focusing on the overlapping Art category. The evaluation was performed under identical prompt settings, including both Direct and Chain-of-Thought (CoT) prompting. The combined results are presented in Table 13.

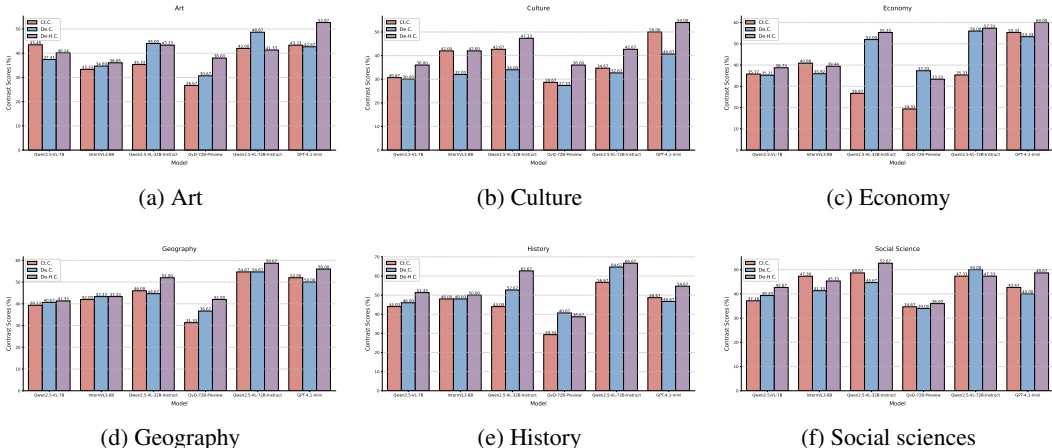

|     |     |     |
| --- | --- | --- |
| (a) Art | (b) Culture | (c) Economy |
| (d) Geography | (e) History | (f) Social sciences |

Figure 11: Contrast Scores (%) about Visual Information Extraction of MLLMs on HSSBench.

| Model | Benchmark (% Accuracy) | | | | Prompting | |
| --- | --- | --- | --- | --- | --- | --- |
| | CMMMU | MME | MMMU | HSSBench | Direct | CoT |
| Qwen2.5-VL-3B-Instruct | 47.73 | 77.32 | 57.76 | 29.01 | 47.73 | 48.86 |
| Qwen2.5-VL-7B-Instruct | 43.18 | 71.21 | 61.21 | 37.88 | 43.18 | 46.59 |
| InternVL3-8B | 65.91 | 85.28 | 68.10 | 42.14 | 65.91 | 55.68 |
| llava-onevision-qwen2-7b | 53.41 | 77.21 | 49.14 | 36.20 | 53.41 | 52.27 |

Table 13: Performance comparison on the Art category across HSS-related benchmarks under Direct and CoT prompting.

As shown, the relative performance trends across these benchmarks are broadly consistent, indicating that the challenges within the Art domain are similar across datasets. Notably, HSSBench presents a more challenging evaluation, reflected by generally lower accuracy scores, which underscores its value in pushing forward research on Humanities and Social Sciences tasks.

## C.6 COMPARISON WITH STEM BENCHMARKS

While our primary focus is on Humanities and Social Sciences (HSS), we also recognize the importance of situating HSSBench within the broader landscape of STEM benchmarks. To this end, we provide a comparative overview of model performance on several representative STEM benchmarks alongside HSSBench under the Chain-of-Thought (CoT) prompting setting. The results are summarized in Table 14.

| Model | MMLU Pro | GPQA Diamond | SWE-bench Verified | MATH-500 | AIME 2024 | LiveCode Bench | OpenCompass-Reasoning | HSSBench |
| --- | --- | --- | --- | --- | --- | --- | --- | --- |
| Qwen2.5-VL-72B | 71.2 | — | — | — | — | — | 50.2 | 51.87 |
| GPT-4.1 mini | — | 65.00 | 23.60 | — | 49.60 | — | 46.0 | 48.03 |
| GPT-4o | 79.80 | 66.90 | — | — | — | 35.80 | 54.8 | 46.88 |
| GPT-4.1 | 81.80 | 66.30 | 54.6 | 92.80 | 48.10 | 40.50 | 54.0 | 42.66 |
| InternVL3-8B | — | — | — | — | — | — | 41.4 | 41.42 |
| GPT-4.1 nano | — | 50.30 | — | — | 29.40 | — | 34.2 | 35.83 |
| Janus-Pro-7B | — | — | — | — | — | — | 19.1 | 31.66 |

Table 14: Model performance comparison on STEM benchmarks and HSSBench (CoT prompting).

**Notes:** All STEM benchmark and OpenCompass results are sourced from official publications or model release notes. A dash (—) indicates that the model did not publicly report results for that

benchmark. Minor discrepancies may exist due to testing variability but do not affect the overall trend.

This comparison reveals that while models generally achieve higher accuracy on STEM benchmarks (e.g., MMLU Pro, MATH-500), their performance on HSSBench is comparatively lower. This gap highlights the unique challenges posed by HSS tasks and the necessity of dedicated benchmarks like HSSBench to drive progress in this domain.

## C.7 RETRIEVAL-AUGMENTED GENERATION EVALUATION

We conducted additional experiments integrating Retrieval-Augmented Generation (RAG) with several smaller MLLMs. The retrieval database was constructed from a general knowledge corpus comprising Wikipedia and publicly available documents related to HSS. Model performance was evaluated under two prompting strategies: direct prompting and CoT prompting.

Table 15: Performance of models under direct prompting.

| Model | Geography | Art | Culture | Social Science | History | Economy | Overall |
|---|---|---|---|---|---|---|---|
| *Without RAG* | | | | | | | |
| Qwen2.5-VL-7B-Instruct | 60.09 | 28.05 | 27.19 | 53.50 | 54.51 | 40.09 | 43.89 |
| Qwen2.5-VL-3B-Instruct | 55.40 | 33.48 | 44.24 | 54.50 | 50.82 | 35.59 | 45.56 |
| InternVL3-8B | 54.93 | 37.56 | 43.78 | 62.00 | 51.64 | 44.14 | 48.82 |
| llava-onevision-qwen2-7b | 46.01 | 30.77 | 34.10 | 46.50 | 38.93 | 29.28 | 37.43 |
| *With RAG* | | | | | | | |
| Qwen2.5-VL-7B-Instruct | 47.42 | 29.86 | 29.95 | 51.00 | 45.08 | 38.74 | 40.24 |
| Qwen2.5-VL-3B-Instruct | 49.77 | 33.03 | 35.48 | 48.50 | 43.44 | 33.78 | 40.55 |
| InternVL3-8B | 53.52 | 35.75 | 35.94 | 59.50 | 49.59 | 37.39 | 45.10 |
| llava-onevision-qwen2-7b | 43.19 | 32.13 | 26.73 | 41.00 | 32.38 | 21.17 | 32.57 |

Table 16: Performance of models under COT prompting.

| Model | Geography | Art | Culture | Social Science | History | Economy | Overall |
|---|---|---|---|---|---|---|---|
| *Without RAG* | | | | | | | |
| Qwen2.5-VL-7B-Instruct | 55.40 | 25.79 | 27.65 | 51.00 | 47.54 | 39.64 | 41.08 |
| Qwen2.5-VL-3B-Instruct | 51.17 | 30.32 | 35.48 | 49.00 | 46.31 | 32.43 | 40.70 |
| InternVL3-8B | 53.52 | 34.39 | 40.55 | 57.00 | 51.64 | 38.74 | 45.86 |
| llava-onevision-qwen2-7b | 43.66 | 31.22 | 28.57 | 43.00 | 33.61 | 29.73 | 34.78 |
| *With RAG* | | | | | | | |
| Qwen2.5-VL-7B-Instruct | 47.89 | 27.15 | 27.19 | 46.50 | 42.62 | 36.49 | 37.89 |
| Qwen2.5-VL-3B-Instruct | 41.78 | 31.22 | 29.95 | 44.00 | 40.57 | 36.04 | 37.21 |
| InternVL3-8B | 57.75 | 30.77 | 34.10 | 55.50 | 50.41 | 39.19 | 44.50 |
| llava-onevision-qwen2-7b | 42.25 | 29.86 | 28.11 | 42.50 | 31.56 | 23.42 | 32.73 |

The results indicate that, although RAG occasionally yields modest improvements in specific domains or models, it does not consistently outperform direct prompting without retrieval augmentation. This suggests that augmenting MLLMs with a general retrieval corpus and straightforward prompting strategies may be insufficient to fully exploit the complex and nuanced knowledge required for HSS tasks.

These findings highlight the challenges inherent in applying RAG to Humanities and Social Sciences benchmarks such as HSSBench. We hypothesize that more specialized, domain-specific retrieval corpora, combined with advanced retrieval and integration techniques, are necessary to unlock the full potential of RAG in this context.

## C.8   PROMPT FOR MODEL INFERENCE

Table 17 details the configurations of the four prompts employed in our experiments, specifying the presence or absence of a CoT prompt and indicating whether the questions were open-ended or multiple-choice.

| | **Prompt for Model Inference** |
|---|---|
| w/ MC w/ CoT | Question: [question] Options: [options] Think step by step to determine the correct answer. End your response with [[X]] where X is your final answer (A, B, C, D or E). |
| w/ MC w/o CoT | Question: [question] Options: [options] Give the correct answer directly. End your response with [[X]] where X is your final answer (A, B, C, D or E). |
| w/o MC w/ CoT | Question: [question] Think step by step to determine the correct answer. End your response with [[X]] where X is your final answer. |
| w/o MC w/o CoT | Question: [question] Give the correct answer directly. End your response with [[X]] where X is your final answer. |

Table 17: Prompt for model inference.

## C.9   EVALUATION FOR MODEL'S OUTPUT

Table 18 presents the evaluation prompts employed to assess the accuracy of the model's responses.

| | **Prompt for Model Evaluation** |
|---|---|
| w/ MC | You are an evaluation assistant. Please determine whether the answers output by the model below are correct. Question: [question] Options: [options] Correct answer: [correct answer] Model output content: [model output] Please extract its final answer from the model output and determine whether it is consistent with the content of the correct answer. If the answer is correct, reply with "1". Otherwise, reply with "0". Only reply with numbers and no superfluous content. |
| w/o MC | You are an evaluation assistant. Please determine whether the answers output by the model below are correct. Question: [question] Correct answer: [correct answer] Model output content: [model output] Please extract its final answer from the model output and determine whether it is consistent with the content of the correct answer. Attention! If the meanings are roughly consistent, it can be regarded as correct. It is not necessary to be strictly and completely consistent. One needs to carefully consider whether the model's response is reasonable and correct. If the answer is correct, reply with "1". Otherwise, reply with "0". Only reply with numbers and no superfluous content. |

Table 18: Prompt for model evaluation.

In our evaluation process, we employed both expert sampling and automated assessment using GPT models. To verify the consistency of the evaluation results produced by the models, we randomly sampled 2,000 data points from the overall experiment, including 1,000 multiple-choice questions and

1,000 open-ended questions. These samples were independently evaluated by three human experts, as well as by Qwen2.5-7B and GPT-4.1.

For both multiple-choice and open-ended questions, the overall agreement among the two models and the three experts, measured by Fleiss' Kappa, was 96.82% and 94.79%, respectively. The agreement between the two models alone, measured by Cohen's Kappa, was 97.32% for multiple-choice questions and 94.79% for open-ended questions.

These results demonstrate that the consistency between the three experts and the two models remains at a reliable level. This indicates that either evaluation method yields results that are largely consistent with those reported in the paper, thereby ensuring the reproducibility of our findings.

## D  MORE CASES

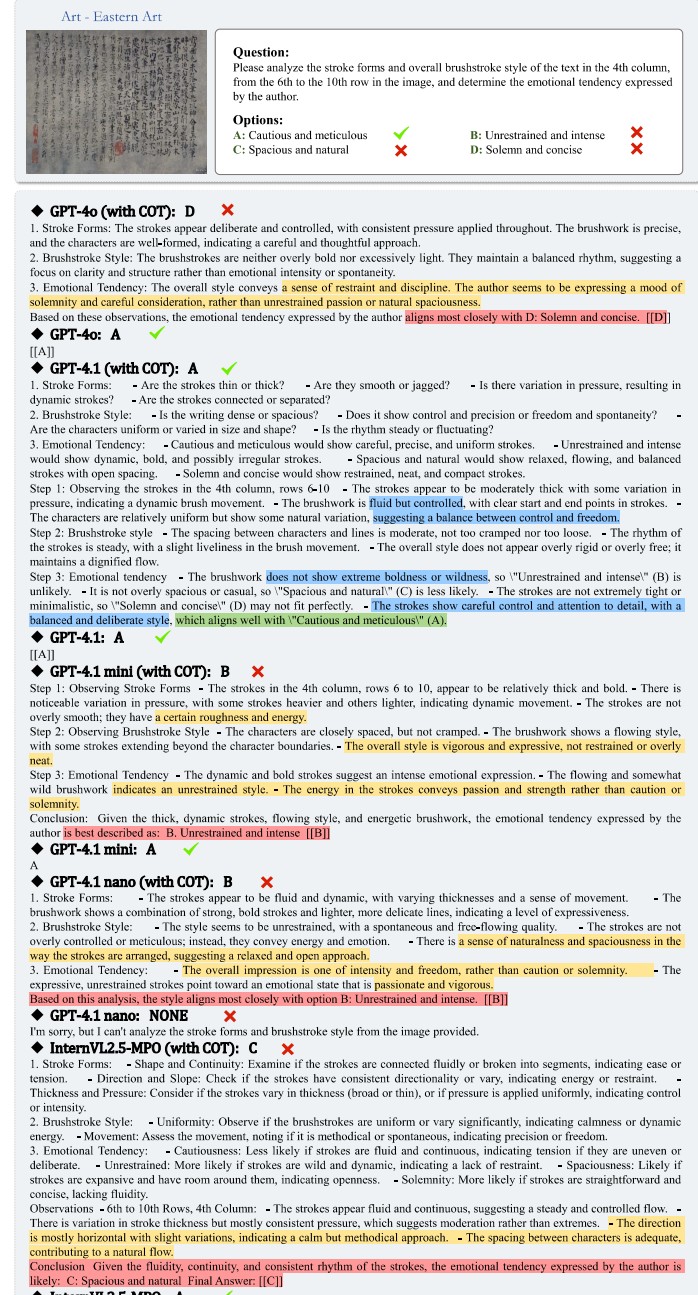

### Art - Eastern Art

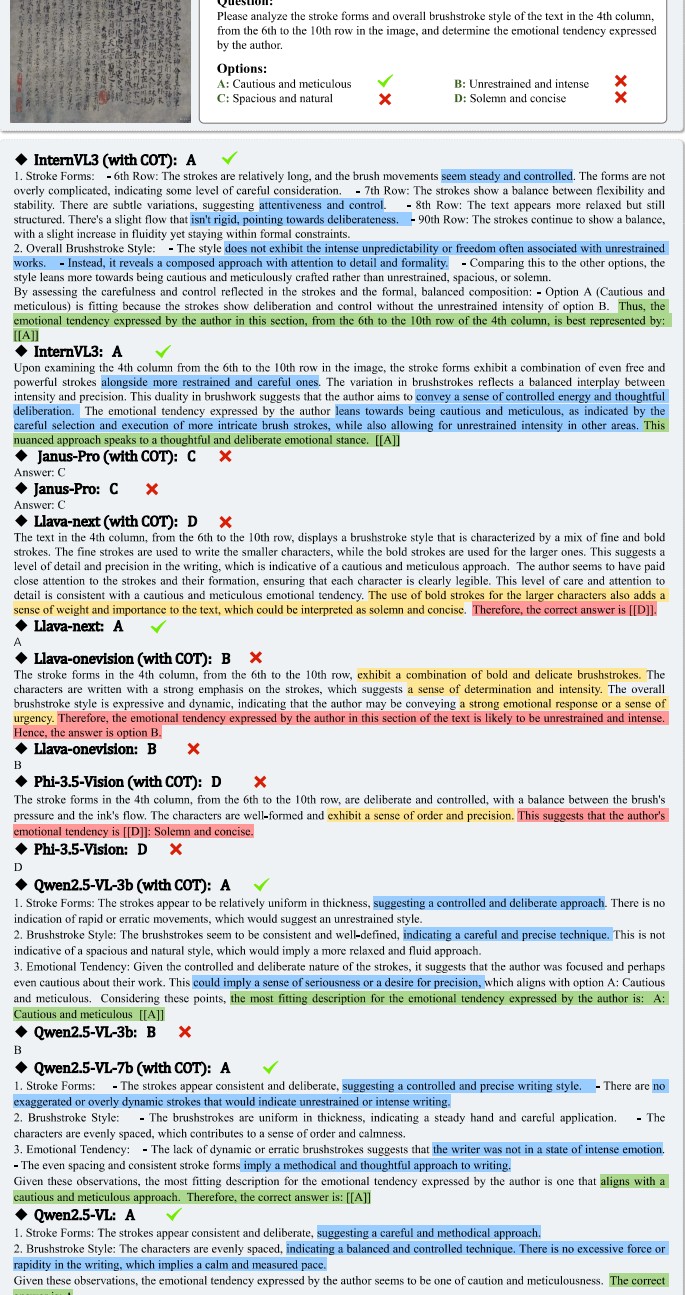

**Question:**
Please analyze the stroke forms and overall brushstroke style of the text in the 4th column, from the 6th to the 10th row in the image, and determine the emotional tendency expressed by the author.

**Options:**
A: Cautious and meticulous ✓   B: Unrestrained and intense ✗
C: Spacious and natural ✗   D: Solemn and concise ✗

◆ **InternVL3 (with COT): A** ✓
1. Stroke Forms: - 6th Row: The strokes are relatively long, and the brush movements seem steady and controlled. The forms are not overly complicated, indicating some level of careful consideration. - 7th Row: The strokes show a balance between flexibility and stability. There are subtle variations, suggesting attentiveness and control. - 8th Row: The text appears more relaxed but still structured. There's a slight flow that isn't rigid, pointing towards deliberateness. - 90th Row: The strokes continue to show a balance, with a slight increase in fluidity yet staying within formal constraints.
2. Overall Brushstroke Style: - The style does not exhibit the intense unpredictability or freedom often associated with unrestrained works. - Instead, it reveals a composed approach with attention to detail and formality. - Comparing this to the other options, the style leans more towards being cautious and meticulously crafted rather than unrestrained, spacious, or solemn.
By assessing the carefulness and control reflected in the strokes and the formal, balanced composition: - Option A (Cautious and meticulous) is fitting because the strokes show deliberation and control without the unrestrained intensity of option B. Thus, the emotional tendency expressed by the author in this section, from the 6th to the 10th row of the 4th column, is best represented by: [[A]]

◆ **InternVL3: A** ✓
Upon examining the 4th column from the 6th to the 10th row in the image, the stroke forms exhibit a combination of even free and powerful strokes alongside more restrained and careful ones. The variation in brushstrokes reflects a balanced interplay between intensity and precision. This duality in brushwork suggests that the author aims to convey a sense of controlled energy and thoughtful deliberation. The emotional tendency expressed by the author leans towards being cautious and meticulous, as indicated by the careful selection and execution of more intricate brush strokes, while also allowing for unrestrained intensity in other areas. This nuanced approach speaks to a thoughtful and deliberate emotional stance. [[A]]

◆ **Janus-Pro (with COT): C** ✗
Answer: C

◆ **Janus-Pro: C** ✗
Answer: C

◆ **Llava-next (with COT): D** ✗
The text in the 4th column, from the 6th to the 10th row, displays a brushstroke style that is characterized by a mix of fine and bold strokes. The fine strokes are used to write the smaller characters, while the bold strokes are used for the larger ones. This suggests a level of detail and precision in the writing, which is indicative of a cautious and meticulous approach. The author seems to have paid close attention to the strokes and their formation, ensuring that each character is clearly legible. This level of care and attention to detail is consistent with a cautious and meticulous emotional tendency. The use of bold strokes for the larger characters also adds a sense of weight and importance to the text, which could be interpreted as solemn and concise. Therefore, the correct answer is [[D]].

◆ **Llava-next: A** ✓
A

◆ **Llava-onevision (with COT): B** ✗
The stroke forms in the 4th column, from the 6th to the 10th row, exhibit a combination of bold and delicate brushstrokes. The characters are written with a strong emphasis on the strokes, which suggests a sense of determination and intensity. The overall brushstroke style is expressive and dynamic, indicating that the author may be conveying a strong emotional response or a sense of urgency. Therefore, the emotional tendency expressed by the author in this section of the text is likely to be unrestrained and intense. Hence, the answer is option B.

◆ **Llava-onevision: B** ✗
B

◆ **Phi-3.5-Vision (with COT): D** ✗
The stroke forms in the 4th column, from the 6th to the 10th row, are deliberate and controlled, with a balance between the brush's pressure and the ink's flow. The characters are well-formed and exhibit a sense of order and precision. This suggests that the author's emotional tendency is [[D]]: Solemn and concise.

◆ **Phi-3.5-Vision: D** ✗
D

◆ **Qwen2.5-VL-3b (with COT): A** ✓
1. Stroke Forms: The strokes appear to be relatively uniform in thickness, suggesting a controlled and deliberate approach. There is no indication of rapid or erratic movements, which would suggest an unrestrained style.
2. Brushstroke Style: The brushstrokes seem to be consistent and well-defined, indicating a careful and precise technique. This is not indicative of a spacious and natural style, which would imply a more relaxed and fluid approach.
3. Emotional Tendency: Given the controlled and deliberate nature of the strokes, it suggests that the author was focused and perhaps even cautious about their work. This could imply a sense of seriousness or a desire for precision, which aligns with option A: Cautious and meticulous. Considering these points, the most fitting description for the emotional tendency expressed by the author is: A: Cautious and meticulous [[A]]

◆ **Qwen2.5-VL-3b: B** ✗
B

◆ **Qwen2.5-VL-7b (with COT): A** ✓
1. Stroke Forms: - The strokes appear consistent and deliberate, suggesting a controlled and precise writing style. - There are no exaggerated or overly dynamic strokes that would indicate unrestrained or intense writing.
2. Brushstroke Style: - The brushstrokes are uniform in thickness, indicating a steady hand and careful application. - The characters are evenly spaced, which contributes to a sense of order and calmness.
3. Emotional Tendency: - The lack of dynamic or erratic brushstrokes suggests that the writer was not in a state of intense emotion. - The even spacing and consistent stroke forms imply a methodical and thoughtful approach to writing.
Given these observations, the most fitting description for the emotional tendency expressed by the author is one that aligns with a cautious and meticulous approach. Therefore, the correct answer is: [[A]]

◆ **Qwen2.5-VL: A** ✓
1. Stroke Forms: The strokes appear consistent and deliberate, suggesting a careful and methodical approach.
2. Brushstroke Style: The characters are evenly spaced, indicating a balanced and controlled technique. There is no excessive force or rapidity in the writing, which implies a calm and measured pace.
Given these observations, the emotional tendency expressed by the author seems to be one of caution and meticulousness. The correct answer is: A

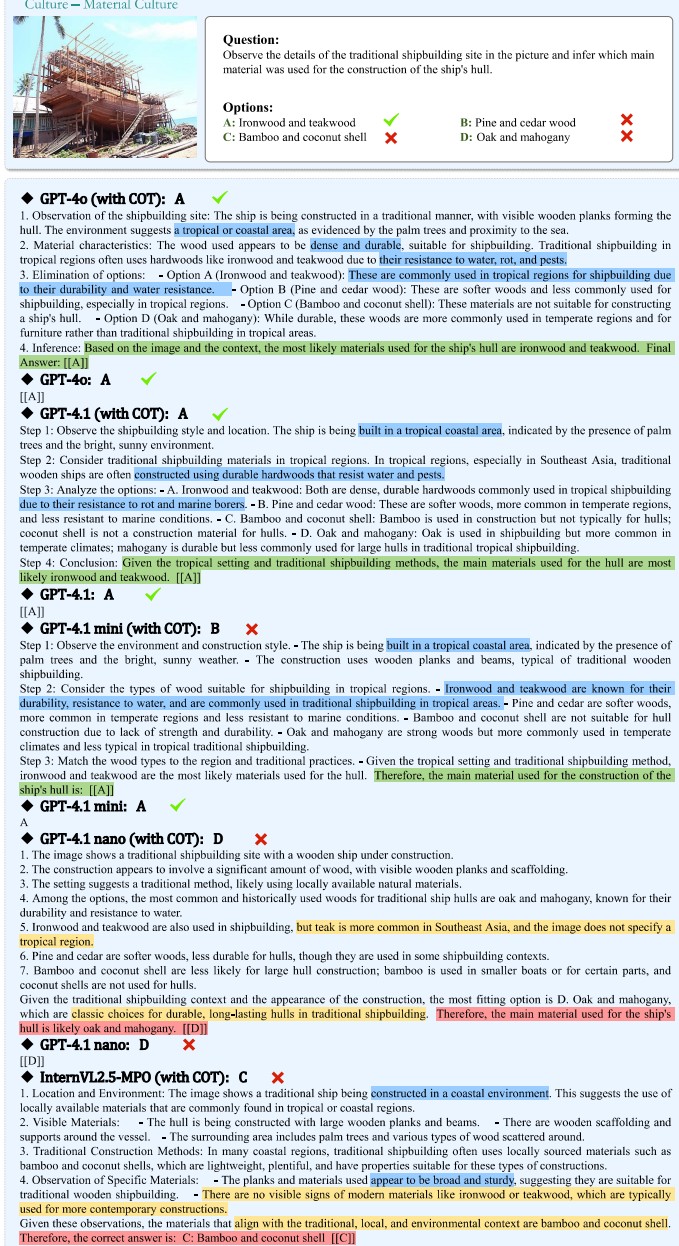

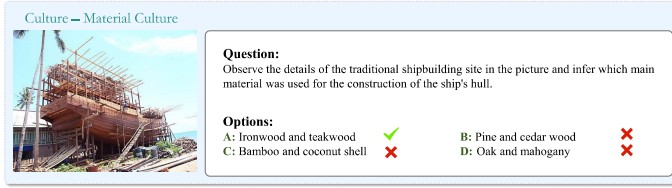

Culture — Material Culture

**Question:**
Observe the details of the traditional shipbuilding site in the picture and infer which main material was used for the construction of the ship's hull.

**Options:**
**A:** Ironwood and teakwood ✓   **B:** Pine and cedar wood ✗
**C:** Bamboo and coconut shell ✗   **D:** Oak and mahogany ✗

---

◆ **InternVL3 (with COT): A** ✓

The image shows shows a large wooden structure, which appears to be a traditional ship under construction. The material used for such constructions is often chosen based on durability and availability. Given the visible large wooden planks and beams, let's analyze the options: - A: Ironwood and teakwood: Both are strong and durable woods commonly used in shipbuilding, especially for those living in areas with access to these materials. - B: Pine and cedar wood: These are also strong and flexible but are generally used for less demanding structures compared to large ships. - C: Bamboo and coconut shell: These materials could be used in more flexible, lightweight constructions but are less common in traditional ship hulls unless in very specific regional contexts. - D: Oak and mahogany: Known for strength and durability, these woods are commonly used in shipbuilding, though they are typically found in more Western locations. The shipbuilding site appears to be using materials accessible in tropical or subtropical regions, and Ironwood and teakwood are often used in such climates due to their durability against water and rot. Considering the context of the image and the type of ship it's likely to be constructed (common in Southeast Asia, for instance), Ironwood and teakwood seem more appropriate than the other options. [[A]]

◆ **InternVL3: C** ✗

From the visible wooden structure in the image, it appears the ship is made from sturdy woods typical for traditional boat construction in many shore regions. C: Bamboo and coconut shell. This option seems plausible due to the natural materials often used in shipbuilding in tropical regions. The scaffolding and raw materials suggest a reliance on locally sourced and easily available materials. [[C]]

◆ **Janus-Pro (with COT): A** ✓

Answer: A

◆ **Janus-Pro: A** ✓

Answer: A

◆ **Llava-next (with COT): C** ✗

Based on the image, the main material used for the construction of the ship's hull appears to be bamboo and coconut shell. The structure is made of these materials, which are commonly used in traditional shipbuilding in certain regions. Therefore, the correct answer is: C

◆ **Llava-next: C** ✗

C

◆ **Llava-onevision (with COT): A** ✓

The image shows a traditional shipbuilding site with a wooden ship under construction. The hull of the ship is made of wood, and the construction is supported by wooden scaffolding. The surrounding area has palm trees and a clear sky, suggesting a tropical location. Based on the visible materials and the traditional construction method, the main material used for the ship's hull is likely to be a hardwood, such as teakwood or oak, which are commonly used in shipbuilding for their durability and resistance to rot and insect damage. Therefore, the answer is [A].

◆ **Llava-onevision: A** ✓

A

◆ **Phi-3.5-Vision (with COT): C** ✗

The image shows a wooden structure with a hull that appears to be in the process of being built. The surrounding environment includes palm trees, which suggests a tropical location. The hull is made of large wooden planks, and there are no visible signs of iron or steel, which are commonly used in modern shipbuilding. The planks are not uniform in color, which could indicate the use of different types of wood. The presence of palm trees and the style of the construction suggest that the ship is being built in a region where such trees are common, which is consistent with tropical or subtropical areas. Considering the options provided: A: Ironwood and teakwood are strong and durable woods, but they are not typically associated with tropical regions. B: Pine and cedar wood are common in many regions, but they are not as strong as ironwood or teakwood. C: Bamboo and coconut shell are both materials that are commonly found in tropical regions and are used in traditional shipbuilding. D: Oak and mahogany are also strong woods but are not as commonly found in tropical regions as bamboo and coconut shell. Based on the environmental clues and the materials visible in the image, the most likely answer is C: Bamboo and coconut shell. These materials are both strong and suitable for shipbuilding and are commonly found in tropical regions. [[C]]

◆ **Phi-3.5-Vision: C** ✗

C

◆ **Qwen2.5-VL-3b (with COT): A** ✓

A: Ironwood and teakwood - These are known for their durability and resistance to marine growth, making them suitable for shipbuilding.
B: Pine and cedar wood - These are also commonly used in shipbuilding due to their strength and resistance to rot.
C: Bamboo and coconut shell - While bamboo can be used in some aspects of shipbuilding, coconut shell is not typically used as a primary material for ship hulls.
D: Oak and mahogany - These are also strong and durable woods, often used in traditional shipbuilding.
Given that the image shows a traditional shipbuilding site, we need to consider the most common and durable materials used historically. Both ironwood and teakwood are highly regarded for their qualities in shipbuilding. However, considering the tropical setting suggested by the palm trees and the type of construction visible in the image, teakwood might be more commonly associated with such environments due to its natural resistance to tropical conditions. Therefore, the most likely answer is: A

◆ **Qwen2.5-VL-3b: B** ✗

B

◆ **Qwen2.5-VL-7b (with COT): B** ✗

1. Ironwood and teakwood: These are known for their durability and resistance to water, making them suitable for shipbuilding. However, they are not commonly found in tropical regions where the ship appears to be located.
2. Pine and cedar wood: These are common materials used in shipbuilding due to their availability and good strength-to-weight ratio. They are often used in both tropical and temperate regions.
3. Bamboo and coconut shell: While bamboo can be used in construction, it is not typically used as a primary material for shipbuilding. Coconut shells might be used for decorative purposes but not for the hull.
4. Oak and mahogany: These are also durable woods used in shipbuilding, especially in colder climates. However, they are less common in tropical regions compared to pine and cedar.
Given the tropical setting suggested by the palm trees and the type of wood visible in the image, which appears to be light-colored and abundant, the most likely materials would be pine and cedar wood. Therefore, the correct answer is: [[B]]

◆ **Qwen2.5-VL: D** ✗

The image shows a ship under construction on a beach, with wooden planks and beams visible. The materials appear to be large and sturdy, which are typical characteristics of oak and mahogany, both commonly used in traditional shipbuilding due to their durability and strength. Therefore, the correct answer is: D

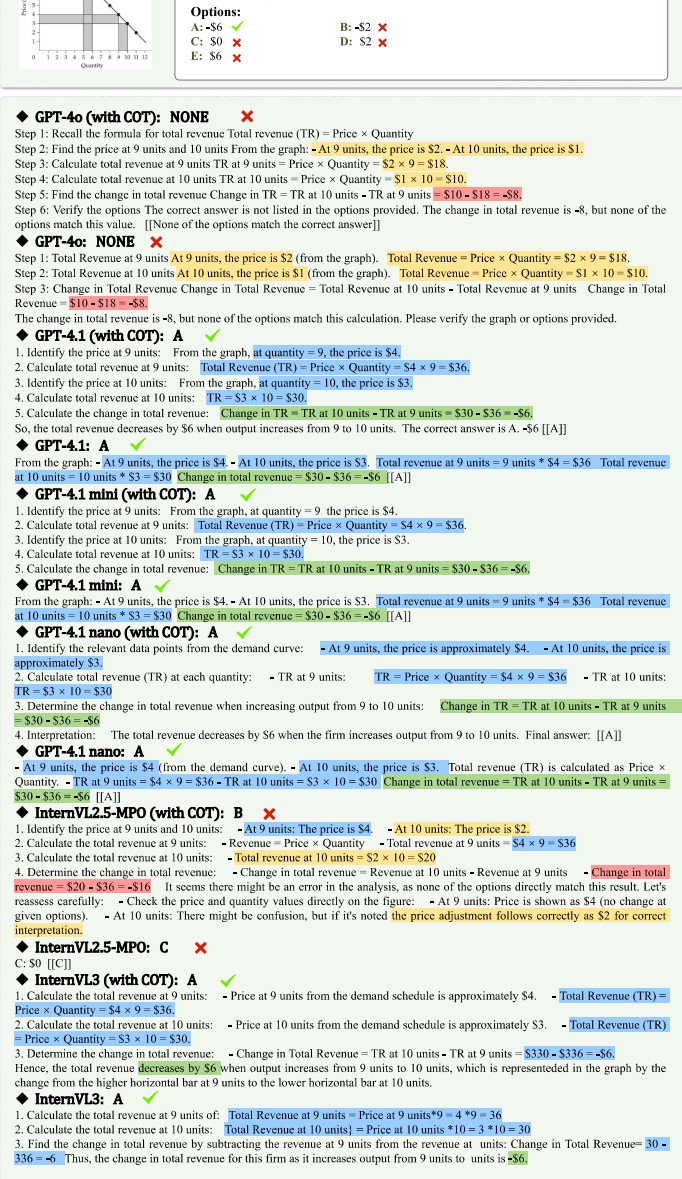

**Economy - Finance**

**Question:**
The figure below shows the demand schedule and demand curve for a product produced by a single-price monopolist. Refer to Figure 10-1. What is the change in total revenue for this firm as it increases output from 9 units to 10 units?

**Options:**
A: -$6 ✓          B: -$2 ✗
C: $0 ✗          D: $2 ✗
E: $6 ✗

◆ **GPT-4o (with COT): NONE** ✗
Step 1: Recall the formula for total revenue Total revenue (TR) = Price × Quantity
Step 2: Find the price at 9 units and 10 units From the graph: - At 9 units, the price is $2. - At 10 units, the price is $1.
Step 3: Calculate total revenue at 9 units TR at 9 units = Price × Quantity = $2 × 9 = $18.
Step 4: Calculate total revenue at 10 units TR at 10 units = Price × Quantity = $1 × 10 = $10.
Step 5: Find the change in total revenue Change in TR = TR at 10 units - TR at 9 units = $10 - $18 = -$8.
Step 6: Verify the options The correct answer is not listed in the options provided. The change in total revenue is -8, but none of the options match this value.   [[None of the options match the correct answer]]

◆ **GPT-4o: NONE** ✗
Step 1: Total Revenue at 9 units At 9 units, the price is $2 (from the graph). Total Revenue = Price × Quantity = $2 × 9 = $18.
Step 2: Total Revenue at 10 units At 10 units, the price is $1 (from the graph). Total Revenue = Price × Quantity = $1 × 10 = $10.
Step 3: Change in Total Revenue Change in Total Revenue = Total Revenue at 10 units - Total Revenue at 9 units   Change in Total Revenue = $10 - $18 = -$8.
The change in total revenue is -8, but none of the options match this calculation. Please verify the graph or options provided.

◆ **GPT-4.1 (with COT): A** ✓
1. Identify the price at 9 units:   From the graph, at quantity = 9, the price is $4.
2. Calculate total revenue at 9 units:   Total Revenue (TR) = Price × Quantity = $4 × 9 = $36.
3. Identify the price at 10 units:   From the graph, at quantity = 10, the price is $3.
4. Calculate total revenue at 10 units:   TR = $3 × 10 = $30.
5. Calculate the change in total revenue:   Change in TR = TR at 10 units - TR at 9 units = $30 - $36 = -$6.
So, the total revenue decreases by $6 when output increases from 9 to 10 units.   The correct answer is A. -$6 [[A]]

◆ **GPT-4.1: A** ✓
From the graph: - At 9 units, the price is $4. - At 10 units, the price is $3.  Total revenue at 9 units = 9 units * $4 = $36   Total revenue at 10 units = 10 units * $3 = $30   Change in total revenue = $30 - $36 = -$6 [[A]]

◆ **GPT-4.1 mini (with COT): A** ✓
1. Identify the price at 9 units:   From the graph, at quantity = 9  the price is $4.
2. Calculate total revenue at 9 units:   Total Revenue (TR) = Price × Quantity = $4 × 9 = $36.
3. Identify the price at 10 units:   From the graph, at quantity = 10, the price is $3.
4. Calculate total revenue at 10 units:   TR = $3 × 10 = $30.
5. Calculate the change in total revenue:   Change in TR = TR at 10 units - TR at 9 units = $30 - $36 = -$6.

◆ **GPT-4.1 mini: A** ✓
From the graph: - At 9 units, the price is $4. - At 10 units, the price is $3.  Total revenue at 9 units = 9 units * $4 = $36   Total revenue at 10 units = 10 units * $3 = $30   Change in total revenue = $30 - $36 = -$6 [[A]]

◆ **GPT-4.1 nano (with COT): A** ✓
1. Identify the relevant data points from the demand curve:   - At 9 units, the price is approximately $4.   - At 10 units, the price is approximately $3.
2. Calculate total revenue (TR) at each quantity:   - TR at 9 units:   TR = Price × Quantity = $4 × 9 = $36   - TR at 10 units: TR = $3 × 10 = $30
3. Determine the change in total revenue when increasing output from 9 to 10 units:   Change in TR = TR at 10 units - TR at 9 units = $30 - $36 = -$6
4. Interpretation:   The total revenue decreases by $6 when the firm increases output from 9 to 10 units.  Final answer: [[A]]

◆ **GPT-4.1 nano: A** ✓
- At 9 units, the price is $4 (from the demand curve). - At 10 units, the price is $3.  Total revenue (TR) is calculated as Price × Quantity.  - TR at 9 units = $4 × 9 = $36 - TR at 10 units = $3 × 10 = $30  Change in total revenue = TR at 10 units - TR at 9 units = $30 - $36 = -$6 [[A]]

◆ **InternVL2.5-MPO (with COT): B** ✗
1. Identify the price at 9 units and 10 units:   - At 9 units: The price is $4.   - At 10 units: The price is $2.
2. Calculate the total revenue at 9 units:   - Revenue = Price × Quantity   - Total revenue at 9 units = $4 × 9 = $36
3. Calculate the total revenue at 10 units:   - Total revenue at 10 units = $2 × 10 = $20
4. Determine the change in total revenue:   - Change in total revenue = Revenue at 10 units - Revenue at 9 units   - Change in total revenue = $20 - $36 = -$16   It seems there might be an error in the analysis, as none of the options directly match this result. Let's reassess carefully:   - Check the price and quantity values directly on the figure:   - At 9 units: Price is shown as $4 (no change at given options).   - At 10 units: There might be confusion, but if it's noted the price adjustment follows correctly as $2 for correct interpretation.

◆ **InternVL2.5-MPO: C** ✗
C: $0 [[C]]

◆ **InternVL3 (with COT): A** ✓
1. Calculate the total revenue at 9 units:   - Price at 9 units from the demand schedule is approximately $4.   - Total Revenue (TR) = Price × Quantity = $4 × 9 = $36.
2. Calculate the total revenue at 10 units:   - Price at 10 units from the demand schedule is approximately $3.   - Total Revenue (TR) = Price × Quantity = $3 × 10 = $30.
3. Determine the change in total revenue:   - Change in Total Revenue = TR at 10 units - TR at 9 units = $330 - $336 = -$6.
Hence, the total revenue decreases by $6 when output increases from 9 to 10 units, which is representeded in the graph by the change from the higher horizontal bar at 9 units to the lower horizontal bar at 10 units.

◆ **InternVL3: A** ✓
1. Calculate the total revenue at 9 units of:   Total Revenue at 9 units = Price at 9 units*9 = 4 *9 = 36
2. Calculate the total revenue at 10 units:   Total Revenue at 10 units} = Price at 10 units *10 = 3 *10 = 30
3. Find the change in total revenue by subtracting the revenue at 9 units from the revenue at  units: Change in Total Revenue= 30 - 336 = -6   Thus, the change in total revenue for this firm as it increases output from 9 units to  units is -$6.

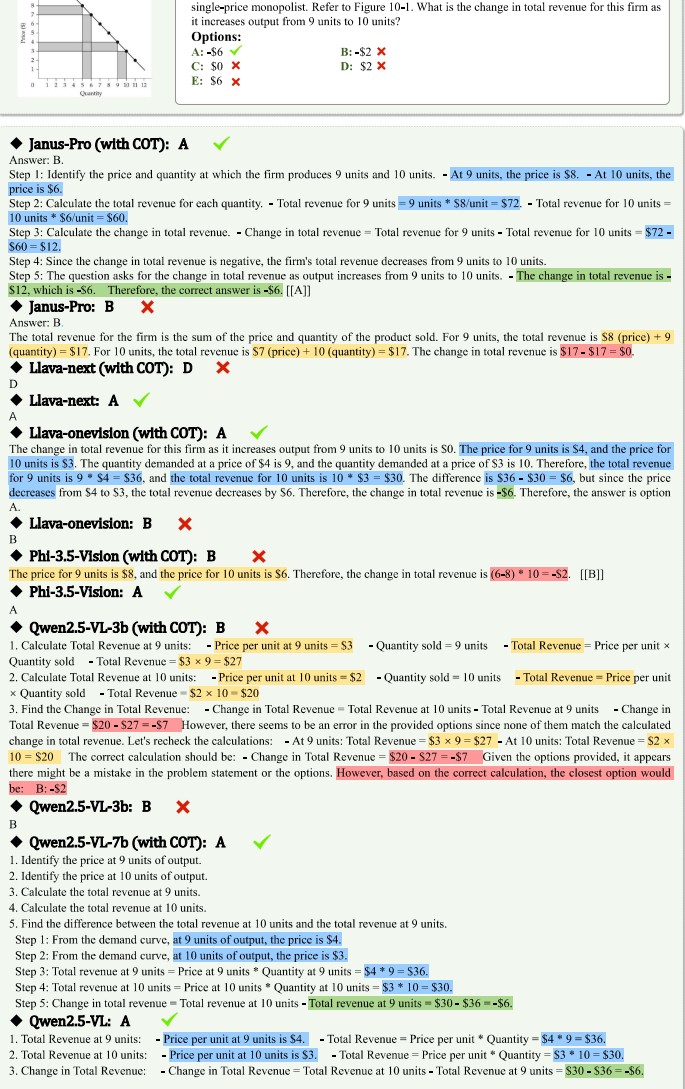

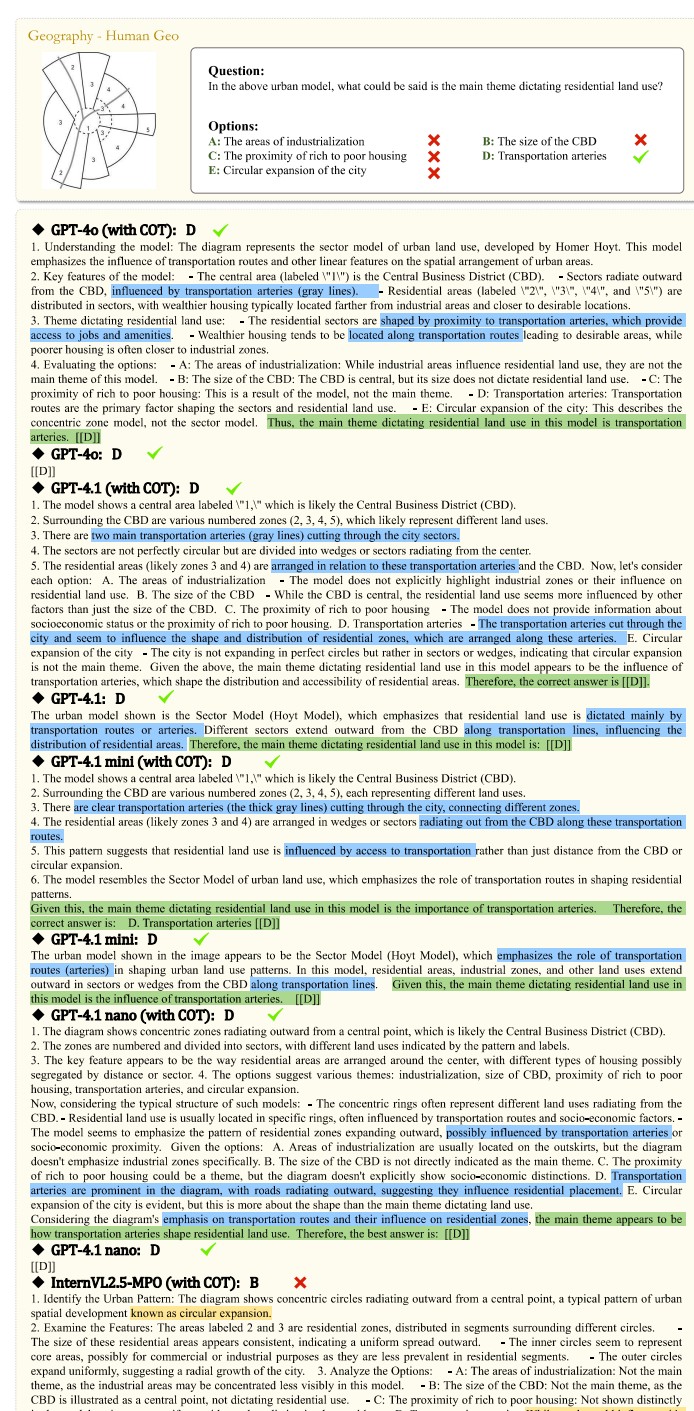

**Geography - Human Geo**

**Question:**
In the above urban model, what could be said is the main theme dictating residential land use?

**Options:**
A: The areas of industrialization ✗  B: The size of the CBD ✗
C: The proximity of rich to poor housing ✗  D: Transportation arteries ✓
E: Circular expansion of the city ✗

◆ **GPT-4o (with COT): D** ✓
1. Understanding the model: The diagram represents the sector model of urban land use, developed by Homer Hoyt. This model emphasizes the influence of transportation routes and other linear features on the spatial arrangement of urban areas.
2. Key features of the model: - The central area (labeled \"1\") is the Central Business District (CBD). - Sectors radiate outward from the CBD, influenced by transportation arteries (gray lines). - Residential areas (labeled \"2\", \"3\", \"4\", and \"5\") are distributed in sectors, with wealthier housing typically located farther from industrial areas and closer to desirable locations.
3. Theme dictating residential land use: - The residential sectors are shaped by proximity to transportation arteries, which provide access to jobs and amenities. - Wealthier housing tends to be located along transportation routes leading to desirable areas, while poorer housing is often closer to industrial zones.
4. Evaluating the options: - A: The areas of industrialization: While industrial areas influence residential land use, they are not the main theme of this model. - B: The size of the CBD: The CBD is central, but its size does not dictate residential land use. - C: The proximity of rich to poor housing: This is a result of the model, not the main theme. - D: Transportation arteries: Transportation routes are the primary factor shaping the sectors and residential land use. - E: Circular expansion of the city: This describes the concentric zone model, not the sector model. Thus, the main theme dictating residential land use in this model is transportation arteries. [[D]]

◆ **GPT-4o: D** ✓
[[D]]

◆ **GPT-4.1 (with COT): D** ✓
1. The model shows a central area labeled \"1,\" which is likely the Central Business District (CBD).
2. Surrounding the CBD are various numbered zones (2, 3, 4, 5), which likely represent different land uses.
3. There are two main transportation arteries (gray lines) cutting through the city sectors.
4. The sectors are not perfectly circular but are divided into wedges or sectors radiating from the center.
5. The residential areas (likely zones 3 and 4) are arranged in relation to these transportation arteries and the CBD. Now, let's consider each option: A. The areas of industrialization - The model does not explicitly highlight industrial zones or their influence on residential land use. B. The size of the CBD - While the CBD is central, the residential land use seems more influenced by other factors than just the size of the CBD. C. The proximity of rich to poor housing - The model does not provide information about socioeconomic status or the proximity of rich to poor housing. D. Transportation arteries - The transportation arteries cut through the city and seem to influence the shape and distribution of residential zones, which are arranged along these arteries. E. Circular expansion of the city - The city is not expanding in perfect circles but rather in sectors or wedges, indicating that circular expansion is not the main theme. Given the above, the main theme dictating residential land use in this model appears to be the influence of transportation arteries, which shape the distribution and accessibility of residential areas. Therefore, the correct answer is [[D]].

◆ **GPT-4.1: D** ✓
The urban model shown is the Sector Model (Hoyt Model), which emphasizes that residential land use is dictated mainly by transportation routes or arteries. Different sectors extend outward from the CBD along transportation lines, influencing the distribution of residential areas. Therefore, the main theme dictating residential land use in this model is: [[D]]

◆ **GPT-4.1 mini (with COT): D** ✓
1. The model shows a central area labeled \"1,\" which is likely the Central Business District (CBD).
2. Surrounding the CBD are various numbered zones (2, 3, 4, 5), each representing different land uses.
3. There are clear transportation arteries (the thick gray lines) cutting through the city, connecting different zones.
4. The residential areas (likely zones 3 and 4) are arranged in wedges or sectors radiating out from the CBD along these transportation routes.
5. This pattern suggests that residential land use is influenced by access to transportation rather than just distance from the CBD or circular expansion.
6. The model resembles the Sector Model of urban land use, which emphasizes the role of transportation routes in shaping residential patterns.
Given this, the main theme dictating residential land use in this model is the importance of transportation arteries. Therefore, the correct answer is: D. Transportation arteries [[D]]

◆ **GPT-4.1 mini: D** ✓
The urban model shown in the image appears to be the Sector Model (Hoyt Model), which emphasizes the role of transportation routes (arteries) in shaping urban land use patterns. In this model, residential areas, industrial zones, and other land uses extend outward in sectors or wedges from the CBD along transportation lines. Given this, the main theme dictating residential land use in this model is the influence of transportation arteries. [[D]]

◆ **GPT-4.1 nano (with COT): D** ✓
1. The diagram shows concentric zones radiating outward from a central point, which is likely the Central Business District (CBD).
2. The zones are numbered and divided into sectors, with different land uses indicated by the numbers and labels.
3. The key feature appears to be the way residential areas are arranged around the center, with different types of housing possibly segregated by distance or sector. 4. The options suggest various themes: industrialization, size of CBD, proximity of rich to poor housing, transportation arteries, and circular expansion.
Now, considering the typical structure of such models: - The concentric rings often represent different land uses radiating from the CBD. - Residential land use is usually located in specific rings, often influenced by transportation routes and socio-economic factors. - The model seems to emphasize the pattern of residential zones expanding outward, possibly influenced by transportation arteries or socio-economic proximity. Given the options: A. Areas of industrialization are usually located on the outskirts, but the diagram doesn't emphasize industrial zones specifically. B. The size of the CBD is not directly indicated as the main theme. C. The proximity of rich to poor housing could be a theme, but the diagram doesn't explicitly show socio-economic distinctions. D. Transportation arteries are prominent in the diagram, with roads radiating outward, suggesting they influence residential placement. E. Circular expansion of the city is evident, but this is more about the shape than the main theme dictating land use.
Considering the diagram's emphasis on transportation routes and their influence on residential zones, the main theme appears to be how transportation arteries shape residential land use. Therefore, the best answer is: [[D]]

◆ **GPT-4.1 nano: D** ✓
[[D]]

◆ **InternVL2.5-MPO (with COT): B** ✗
1. Identify the Urban Pattern: The diagram shows concentric circles radiating outward from a central point, a typical pattern of urban spatial development known as circular expansion.
2. Examine the Features: The areas labeled 2 and 3 are residential zones, distributed in segments surrounding different zones. - The size of these residential areas appears consistent, indicating a uniform spread outward. - The inner circles seem to represent core areas, possibly for commercial or industrial purposes as they are less prevalent in residential segments. - The outer circles expand uniformly, suggesting a radial growth of the city. 3. Analyze the Options: - A: The areas of industrialization: Not the main theme, as the industrial areas may be concentrated less visibly in this model. - B: The size of the CBD: Not the main theme, as the CBD is illustrated as a central point, not dictating residential use. - C: The proximity of rich to poor housing: Not shown distinctly in the model, as it appears uniform without clear distinction by wealth. - D: Transportation arteries: While roads could influence, it's not the primary driver shown here. - E: Circular expansion of the city: Correctly describes the pattern of outward, circular growth visible in the layout of residential and possibly other zones.
Considering the visual cues and patterns, the most fitting theme is the circular expansion of the city. Thus, the correct answer is: [[E]]

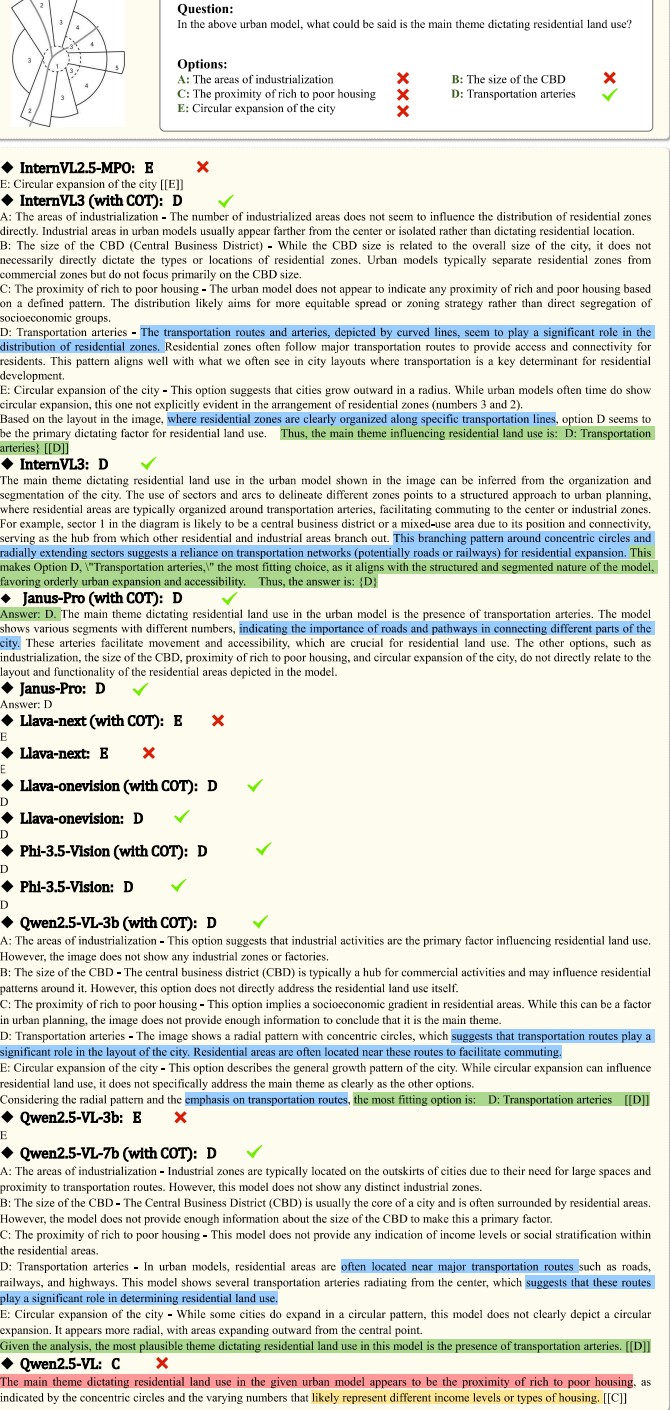

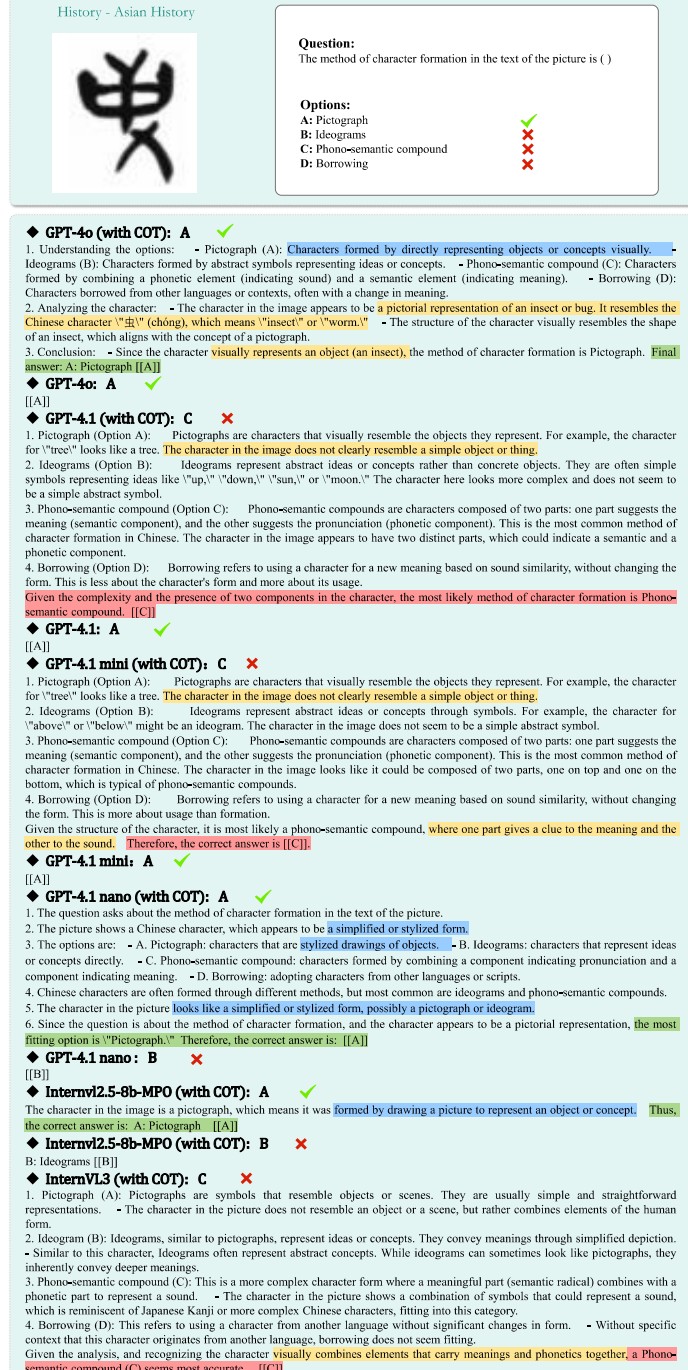

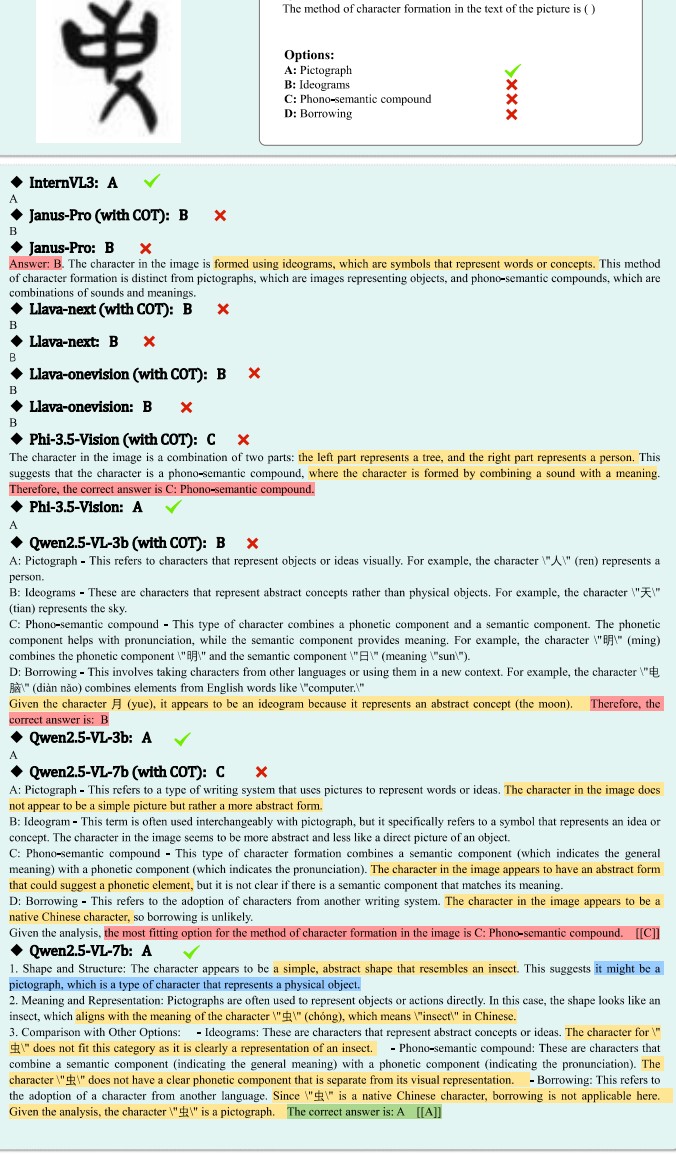

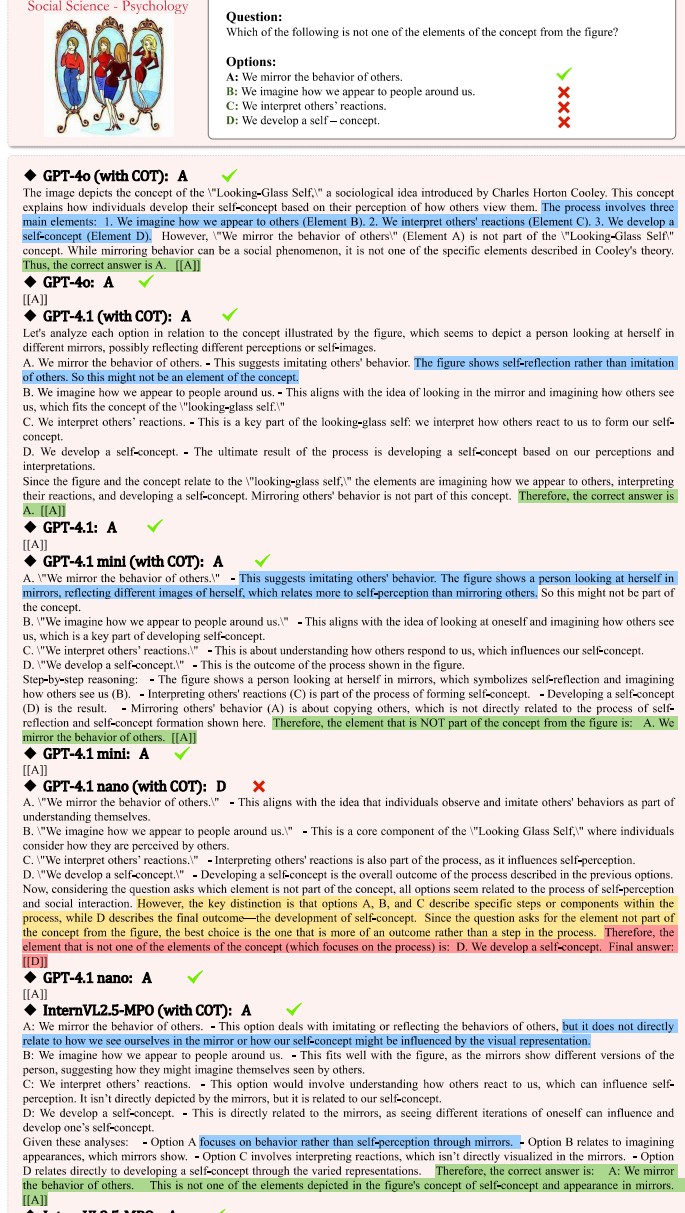

**Social Science - Psychology**

**Question:**
Which of the following is not one of the elements of the concept from the figure?

**Options:**
**A:** We mirror the behavior of others. ✓
**B:** We imagine how we appear to people around us. ✗
**C:** We interpret others' reactions. ✗
**D:** We develop a self – concept. ✗

◆ **GPT-4o (with COT): A** ✓
The image depicts the concept of the \"Looking-Glass Self,\" a sociological idea introduced by Charles Horton Cooley. This concept explains how individuals develop their self-concept based on their perception of how others view them. The process involves three main elements: 1. We imagine how we appear to others (Element B). 2. We interpret others' reactions (Element C). 3. We develop a self-concept (Element D). However, \"We mirror the behavior of others\" (Element A) is not part of the \"Looking-Glass Self\" concept. While mirroring behavior can be a social phenomenon, it is not one of the specific elements described in Cooley's theory. Thus, the correct answer is A. **[[A]]**

◆ **GPT-4o: A** ✓
[[A]]

◆ **GPT-4.1 (with COT): A** ✓
Let's analyze each option in relation to the concept illustrated by the figure, which seems to depict a person looking at herself in different mirrors, possibly reflecting different perceptions or self-images.
A. We mirror the behavior of others. - This suggests imitating others' behavior. The figure shows self-reflection rather than imitation of others. So this might not be an element of the concept.
B. We imagine how we appear to people around us. - This aligns with the idea of looking in the mirror and imagining how others see us, which fits the concept of the \"looking-glass self.\"
C. We interpret others' reactions. - This is a key part of the looking-glass self: we interpret how others react to us to form our self-concept.
D. We develop a self-concept. - The ultimate result of the process is developing a self-concept based on our perceptions and interpretations.
Since the figure and the concept relate to the \"looking-glass self,\" the elements are imagining how we appear to others, interpreting their reactions, and developing a self-concept. Mirroring others' behavior is not part of this concept. Therefore, the correct answer is A. **[[A]]**

◆ **GPT-4.1: A** ✓
[[A]]

◆ **GPT-4.1 mini (with COT): A** ✓
A. \"We mirror the behavior of others.\" - This suggests imitating others' behavior. The figure shows a person looking at herself in mirrors, reflecting different images of herself, which relates more to self-perception than mirroring others. So this might not be part of the concept.
B. \"We imagine how we appear to people around us.\" - This aligns with the idea of looking at oneself and imagining how others see us, which is a key part of developing self-concept.
C. \"We interpret others' reactions.\" - This is about understanding how others respond to us, which influences our self-concept.
D. \"We develop a self-concept.\" - This is the outcome of the process shown in the figure.
Step-by-step reasoning: - The figure shows a person looking at herself in mirrors, which symbolizes self-reflection and imagining how others see us (B). - Interpreting others' reactions (C) is part of the process of forming self-concept. - Developing a self-concept (D) is the result. - Mirroring others' behavior (A) is about copying others, which is not directly related to the process of self-reflection and self-concept formation shown here. Therefore, the element that is NOT part of the concept from the figure is: A. We mirror the behavior of others. **[[A]]**

◆ **GPT-4.1 mini: A** ✓
[[A]]

◆ **GPT-4.1 nano (with COT): D** ✗
A. \"We mirror the behavior of others.\" - This aligns with the idea that individuals observe and imitate others' behaviors as part of understanding themselves.
B. \"We imagine how we appear to people around us.\" - This is a core component of the \"Looking Glass Self,\" where individuals consider how they are perceived by others.
C. \"We interpret others' reactions.\" - Interpreting others' reactions is also part of the process, as it influences self-perception.
D. \"We develop a self-concept.\" - Developing a self-concept is the overall outcome of the process described in the previous options.
Now, considering the question asks which element is not part of the concept, all options seem related to the process of self-perception and social interaction. However, the key distinction is that options A, B, and C describe specific steps or components in the process, while D describes the final outcome—the development of self-concept. Since the question asks for the element not part of the concept from the figure, the best choice is the one that is more of an outcome rather than a step in the process. Therefore, the element that is not one of the elements of the concept (which focuses on the process) is: D. We develop a self-concept. Final answer: **[[D]]**

◆ **GPT-4.1 nano: A** ✓
[[A]]

◆ **InternVL2.5-MPO (with COT): A** ✓
A: We mirror the behavior of others. - This option deals with imitating or reflecting the behaviors of others, but it does not directly relate to how we see ourselves in the mirror or how our self-concept might be influenced by the visual representation.
B: We imagine how we appear to people around us. - This fits well with the figure, as the mirrors show different versions of the person, suggesting how they might imagine themselves seen by others.
C: We interpret others' reactions. - This option would involve understanding how others react to us, which can influence self-perception. It isn't directly depicted by the mirrors, but it is related to our self-concept.
D: We develop a self-concept. - This is directly related to the mirrors, as seeing different iterations of oneself can influence and develop one's self-concept.
Given these analyses: - Option A focuses on behavior rather than self-perception through mirrors. - Option B relates to imagining appearances, which mirrors show. - Option C involves interpreting reactions, which isn't directly visualized in the mirrors. - Option D relates directly to developing a self-concept through the varied representations. Therefore, the correct answer is: A: We mirror the behavior of others. This is not one of the elements depicted in the figure's concept of self-concept and appearance in mirrors. **[[A]]**

◆ **InternVL2.5-MPO: A** ✓
A: We mirror the behavior of others. [[A]]

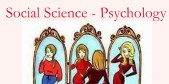

**Social Science - Psychology**

**Question:**
Which of the following is not one of the elements of the concept from the figure?

**Options:**
**A:** We mirror the behavior of others. ✓
**B:** We imagine how we appear to people around us. ✗
**C:** We interpret others' reactions. ✗
**D:** We develop a self – concept. ✗

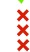

◆ **InternVL3 (with COT): C** ✗
The figure shows shows: 1 - Three mirrors with an illustration of the same girl in different outfits, standing in front of them. - This illustration emphasizes self-image and perception of oneself. 1 - We mirror the behavior of others. - The concept illustrated does not focus on mirroring the behavior of others. The mirrors show different outfits, emphasizing personal perception and appearance, not reflective behavior. - We imagine how we appear to people around us. - This is related to the concept as the girl is likely imagining different looks, how she appears in these different outfits. - We interpret others' reactions. - This option does not fit as the image does not show or indicate any interaction with others to interpret reactions. - We develop a self-concept. - This is relevant as the girl is shown in different outfits, possibly representing different facets or roles of her self-concept. The concept in the figure relates to self-image, perception, and self-concept, but it doesn't encompass interpreting others' reactions or mirroring behaviors. The correct answer is: [C]

◆ **InternVL3: A** ✓
The figure shows illustrates the concept of self-perception through mirrors, highlighting how individuals might perceive themselves and others' reactions. Among the options provided: A: We mirror the behavior of others. - This is related to observational learning, not directly shown in the figure. B: We imagine how we appear to people around us. - This is closely related to how the figure shows depicts individuals evaluating themselves in mirrors, simulating others' perceptions. C: We interpret others' reactions. - This can be implied as part of self-perception, especially in understanding how others might view. D: We develop a self-concept. - This is a direct outcome of perceiving oneself, which is the core of the concept illustrated by the mirrors. Thus, the correct answer, which is not depicted in the figure, is: A: We mirror the behavior of others. [[A]]

◆ **Janus-Pro (with COT): A** ✓
Answer: A

◆ **Janus-Pro: D** ✗
Answer: D

◆ **Llava-next (with COT): D** ✗
The correct answer is [[D]]. The concept being depicted in the image is self-concept, which refers to how an individual perceives themselves in relation to their appearance. The image shows three women looking at their reflections in mirrors, which is a visual representation of self-concept. The other options are not directly related to self-concept: A. We mirror the behavior of others refers to the process of imitation or learning from others, which is not the focus of the image. B. We imagine how we appear to people around us is a part of self-concept, but it is not the central idea in the image. C. We interpret others' reactions is a part of social cognition, which is not the main theme of the image. E. We develop a self-concept is the overarching concept that includes the other elements, but it is not the specific element being depicted in the image.

◆ **Llava-next: B** ✗
B

◆ **Llava-onevision (with COT): A** ✓
A

◆ **Llava-onevision: A** ✓
A

◆ **Phi-3.5-Vision (with COT): E** ✗
The correct answer is [[E]]. The figure does not mention the development of a self-concept as one of its elements. The other options (A, B, and C) are all mentioned as elements of the concept.

◆ **Phi-3.5-Vision: A** ✓
A

◆ **Qwen2.5-VL-3b (with COT): C** ✗
A: \"We mirror the behavior of others.\" - This option suggests that individuals observe and imitate the actions of others. This aligns with the concept depicted in the image, as it shows a person observing themselves in mirrors.
B: \"We imagine how we appear to people around us.\" - This option implies that individuals consider their appearance and how they are perceived by others. This is also consistent with the image, as it shows a person looking at themselves in mirrors.
C: \"We interpret others' reactions.\" - This option involves understanding and interpreting the reactions of others. While this could be inferred from the image, it is more about understanding others' perspectives rather than directly mirroring their behavior or imagining their appearance.
D: \"We develop a self-concept.\" - This option refers to forming an understanding of oneself based on various experiences and perceptions. The image does not explicitly show this process but can be inferred as part of the overall concept of self-awareness.
Given the analysis, the option that is least directly related to the concept depicted in the image is: C: \"We interpret others' reactions.\" Therefore, the final answer is C.

◆ **Qwen2.5-VL-3b: E** ✗
E

◆ **Qwen2.5-VL-7b (with COT): C** ✗
- Option A: \"We mirror the behavior of others.\" This suggests imitation or copying behaviors, which can be seen in the figure as the woman is mimicking different poses and expressions in front of the mirrors.
- Option B: \"We imagine how we appear to people around us.\" This implies self-reflection and perception, which aligns with the act of looking at oneself in the mirror and considering how one appears.
- Option C: \"We interpret others' reactions.\" The figure does not show any interaction with other people or reactions from others; it only shows the woman looking at herself in the mirrors.
- Option D: \"We develop a self-concept.\" This involves forming an understanding of oneself based on self-reflection, which is evident in the woman's various poses and expressions.
Given this analysis, the element that is not present in the figure is: [[C]]

◆ **Qwen2.5-VL: A** ✓
The image depicts a person looking at themselves in three different mirrors, which suggests a focus on self-perception and self-image. The options provided relate to different aspects of self-concept development: - A: We mirror the behavior of others. - B: We imagine how we appear to people around us. - C: We interpret others' reactions. - D: We develop a self-concept. Given these descriptions, option A does not seem to be directly related to the process of developing a self-concept as depicted in the image. Instead, it seems more aligned with imitation or social learning rather than self-reflection. Therefore, the correct answer is: [[A]]

**Geography - Regional Geo**

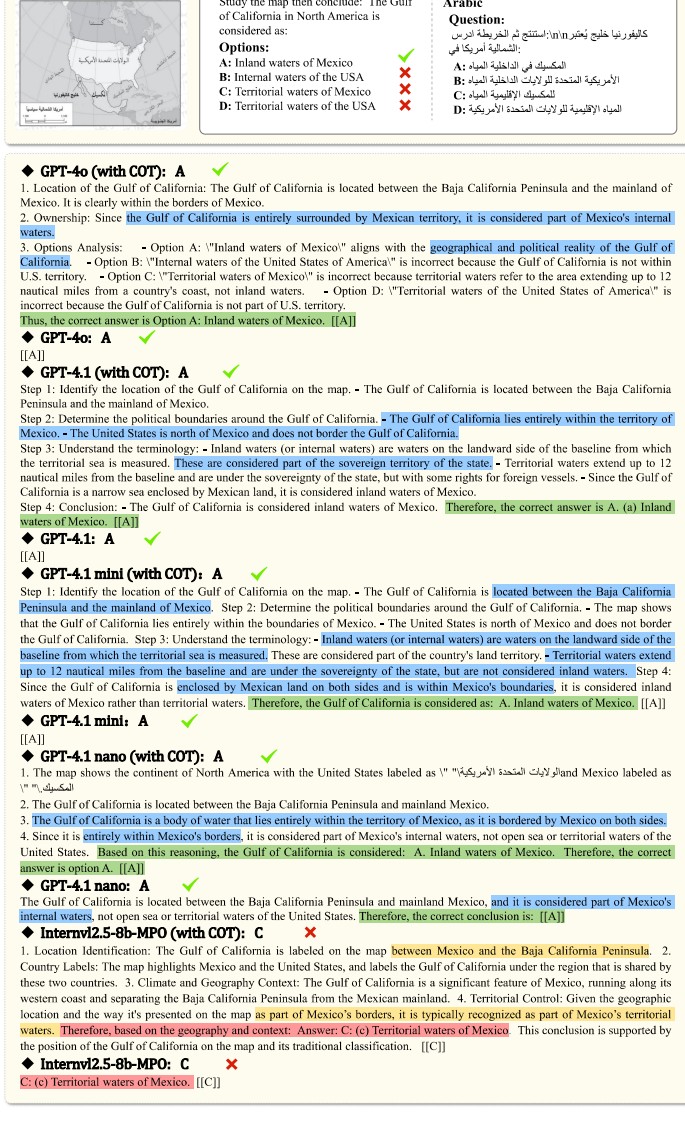

**Question:**
Study the map then conclude: The Gulf of California in North America is considered as:
**Options:**
**A:** Inland waters of Mexico ✓
**B:** Internal waters of the USA ✗
**C:** Territorial waters of Mexico ✗
**D:** Territorial waters of the USA ✗

**\*The source language for this issue:**
**Arabic**
**Question:**
ادرس الخريطة ثم استنتج:\n\n يُعتبر خليج كاليفورنيا في أمريكا الشمالية:
**A:** المياه الداخلية في المكسيك
**B:** المياه الداخلية للولايات المتحدة الأمريكية
**C:** المياه الإقليمية للمكسيك
**D:** المياه الإقليمية للولايات المتحدة الأمريكية

◆ **GPT-4o (with COT): A** ✓
1. Location of the Gulf of California: The Gulf of California is located between the Baja California Peninsula and the mainland of Mexico. It is clearly within the borders of Mexico.
2. Ownership: Since the Gulf of California is entirely surrounded by Mexican territory, it is considered part of Mexico's internal waters.
3. Options Analysis: - Option A: \"Inland waters of Mexico\" aligns with the geographical and political reality of the Gulf of California. - Option B: \"Internal waters of the United States of America\" is incorrect because the Gulf of California is not within U.S. territory. - Option C: \"Territorial waters of Mexico\" is incorrect because territorial waters refer to the area extending up to 12 nautical miles from a country's coast, not inland waters. - Option D: \"Territorial waters of the United States of America\" is incorrect because the Gulf of California is not part of U.S. territory.
Thus, the correct answer is Option A: Inland waters of Mexico. [[A]]

◆ **GPT-4o: A** ✓
[[A]]

◆ **GPT-4.1 (with COT): A** ✓
Step 1: Identify the location of the Gulf of California on the map. - The Gulf of California is located between the Baja California Peninsula and the mainland of Mexico.
Step 2: Determine the political boundaries around the Gulf of California. - The Gulf of California lies entirely within the territory of Mexico. - The United States is north of Mexico and does not border the Gulf of California.
Step 3: Understand the terminology: - Inland waters (or internal waters) are waters on the landward side of the baseline from which the territorial sea is measured. These are considered part of the sovereign territory of the state. - Territorial waters extend up to 12 nautical miles from the baseline and are under the sovereignty of the state, but with some rights for foreign vessels. - Since the Gulf of California is a narrow sea enclosed by Mexican land, it is considered inland waters of Mexico.
Step 4: Conclusion: - The Gulf of California is considered inland waters of Mexico. Therefore, the correct answer is A. (a) Inland waters of Mexico. [[A]]

◆ **GPT-4.1: A** ✓
[[A]]

◆ **GPT-4.1 mini (with COT): A** ✓
Step 1: Identify the location of the Gulf of California on the map. - The Gulf of California is located between the Baja California Peninsula and the mainland of Mexico. Step 2: Determine the political boundaries around the Gulf of California. - The map shows that the Gulf of California lies entirely within the boundaries of Mexico. - The United States is north of Mexico and does not border the Gulf of California. Step 3: Understand the terminology: - Inland waters (or internal waters) are waters on the landward side of the baseline from which the territorial sea is measured. These are considered part of the country's land territory. - Territorial waters extend up to 12 nautical miles from the baseline and are under the sovereignty of the state, but are not considered inland waters. Step 4: Since the Gulf of California is enclosed by Mexican land on both sides and is within Mexico's boundaries, it is considered inland waters of Mexico rather than territorial waters. Therefore, the Gulf of California is considered as: A. Inland waters of Mexico. [[A]]

◆ **GPT-4.1 mini: A** ✓
[[A]]

◆ **GPT-4.1 nano (with COT): A** ✓
1. The map shows the continent of North America with the United States labeled as \" "والولايات المتحدة الأمريكية" \" and Mexico labeled as \" "المكسيك" \".
2. The Gulf of California is located between the Baja California Peninsula and mainland Mexico.
3. The Gulf of California is a body of water that lies entirely within the territory of Mexico, as it is bordered by Mexico on both sides.
4. Since it is entirely within Mexico's borders, it is considered part of Mexico's internal waters, not open sea or territorial waters of the United States. Based on this reasoning, the Gulf of California is considered: A. Inland waters of Mexico. Therefore, the correct answer is option A. [[A]]

◆ **GPT-4.1 nano: A** ✓
The Gulf of California is located between the Baja California Peninsula and mainland Mexico, and it is considered part of Mexico's internal waters, not open sea or territorial waters of the United States. Therefore, the correct conclusion is: [[A]]

◆ **Internvl2.5-8b-MPO (with COT): C** ✗
1. Location Identification: The Gulf of California is labeled on the map between Mexico and the Baja California Peninsula. 2. Country Labels: The map highlights Mexico and the United States, and labels the Gulf of California under the region that is shared by these two countries. 3. Climate and Geography Context: The Gulf of California is a significant feature of Mexico, running along its western coast and separating the Baja California Peninsula from the Mexican mainland. 4. Territorial Control: Given the geographic location and the way it's presented on the map as part of Mexico's borders, it is typically recognized as part of Mexico's territorial waters. Therefore, based on the geography and context: Answer: C: (c) Territorial waters of Mexico. This conclusion is supported by the position of the Gulf of California on the map and its traditional classification. [[C]]

◆ **Internvl2.5-8b-MPO: C** ✗
C: (c) Territorial waters of Mexico. [[C]]

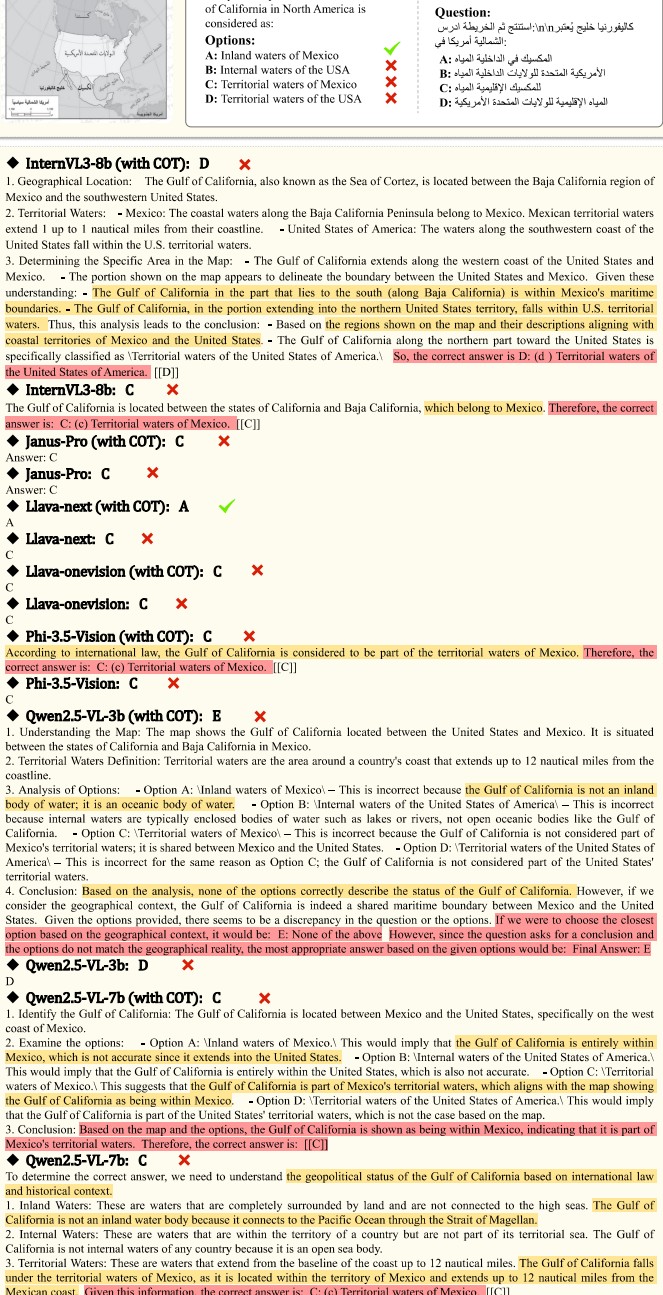

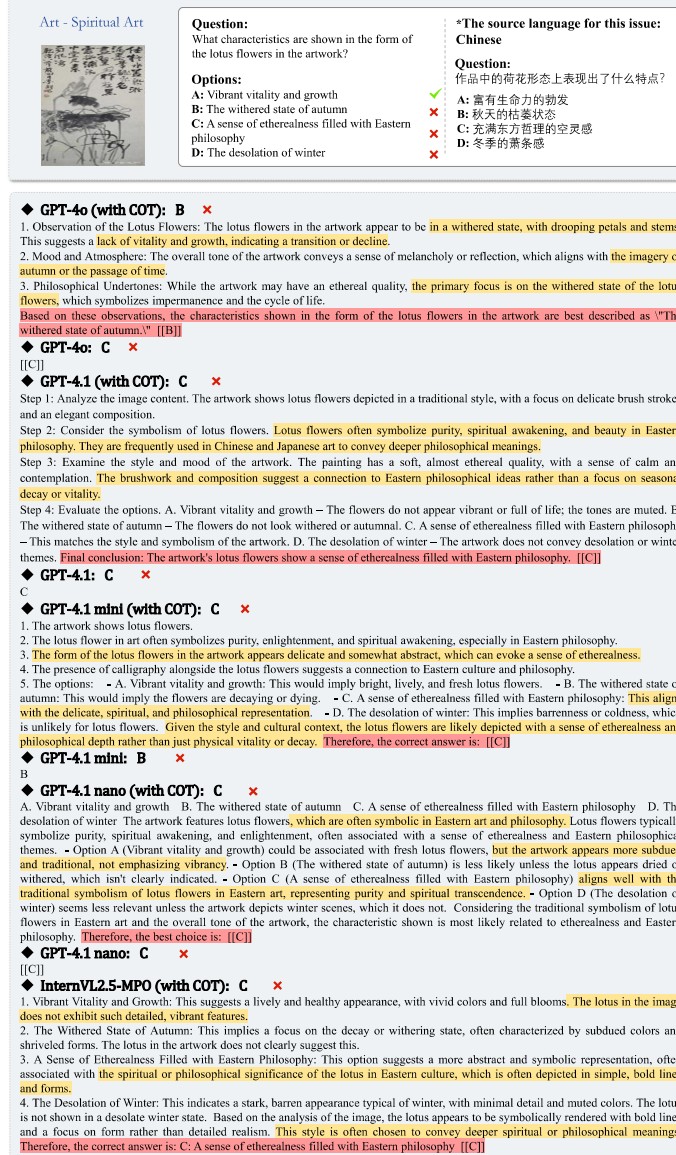

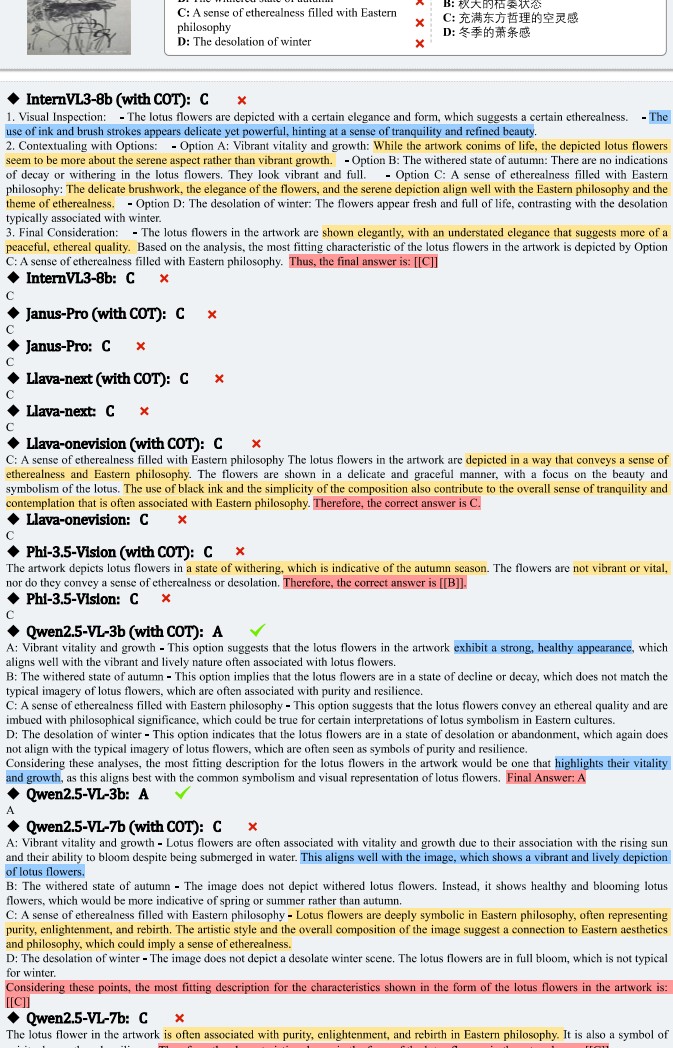

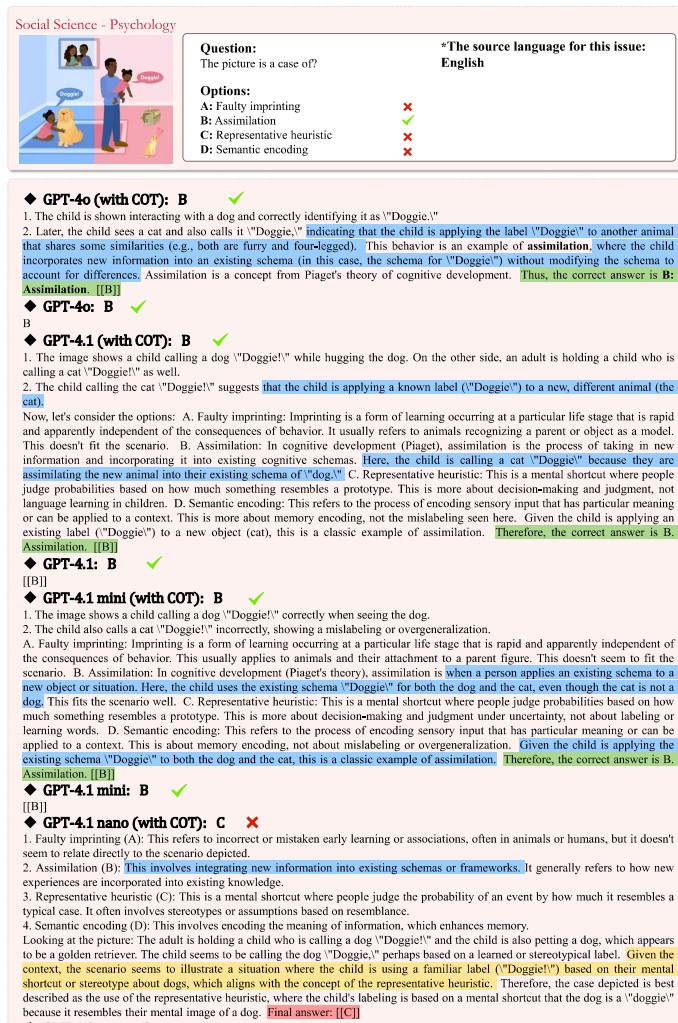

**Social Science - Psychology**

**Question:**
The picture is a case of?

**\*The source language for this issue:**
**English**

**Options:**
**A:** Faulty imprinting ✗
**B:** Assimilation ✓
**C:** Representative heuristic ✗
**D:** Semantic encoding ✗

---

◆ **GPT-4o (with COT): B** ✓
1. The child is shown interacting with a dog and correctly identifying it as \"Doggie.\"
2. Later, the child sees a cat and also calls it \"Doggie,\" indicating that the child is applying the label \"Doggie\" to another animal that shares some similarities (e.g., both are furry and four-legged). This behavior is an example of **assimilation**, where the child incorporates new information into an existing schema (in this case, the schema for \"Doggie\") without modifying the schema to account for differences. Assimilation is a concept from Piaget's theory of cognitive development. Thus, the correct answer is **B: Assimilation**. [[B]]

◆ **GPT-4o: B** ✓
B

◆ **GPT-4.1 (with COT): B** ✓
1. The image shows a child calling a dog \"Doggie!\" while hugging the dog. On the other side, an adult is holding a child who is calling a cat \"Doggie!\" as well.
2. The child calling the cat \"Doggie!\" suggests that the child is applying a known label (\"Doggie\") to a new, different animal (the cat).
Now, let's consider the options: A. Faulty imprinting: Imprinting is a form of learning occurring at a particular life stage that is rapid and apparently independent of the consequences of behavior. It usually refers to animals recognizing a parent or object as a model. This doesn't fit the scenario. B. Assimilation: In cognitive development (Piaget), assimilation is the process of taking in new information and incorporating it into existing cognitive schemas. Here, the child is calling a cat \"Doggie\" because they are assimilating the new animal into their existing schema of \"dog.\" C. Representative heuristic: This is a mental shortcut where people judge probabilities based on how much something resembles a prototype. This is more about decision-making and judgment, not language learning in children. D. Semantic encoding: This refers to the process of encoding sensory input that has particular meaning or can be applied to a context. This is more about memory encoding, not the mislabeling seen here. Given the child is applying an existing label (\"Doggie\") to a new object (cat), this is a classic example of assimilation. Therefore, the correct answer is B. Assimilation. [[B]]

◆ **GPT-4.1: B** ✓
[[B]]

◆ **GPT-4.1 mini (with COT): B** ✓
1. The image shows a child calling a dog \"Doggie!\" correctly when seeing the dog.
2. The child also calls a cat \"Doggie!\" incorrectly, showing a mislabeling or overgeneralization.
A. Faulty imprinting: Imprinting is a form of learning occurring at a particular life stage that is rapid and apparently independent of the consequences of behavior. This usually applies to animals and their attachment to a parent figure. This doesn't seem to fit the scenario. B. Assimilation: In cognitive development (Piaget's theory), assimilation is the process when a person applies an existing schema to a new object or situation. Here, the child uses the existing schema \"Doggie\" for both the dog and the cat, even though the cat is not a dog. This fits the scenario well. C. Representative heuristic: This is a mental shortcut where people judge probabilities based on how much something resembles a prototype. This is more about decision-making and judgment under uncertainty, not about labeling or learning words. D. Semantic encoding: This refers to the process of encoding sensory input that has particular meaning or can be applied to a context. This is about memory encoding, not about mislabeling or overgeneralization. Given the child is applying the existing schema \"Doggie\" to both the dog and the cat, this is a classic example of assimilation. Therefore, the correct answer is B. Assimilation. [[B]]

◆ **GPT-4.1 mini: B** ✓
[[B]]

◆ **GPT-4.1 nano (with COT): C** ✗
1. Faulty imprinting (A): This refers to incorrect or mistaken early learning or associations, often in animals or humans, but it doesn't seem to relate directly to the scenario depicted.
2. Assimilation (B): This involves integrating new information into existing schemas or frameworks. It generally refers to how new experiences are incorporated into existing knowledge.
3. Representative heuristic (C): This is a mental shortcut where people judge the probability of an event by how much it resembles a typical case. It often involves stereotypes or assumptions based on resemblance.
4. Semantic encoding (D): This involves encoding the meaning of information, which enhances memory.
Looking at the picture: The adult is holding a child who is calling a dog \"Doggie!\" and the child is also petting a dog, which appears to be a golden retriever. The child seems to be calling the dog \"Doggie,\" perhaps based on a learned or stereotypical label. Given the context, the scenario seems to illustrate a situation where the child is using a familiar label (\"Doggie!\") based on their mental shortcut or stereotype about dogs, which aligns with the concept of the representative heuristic. Therefore, the case depicted is best described as the use of the representative heuristic, where the child's labeling is based on a mental shortcut that the dog is a \"doggie\" because it resembles their mental image of a dog. Final answer: [[C]]

◆ **GPT-4.1 nano: C** ✗
[[C]]

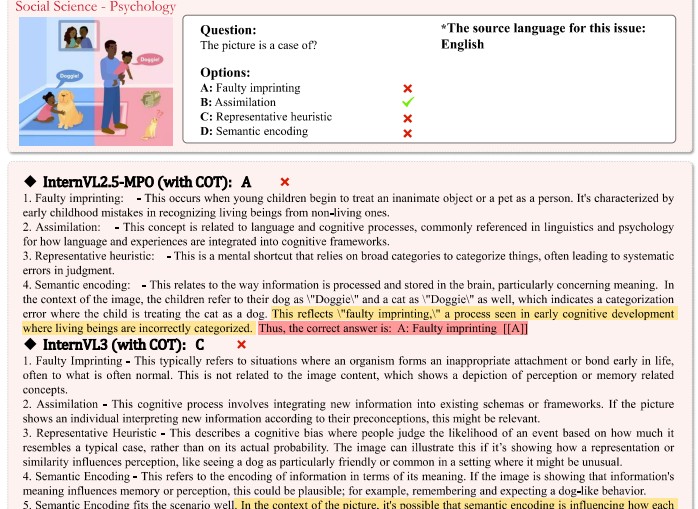

**Social Science - Psychology**

**Question:**
The picture is a case of?

***The source language for this issue:**
**English**

**Options:**
**A:** Faulty imprinting ✗
**B:** Assimilation ✓
**C:** Representative heuristic ✗
**D:** Semantic encoding ✗

◆ **InternVL2.5-MPO (with COT): A** ✗

1. Faulty imprinting: - This occurs when young children begin to treat an inanimate object or a pet as a person. It's characterized by early childhood mistakes in recognizing living beings from non-living ones.
2. Assimilation: - This concept is related to language and cognitive processes, commonly referenced in linguistics and psychology for how language and experiences are integrated into cognitive frameworks.
3. Representative heuristic: - This is a mental shortcut that relies on broad categories to categorize things, often leading to systematic errors in judgment.
4. Semantic encoding: - This relates to the way information is processed and stored in the brain, particularly concerning meaning. In the context of the image, the children refer to their dog as \"Doggie\" and a cat as \"Doggie\" as well, which indicates a categorization error where the child is treating the cat as a dog. This reflects \"faulty imprinting,\" a process seen in early cognitive development where living beings are incorrectly categorized. Thus, the correct answer is: A: Faulty imprinting [[A]]

◆ **InternVL3 (with COT): C** ✗

1. Faulty Imprinting - This typically refers to situations where an organism forms an inappropriate attachment or bond early in life, often to what is often normal. This is not related to the image content, which shows a depiction of perception or memory related concepts.
2. Assimilation - This cognitive process involves integrating new information into existing schemas or frameworks. If the picture shows an individual interpreting new information according to their preconceptions, this might be relevant.
3. Representative Heuristic - This describes a cognitive bias where people judge the likelihood of an event based on how much it resembles a typical case, rather than on its actual probability. The image can illustrate this if it's showing a representation or similarity influences perception, like seeing a dog as particularly friendly or common in a setting where it might be unusual.
4. Semantic Encoding - This refers to the encoding of information in terms of its meaning. If the image is showing that information's meaning influences memory or perception, this could be plausible; for example, remembering and expecting a dog-like behavior.
5. Semantic Encoding fits the scenario well. In the context of the picture, it's possible that semantic encoding is influencing how each character interacts with the dog, which represents familiar concepts like pets and family, as opposed to the cat, seen as unusual or out of place. After analyzing the options, it is reasonable to conclude: [[C]]

◆ **InternVL3: B** ✓

The picture illustrates different reactions to a dog and a cat from family members, which could point to differences in familiarity or comfort level. This is indicative of assimilation, where people create categories (like pets they like) and then new information is fit into these categories. [[B]]

◆ **Janus-Pro (with COT): B** ✓

B

◆ **Janus-Pro: B** ✓

B

◆ **Llava-next (with COT): A** ✗

A

◆ **Llava-next: A** ✗

A

◆ **Llava-onevision (with COT): A** ✗

A

◆ **Llava-onevision: A** ✗

A

Social Science - Psychology

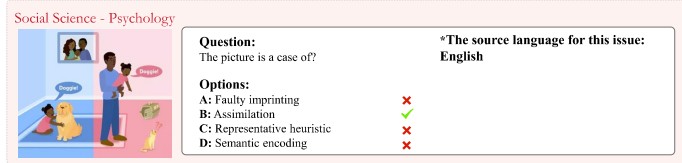

**Question:**
The picture is a case of?

**Options:**
**A:** Faulty imprinting     ✗
**B:** Assimilation     ✓
**C:** Representative heuristic     ✗
**D:** Semantic encoding     ✗

**\*The source language for this issue: English**

◆ **Phi-3.5-Vision (with COT): A** ✗

Step 1: Identify the main elements in the picture. - A man holding a child - A woman holding a dog - A dog in the foreground - A cat in the background

Step 2: Analyze the relationship between the elements. - The man and woman are holding a child and a dog, respectively. - The dog in the foreground is the same breed as the dog in the woman's arms. - The cat in the background is not related to the other elements.

Step 3: Determine which concept best explains the relationship between the elements. - Faulty imprinting: This refers to a situation where an individual forms an incorrect association between a stimulus and a response. - Assimilation: This refers to the process of incorporating new information into existing mental categories. - Representative heuristic: This refers to a mental shortcut that relies on stereotypes or generalizations to make judgments. - Semantic encoding: This refers to the process of encoding information into a meaningful format.

Step 4: Choose the correct answer based on the analysis. - The picture is a case of: [[A]]

◆ **Phi-3.5-Vision: A** ✗

A

◆ **Qwen2.5-VL-3b (with COT): C** ✗

A: Faulty imprinting - This refers to the early learning of a species' social behaviors, typically during a critical period of development. While the image shows a child interacting with a dog, it doesn't specifically indicate that this is happening during a critical period.

B: Assimilation - This term generally refers to the process of integrating new information into existing knowledge structures. It doesn't directly relate to the interaction between the child and the dog in the image.

C: Representative heuristic - This involves using a familiar example to understand something new. In the image, the child is calling the dog \"Doggie,\" which suggests they are using a familiar word (dog) to describe the dog, indicating a representative heuristic.

D: Semantic encoding - This refers to the process of converting sensory information into meaningful representations. While the child is likely processing the visual and auditory information about the dog, the specific action of calling it \"Doggie\" aligns more closely with the representative heuristic.

Given the analysis, the most appropriate answer is: C

◆ **Qwen2.5-VL-3b: A** ✗

A

◆ **Qwen2.5-VL-7b (with COT): C** ✗

1. **Faulty Imprinting**: This refers to a psychological phenomenon where an organism forms an attachment to the first moving object it sees after hatching. In the image, there is no indication that the child is forming an attachment based on the first moving object it sees.

2. **Assimilation**: This is a cognitive process where new information is incorporated into existing knowledge structures. The image does not show any new information being assimilated into existing knowledge structures.

3. **Representative Heuristic**: This is a mental shortcut that involves making judgments based on how well the current situation matches a prototype or example in memory. The image shows a child interacting with a dog and a cat, which could be seen as a representative heuristic if the child is using the dog as a prototype for understanding animals.

4. **Semantic Encoding**: This is the process of encoding meaning into memory. The image does not show any specific semantic encoding; it is more about the child's interaction with the animals.

Given the analysis, the most fitting option is the one that describes the child's interaction with the animals based on a prototype or example in memory. The correct answer is: C [[C]]

◆ **Qwen2.5-VL-7b: B** ✓

The picture shows a child being taught to call a dog \"Doggie!\" by an adult. This scenario illustrates the concept of **assimilation**, where the child learns to apply a new label (\"Doggie!\") to a familiar object (the dog) based on the adult's example. Therefore, the correct answer is: [[B]]

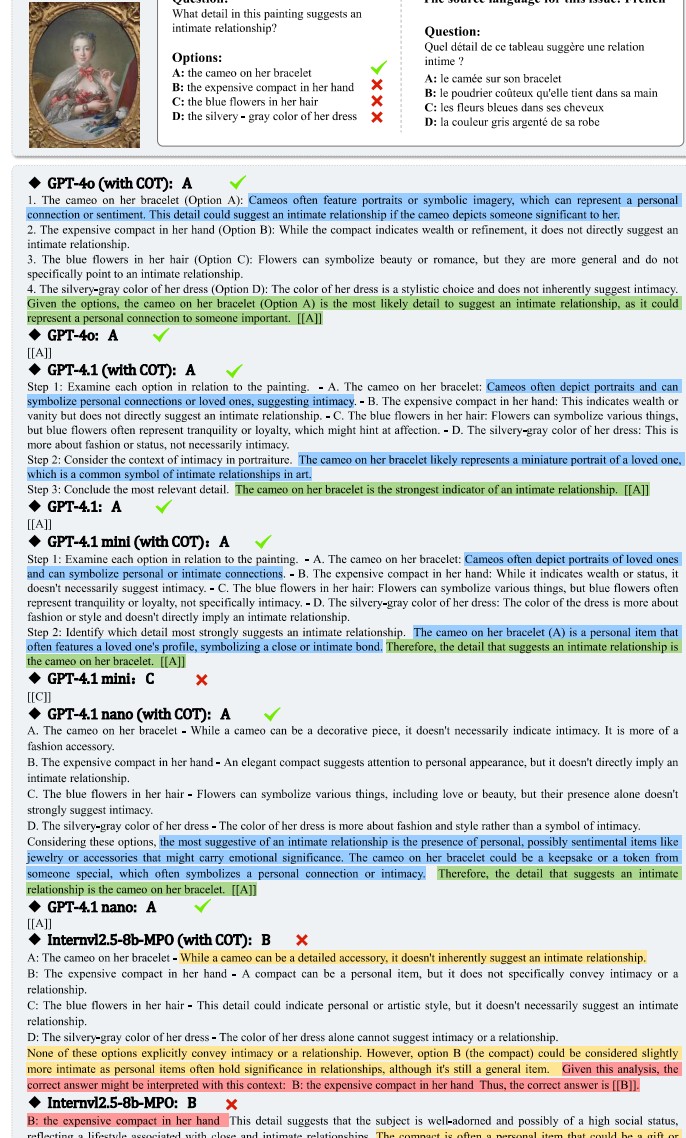

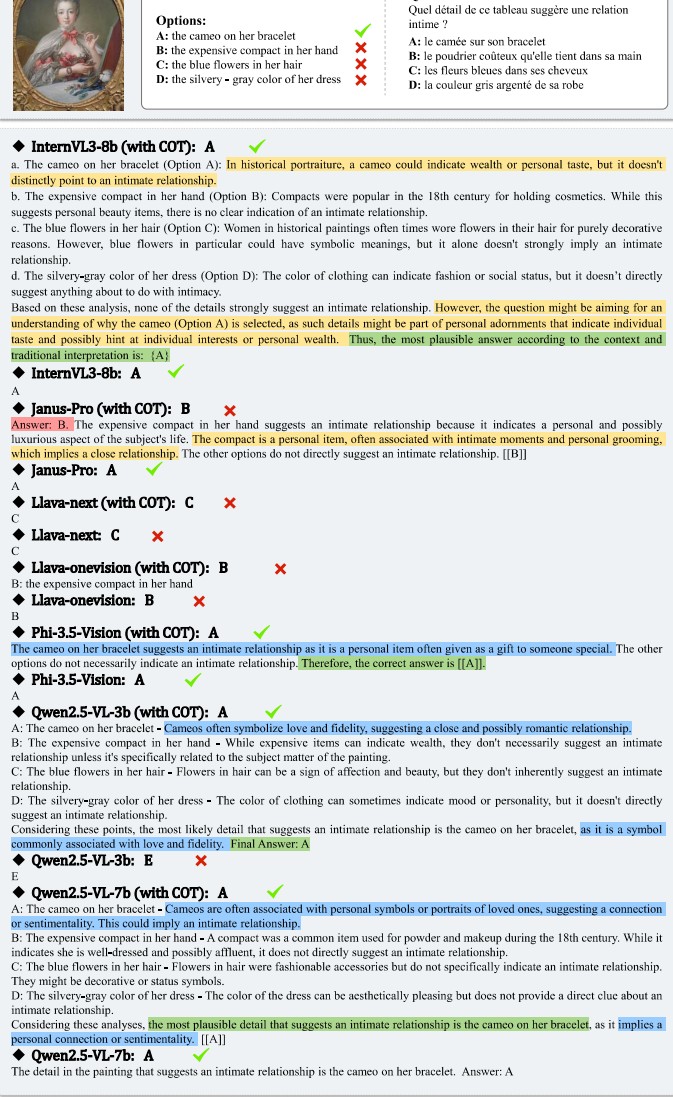

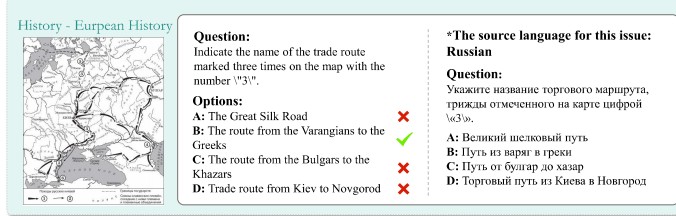

History - Eurpean History

**Question:**
Indicate the name of the trade route marked three times on the map with the number \"3\".

**Options:**
A: The Great Silk Road ✗
B: The route from the Varangians to the Greeks ✓
C: The route from the Bulgars to the Khazars ✗
D: Trade route from Kiev to Novgorod ✗

*The source language for this issue: **Russian**

**Question:**
Укажите название торгового маршрута, трижды отмеченного на карте цифрой \«3».

A: Великий шелковый путь
B: Путь из варяг в греки
C: Путь от булгар до хазар
D: Торговый путь из Киева в Новгород

---

◆ **GPT-4o (with COT): B** ✓

1. Observation of the map: The trade route marked with the number \"3\" connects regions such as Kiev, Novgorod, and Constantinople. This route follows rivers and passes through key trading cities.
2. Understanding the options: - A: The Great Silk Road: This route primarily connected China to Europe and passed through Central Asia. It does not align with the route marked \"3\" on the map. - B: The route from the Varangians to the Greeks: This was a major trade route connecting the Baltic Sea (Varangians) to the Byzantine Empire (Greeks) via rivers such as the Dnieper. The route marked \"3\" matches this description. - C: The route from the Bulgars to the Khazars: This route would primarily connect regions in the Volga and Caspian areas, which are not shown as the focus of the map. - D: Trade route from Kiev to Novgorod: While Kiev and Novgorod are connected, the route marked \"3\" extends further south to Constantinople, indicating a broader trade route.
3. Conclusion: The route marked \"3\" corresponds to the route from the Varangians to the Greeks. Final answer: [[B]]

◆ **GPT-4o: B** ✓

[[B]]

◆ **GPT-4.1 (with COT): B** ✓

1. The map shows several trade routes marked with numbers. The number \"3\" appears three times on the map.
2. The routes marked with \"3\" connect the following key locations: - From the Baltic Sea region (near Novgorod and Ladoga) down through Smolensk and Kiev. - From Kiev down to the Black Sea region (near Chersonesus).
3. The options are: A. The Great Silk Road - This was a trade route connecting China to Europe, mainly through Central Asia, not primarily through Eastern Europe and the Baltic region. B. The route from the Varangians to the Greeks - This was a famous medieval trade route connecting the Baltic Sea (Varangians) to the Byzantine Empire (Greeks) via rivers like the Dnieper, passing through Kiev and down to the Black Sea. C. The route from the Bulgars to the Khazars - This would be more localized in the Volga and Caspian Sea regions. D. Trade route from Kiev to Novgorod - This is a more localized route between two cities in Eastern Europe. 4. The route marked \"3\" on the map corresponds to the famous \"route from the Varangians to the Greeks,\" which connected the Baltic Sea to the Black Sea through Kiev, facilitating trade between Northern Europe and Byzantium.
Therefore, the correct answer is B. [[B]]

◆ **GPT-4.1: B** ✓

The trade route marked three times with the number \"3\" on the map corresponds to the route from the Varangians to the Greeks, which was a major medieval trade route connecting the Baltic Sea with the Byzantine Empire via rivers such as the Dnieper. [[B]]

◆ **GPT-4.1 mini (with COT): B** ✓

1. The map shows several trade routes marked with numbers. The number \"3\" appears three times on the map.
2. The routes marked with \"3\" connect the following key locations: - From the Baltic Sea region (Novgorod area) down through Kiev. - From Kiev towards the Black Sea.
3. The options are: A. The Great Silk Road - This was a vast network of trade routes connecting East Asia to Europe, mainly through Central Asia, not specifically the routes shown here. B. The route from the Varangians to the Greeks - This was a medieval trade route connecting the Baltic Sea (Varangians) to the Byzantine Empire (Greeks) via rivers like the Dnieper, passing through Kiev and down to the Black Sea. C. The route from the Bulgars to the Khazars - This would be more localized in the Volga and Caspian regions. D. Trade route from Kiev to Novgorod - This is a more localized route between two cities in Rus', but the map shows a broader route including the Black Sea and Baltic Sea.
4. The route marked \"3\" on the map corresponds to the famous \"route from the Varangians to the Greeks,\" which connected the Baltic Sea (Varangians) through rivers and Kiev to the Black Sea and Byzantium (Greeks).
Therefore, the correct answer is: [[B]]

◆ **GPT-4.1 mini：B** ✓

[[B]]

◆ **GPT-4.1 nano (with COT): D** ✗

1. The map shows several trade routes marked with numbers.
2. The question asks for the name of the trade route marked three times with the number \"3\".
3. Looking at the map, the route marked with \"3\" appears in three locations: - From Kiev towards the northeast, passing through Chernihiv. - From Kiev towards the south, passing through Pereiaslav. - From Kiev towards the southwest, passing through Chersonesus (Korsun). 4. These routes seem to connect Kiev with different regions, indicating a significant trade route originating or passing through Kiev. 5. Among the options: - A. The Great Silk Road: This was a major trade route connecting China to the Mediterranean, not specifically centered around Kiev. - B. The route from the Varangians to the Greeks: This was a trade route connecting the Varangians (Vikings) to the Byzantine Empire, passing through Eastern Europe and the Black Sea. - C. The route from the Bulgars to the Khazars: This is less prominent in the context of Kiev's trade routes. - D. Trade route from Kiev to Novgorod: This is a well-known trade route connecting Kiev to Novgorod, passing through the regions shown on the map.
Given the map's focus on Kiev and the multiple routes marked \"3\" connecting Kiev to various regions, the most fitting option is: D. Trade route from Kiev to Novgorod. [[D]]

◆ **GPT-4.1 nano：A** ✗

[[A]]

◆ **Internvl2.5-8b-MPO (with COT): C** ✗

1. Option A: The Great Silk Road - The Great Silk Road is a historic trade route that connected China with the Western world, often traversing central Asia and the Middle East. - The routes marked in this manner would typically travel through regions such as modern-day China, Central Asia, and parts of Europe.
2. Option B: The route from the Varangians to the Greeks - This route involves a trade or cultural connection between Scandinavian regions (Varangians) and the Eastern Mediterranean (Greeks). - Such a route would likely include sea journeys, notably through the Byzantine Empire or along coastal regions.
3. Option C: The route from the Bulgars to the Khazars - This specifically involves connections between the Bulgarian states and the Khazar Empire. - The Bulgars were a group in regions that are part of modern-day Bulgaria, Serbia, and Romania, while the Khazars were centered in modern Ukraine and southern Russia. - The route shown on the map aligns with historical reconstructions supporting trade interactions between these regions.
4. Option D: Trade route from Kiev to Novgorod - This involves direct internal routes within what is now Ukraine, connecting major cities like Kiev and Novgorod. - While this describes internal regional connections in Ukraine, the routes on the map suggest broader continental connections not limited to this geographical area. Chain of Thought Analysis: - The map shows a route that spans widely across Europe and reaches Asia. - The routes are marked over areas historically associated with the Khazars (part of Russia and Eastern Europe) and extend towards regions connected to the Bulgarian cultural sphere. - Considering the positions and directions, these specific routes are indicative of connections fitting historical descriptions of trade routes between the Bulgars and the Khazars rather than internal Ukrainian trade routes or purely northern maritime paths. - The patterns and geographical spans described do most accurately align with the connection from the Bulgars to the Khazars. Therefore, the correct answer is: [[C]]

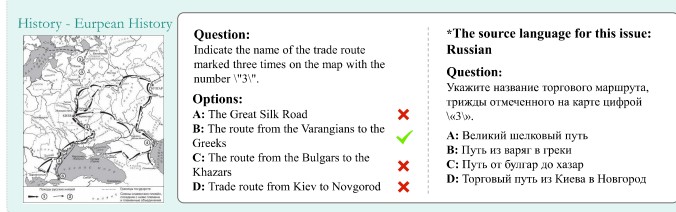

History - Eurpean History

**Question:**
Indicate the name of the trade route marked three times on the map with the number \"3\".

**Options:**
A: The Great Silk Road  ✗
B: The route from the Varangians to the Greeks  ✓
C: The route from the Bulgars to the Khazars  ✗
D: Trade route from Kiev to Novgorod  ✗

**\*The source language for this issue: Russian**

**Question:**
Укажите название торгового маршрута, трижды отмеченного на карте цифрой \«3\».

A: Великий шелковый путь
B: Путь из варяг в греки
C: Путь от булгар до хазар
D: Торговый путь из Киева в Новгород

◆ **GPT-4o (with COT): B** ✓

1. Observation of the map: The trade route marked with the number \"3\" connects regions such as Kiev, Novgorod, and Constantinople. This route follows rivers and passes through key trading cities.
2. Understanding the options:  - A: The Great Silk Road: This route primarily connected China to Europe and passed through Central Asia. It does not align with the route marked \"3\" on the map.  - B: The route from the Varangians to the Greeks: This was a major trade route connecting the Baltic Sea (Varangians) to the Byzantine Empire (Greeks) via rivers such as the Dnieper. The route marked \"3\" matches this description.  - C: The route from the Bulgars to the Khazars: This route would primarily connect regions in the Volga and Caspian areas, which are not shown as the focus of the map.  - D: Trade route from Kiev to Novgorod: While Kiev and Novgorod are connected, the route marked \"3\" extends further south to Constantinople, indicating a broader trade route.
3. Conclusion: The route marked \"3\" corresponds to the route from the Varangians to the Greeks.  Final answer: [[B]]

◆ **GPT-4o: B** ✓

[[B]]

◆ **GPT-4.1 (with COT): B** ✓

1. The map shows several trade routes marked with numbers. The number \"3\" appears three times on the map.
2. The routes marked with \"3\" connect the following key locations:  - From the Baltic Sea region (near Novgorod and Ladoga) down through Smolensk and Kiev.  - From Kiev down to the Black Sea region (near Chersonesus).
3. The options are:  A. The Great Silk Road - This was a trade route connecting China to Europe, mainly through Central Asia, not primarily through Eastern Europe and the Baltic region.  B. The route from the Varangians to the Greeks - This was a famous medieval trade route connecting the Baltic Sea (Varangians) to the Byzantine Empire (Greeks) via rivers like the Dnieper, passing through Kiev and down to the Black Sea.  C. The route from the Bulgars to the Khazars - This would be more localized in the Volga and Caspian Sea regions.  D. Trade route from Kiev to Novgorod - This is a more localized route between two cities in Eastern Europe.  4. The route marked \"3\" on the map corresponds to the famous \"route from the Varangians to the Greeks,\" which connected the Baltic Sea to the Black Sea through Kiev, facilitating trade between Northern Europe and Byzantium.  Therefore, the correct answer is B.  [[B]]

◆ **GPT-4.1: B** ✓

The trade route marked three times with the number \"3\" on the map corresponds to the route from the Varangians to the Greeks, which was a major medieval trade route connecting the Baltic Sea with the Byzantine Empire via rivers such as the Dnieper.  [[B]]

◆ **GPT-4.1 mini (with COT): B** ✓

1. The map shows several trade routes marked with numbers. The number \"3\" appears three times on the map.
2. The routes marked with \"3\" connect the following key locations:  - From the Baltic Sea region (Novgorod area) down through Kiev.  - From Kiev towards the Black Sea.
3. The options are:  A. The Great Silk Road - This was a vast network of trade routes connecting East Asia to Europe, mainly through Central Asia, not specifically the routes shown here.  B. The route from the Varangians to the Greeks - This was a medieval trade route connecting the Baltic Sea (Varangians) to the Byzantine Empire (Greeks) via rivers like the Dnieper, passing through Kiev and down to the Black Sea.  C. The route from the Bulgars to the Khazars - This would be more localized in the Volga and Caspian Sea regions.  D. Trade route from Kiev to Novgorod - This is a more localized route between two cities in Rus', but the map shows a broader route including the Black Sea and Baltic Sea.
4. The route marked \"3\" on the map corresponds to the famous \"route from the Varangians to the Greeks,\" which connected the Baltic Sea (Varangians) through rivers and Kiev to the Black Sea and Byzantium (Greeks).  Therefore, the correct answer is: [[B]]

◆ **GPT-4.1 mini：B** ✓

[[B]]

◆ **GPT-4.1 nano (with COT): D** ✗

1. The map shows several trade routes marked with numbers.
2. The question asks for the name of the trade route marked three times with the number \"3\".
3. Looking at the map, the route marked with \"3\" appears in three locations:  - From Kiev towards the northeast, passing through Chernihiv.  - From Kiev towards the south, passing through Pereiaslav.  - From Kiev towards the southwest, passing through Chersonesus (Korsun). 4. These routes seem to connect Kiev with different regions, indicating a significant trade route originating or passing through Kiev. 5. Among the options:  - A. The Great Silk Road: This was a major trade route connecting China to the Mediterranean, not specifically centered around Kiev.  - B. The route from the Varangians to the Greeks: This was a trade route connecting the Varangians (Vikings) to the Byzantine Empire, passing through Eastern Europe and the Black Sea.  - C. The route from the Bulgars to the Khazars: This is less prominent in the context of Kiev's trade routes.  - D. Trade route from Kiev to Novgorod: This is a well-known trade route connecting Kiev to Novgorod, passing through the regions shown on the map.  Given the map's focus on Kiev and the multiple routes marked \"3\" connecting Kiev to various regions, the most fitting option is:  D. Trade route from Kiev to Novgorod. [[D]]

◆ **GPT-4.1 nano：A** ✗

[[A]]

◆ **Internvl2.5-8b-MPO (with COT): C** ✗

1. Option A: The Great Silk Road  - The Great Silk Road is a historic trade route that connected China with the Western world, often traversing central Asia and the Middle East.  - The routes marked in this manner would typically travel through regions such as modern-day China, Central Asia, and parts of Europe.
2. Option B: The route from the Varangians to the Greeks  - This route involves a trade or cultural connection between Scandinavian regions (Varangians) and the Eastern Mediterranean (Greeks).  - Such a route would likely include sea journeys, notably through the Byzantine Empire or along coastal regions.
3. Option C: The route from the Bulgars to the Khazars  - This specifically involves connections between the Bulgarian states and the Khazar Empire.  - The Bulgars were a group in regions that are part of modern-day Bulgaria, Serbia, and Romania, while the Khazars were centered in modern Ukraine and southern Russia.  - The route shown on the map aligns with historical reconstructions supporting trade interactions between these regions.
4. Option D: Trade route from Kiev to Novgorod  - This involves direct internal routes within what is now Ukraine, connecting major cities like Kiev and Novgorod.  - While this describes internal regional connections in Ukraine, the routes on the map suggest broader continental connections not limited to this geographical area.  Chain of Thought Analysis:  - The map shows a route that spans widely across Europe and reaches Asia. - The routes are marked over areas historically associated with the Khazars (part of Russia and Eastern Europe) and extend towards regions connected to the Bulgarian cultural sphere.  - Considering the positions and directions, these specific routes are indicative of connections fitting historical descriptions of trade routes between the Bulgars and the Khazars rather than internal Ukrainian trade routes or purely northern maritime paths.  - The patterns and geographical spans described do most accurately align with the connection from the Bulgars to the Khazars.  Therefore, the correct answer is: [[C]]

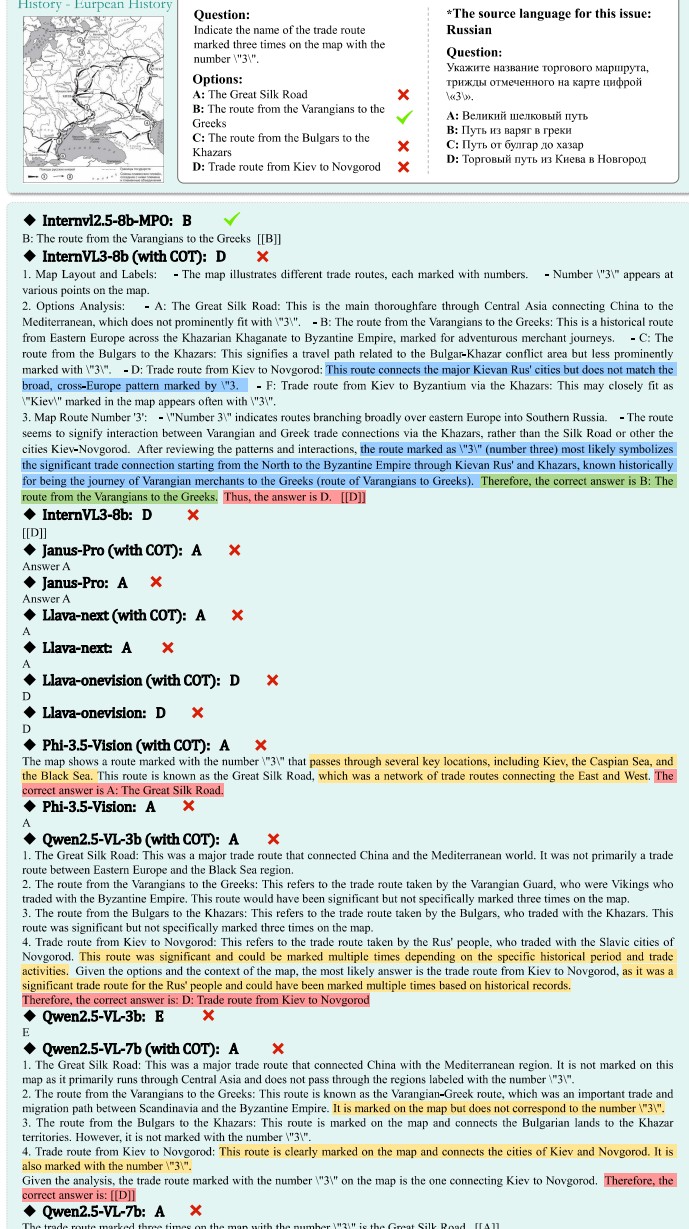

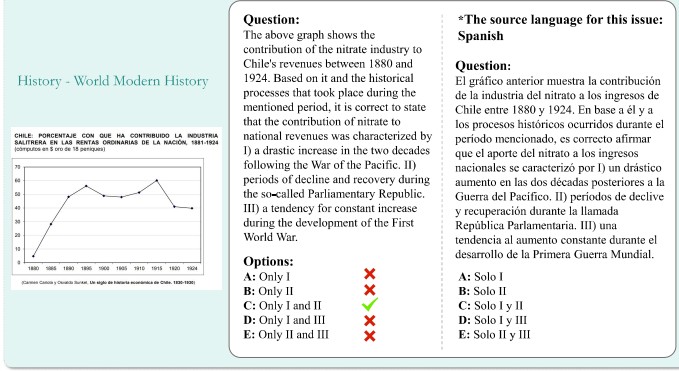

**Question:**
The above graph shows the contribution of the nitrate industry to Chile's revenues between 1880 and 1924. Based on it and the historical processes that took place during the mentioned period, it is correct to state that the contribution of nitrate to national revenues was characterized by I) a drastic increase in the two decades following the War of the Pacific. II) periods of decline and recovery during the so-called Parliamentary Republic. III) a tendency for constant increase during the development of the First World War.

**Options:**
A: Only I ✗
B: Only II ✗
C: Only I and II ✓
D: Only I and III ✗
E: Only II and III ✗

*The source language for this issue: Spanish

**Question:**
El gráfico anterior muestra la contribución de la industria del nitrato a los ingresos de Chile entre 1880 y 1924. En base a él y a los procesos históricos ocurridos durante el periodo mencionado, es correcto afirmar que el aporte del nitrato a los ingresos nacionales se caracterizó por I) un drástico aumento en las dos décadas posteriores a la Guerra del Pacífico. II) períodos de declive y recuperación durante la llamada República Parlamentaria. III) una tendencia al aumento constante durante el desarrollo de la Primera Guerra Mundial.

A: Solo I
B: Solo II
C: Solo I y II
D: Solo I y III
E: Solo II y III

---

◆ **GPT-4o (with COT): C** ✓

Statement I: \"A drastic increase in the two decades following the War of the Pacific.\" ▬ The War of the Pacific ended in 1884, and Chile gained control of nitrate-rich territories. The graph shows a sharp increase in nitrate contributions to national revenues from 1880 to 1895, going from approximately 10% to over 50%. This aligns with the post-war economic boom driven by nitrate exports. ▬ Statement I is correct.
Statement II: \"Periods of decline and recovery during the so-called Parliamentary Republic.\" ▬ The Parliamentary Republic in Chile lasted from 1891 to 1925. During this period, the graph shows fluctuations in nitrate contributions. After peaking in 1895, there is a decline around 1900, followed by a recovery leading up to 1915. This reflects the instability and economic challenges of the Parliamentary Republic era. ▬ Statement II is correct.
Statement III: \"A tendency for constant increase during the development of the First World War.\" ▬ The First World War lasted from 1914 to 1918. The graph shows an increase in nitrate contributions from 1910 to 1915, but after 1915, there is a sharp decline. This indicates that while there was an initial increase during the war, it was not constant throughout the period. ▬ Statement III is incorrect.
Final Evaluation: ▬ Statements I and II are correct, while Statement III is incorrect. Correct Answer: C: Solo I y II [[C]]

◆ **GPT-4o: C** ✓

[[C]]

◆ **GPT-4.1 (with COT): C** ✓

I) \"A drastic increase in the two decades following the War of the Pacific.\" ▬ The War of the Pacific ended in 1883. ▬ The graph shows a sharp increase in nitrate contribution from 1881 (around 5%) to 1895 (around 56%). ▬ This confirms a drastic increase in the two decades after the war. ▬ So, statement I is correct.
II) \"Periods of decline and recovery during the so-called Parliamentary Republic.\" ▬ The Parliamentary Republic in Chile lasted roughly from 1891 to 1925. ▬ From the graph, between 1895 and 1924, the nitrate contribution fluctuates: it declines from 56% (1895) to about 48% (1905), then recovers to 60% (1915), and then declines again to 40% (1924). ▬ This pattern shows periods of decline and recovery. ▬ So, statement II is correct.
III) \"A tendency for constant increase during the development of the First World War.\" ▬ The First World War lasted from 1914 to 1918. ▬ The graph shows an increase from about 51% in 1910 to 60% in 1915, which is during the war. ▬ However, after 1915, there is a sharp decline to 41% in 1920. ▬ So, during the war itself, there was an increase, but not a constant increase throughout the entire period. ▬ The statement says \"a tendency for constant increase during the development of the First World War,\" which is partially true but not fully sustained. ▬ Given the graph, the increase is only until 1915, then it declines. ▬ So, statement III is not fully correct.
Conclusion: ▬ I is correct. ▬ II is correct. ▬ III is not fully correct. Therefore, the correct answer is C: Solo I y II [[C]]

◆ **GPT-4.1: C** ✓

I) A drastic increase in the two decades following the War of the Pacific (1879-1883): ▬ The graph shows a sharp increase from about 5% in 1880 to nearly 56% by 1895. This corresponds to the two decades after the War of the Pacific, during which Chile gained control of nitrate-rich territories. So, statement I is correct.
II) Periods of decline and recovery during the so-called Parliamentary Republic (1891-1925): ▬ The graph shows fluctuations between 1895 and 1924, with declines around 1900-1905, recoveries around 1910-1915, and a decline again after 1915. This matches the Parliamentary Republic period, characterized by political instability and economic fluctuations. So, statement II is correct.
III) A tendency for constant increase during the development of the First World War (1914-1918): ▬ The graph shows an increase from about 51% in 1910 to 60% in 1915, but then a sharp decline after 1915. So, the increase is not constant throughout the WWI period; it peaks and then falls. Therefore, statement III is not entirely correct.
Therefore, the correct statements are I and II. [[C]]

◆ **GPT-4.1 mini (with COT): C** ✓

I) \"A drastic increase in the two decades following the War of the Pacific.\" ▬ The War of the Pacific ended in 1883. ▬ The graph shows a sharp increase in nitrate contribution from 1881 (around 5%) to 1895 (around 56%). ▬ This confirms a drastic increase in the two decades after the war. ▬ So, statement I is correct.
II) \"Periods of decline and recovery during the so-called Parliamentary Republic.\" ▬ The Parliamentary Republic in Chile lasted roughly from 1891 to 1925. ▬ The graph shows fluctuations during this period: a decline from 1895 to 1905 (from 56% to 48%), a recovery from 1905 to 1915 (48% to 60%), and then a decline again after 1915. ▬ This matches the description of periods of decline and recovery. ▬ So, statement II is correct.
III) \"A tendency for constant increase during the development of the First World War.\" ▬ World War I lasted from 1914 to 1918. ▬ The graph shows an increase from 1910 (51%) to 1915 (60%), which aligns with the start and middle of WWI. ▬ However, after 1915, there is a sharp decline to 41% in 1920. ▬ So, the increase is not constant throughout the entire WWI period; it peaks and then declines. ▬ Therefore, statement III is not entirely correct. Conclusion: ▬ I is correct. ▬ II is correct. ▬ III is not correct. Therefore, the correct answer is C: Solo I y II [[C]]

◆ **GPT-4.1 mini: C** ✓

I) A drastic increase in the two decades following the War of the Pacific (1879-1883): ▬ The graph shows a sharp increase from about 5% in 1880 to nearly 56% by 1895. This corresponds to the two decades after the War of the Pacific, during which Chile gained control of nitrate-rich territories. So, statement I is correct.
II) Periods of decline and recovery during the so-called Parliamentary Republic (1891-1925): ▬ The graph shows fluctuations between 1895 and 1924, with declines around 1900-1905 and recoveries around 1910-1915, followed by a decline after 1915. This matches the Parliamentary Republic period, so statement II is correct.
III) A tendency for constant increase during the development of the First World War (1914-1918): ▬ The graph shows an increase from about 51% in 1910 to 60% in 1915, but then a sharp decline after 1915. So, the increase is not constant throughout the WWI period; it peaks and then falls. Therefore, statement III is not entirely correct.
Therefore, the correct statements are I and II. [[C]]

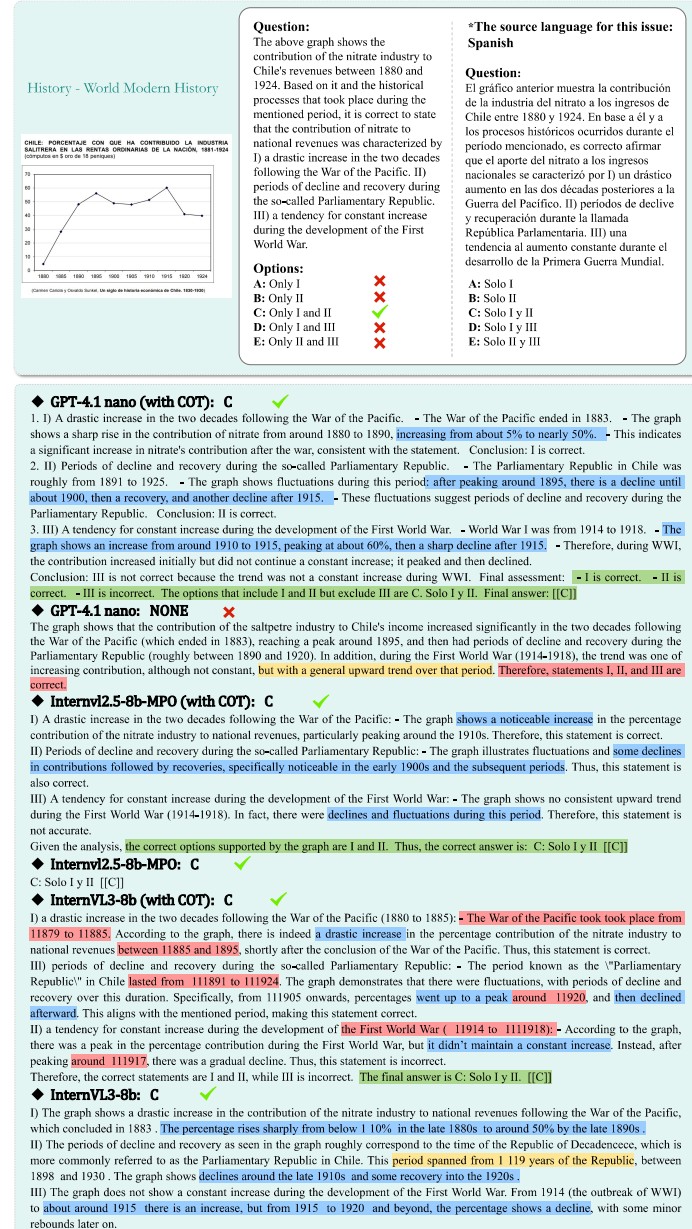

**Question:**
The above graph shows the contribution of the nitrate industry to Chile's revenues between 1880 and 1924. Based on it and the historical processes that took place during the mentioned period, it is correct to state that the contribution of nitrate to national revenues was characterized by I) a drastic increase in the two decades following the War of the Pacific. II) periods of decline and recovery during the so-called Parliamentary Republic. III) a tendency for constant increase during the development of the First World War.

**Options:**
A: Only I ✗
B: Only II ✗
C: Only I and II ✓
D: Only I and III ✗
E: Only II and III ✗

**\*The source language for this issue: Spanish**

**Question:**
El gráfico anterior muestra la contribución de la industria del nitrato a los ingresos de Chile entre 1880 y 1924. En base a él y a los procesos históricos ocurridos durante el período mencionado, es correcto afirmar que el aporte del nitrato a los ingresos nacionales se caracterizó por I) un drástico aumento en las dos décadas posteriores a la Guerra del Pacífico. II) períodos de declive y recuperación durante la llamada República Parlamentaria. III) una tendencia al aumento constante durante el desarrollo de la Primera Guerra Mundial.

A: Solo I
B: Solo II
C: Solo I y II
D: Solo I y III
E: Solo II y III

History - World Modern History

CHILE: PORCENTAJE CON QUE HA CONTRIBUIDO LA INDUSTRIA SALITRERA EN LAS RENTAS ORDINARIAS DE LA NACIÓN, 1881-1924
(cómputos en $ oro de 18 peniques)

(Carmen Cariola y Osvaldo Sunkel, Un siglo de historia económica de Chile. 1830-1930)

---

◆ **GPT-4.1 nano (with COT):  C** ✓
1. I) A drastic increase in the two decades following the War of the Pacific.  ‑ The War of the Pacific ended in 1883.  ‑ The graph shows a sharp rise in the contribution of nitrate from around 1880 to 1890, increasing from about 5% to nearly 50%.  ‑ This indicates a significant increase in nitrate's contribution after the war, consistent with the statement.  Conclusion: I is correct.
2. II) Periods of decline and recovery during the so-called Parliamentary Republic.  ‑ The Parliamentary Republic in Chile was roughly from 1891 to 1925.  ‑ The graph shows fluctuations during this period: after peaking around 1895, there is a decline until about 1900, then a recovery, and another decline after 1915.  ‑ These fluctuations suggest periods of decline and recovery during the Parliamentary Republic.  Conclusion: II is correct.
3. III) A tendency for constant increase during the development of the First World War.  ‑ World War I was from 1914 to 1918.  ‑ The graph shows an increase from around 1910 to 1915, peaking at about 60%, then a sharp decline after 1915.  ‑ Therefore, during WWI, the contribution increased initially but did not continue a constant increase; it peaked and then declined.
Conclusion: III is not correct because the trend was not a constant increase during WWI.  Final assessment:  ‑ I is correct.  ‑ II is correct.  ‑ III is incorrect.  The options that include I and II but exclude III are C. Solo I y II.  Final answer: [[C]]

◆ **GPT-4.1 nano:  NONE** ✗
The graph shows that the contribution of the saltpetre industry to Chile's income increased significantly in the two decades following the War of the Pacific (which ended in 1883), reaching a peak around 1895, and then had periods of decline and recovery during the Parliamentary Republic (roughly between 1890 and 1920). In addition, during the First World War (1914-1918), the trend was one of increasing contribution, although not constant, but with a general upward trend over that period.  Therefore, statements I, II, and III are correct.

◆ **Internvl2.5-8b-MPO (with COT):  C** ✓
I) A drastic increase in the two decades following the War of the Pacific: - The graph shows a noticeable increase in the percentage contribution of the nitrate industry to national revenues, particularly peaking around the 1910s. Therefore, this statement is correct.
II) Periods of decline and recovery during the so-called Parliamentary Republic: - The graph illustrates fluctuations and some declines in contributions followed by recoveries, specifically noticeable in the early 1900s and the subsequent periods. Thus, this statement is also correct.
III) A tendency for constant increase during the development of the First World War: - The graph shows no consistent upward trend during the First World War (1914-1918). In fact, there were declines and fluctuations during this period. Therefore, this statement is not accurate.
Given the analysis, the correct options supported by the graph are I and II. Thus, the correct answer is:  C: Solo I y II  [[C]]

◆ **Internvl2.5-8b-MPO:  C** ✓
C: Solo I y II  [[C]]

◆ **InternVL3-8b (with COT):  C** ✓
I) a drastic increase in the two decades following the War of the Pacific (1880 to 1885): - The War of the Pacific took took place from 11879 to 11885. According to the graph, there is indeed a drastic increase in the percentage contribution of the nitrate industry to national revenues between 11885 and 1895, shortly after the conclusion of the War of the Pacific. Thus, this statement is correct.
III) periods of decline and recovery during the so-called Parliamentary Republic: - The period known as the \"Parliamentary Republic\" in Chile lasted from  111891 to 111924. The graph demonstrates that there were fluctuations, with periods of decline and recovery over this duration. Specifically, from 111905 onwards, percentages went up to a peak around  11920, and then declined afterward. This aligns with the mentioned period, making this statement correct.
II) a tendency for constant increase during the development of the First World War ( 11914 to  1111918): - According to the graph, there was a peak in the percentage contribution during the First World War, but it didn't maintain a constant increase. Instead, after peaking around  111917, there was a gradual decline. Thus, this statement is incorrect.
Therefore, the correct statements are I and II, while III is incorrect. The final answer is C: Solo I y II.  [[C]]

◆ **InternVL3-8b:  C** ✓
I) The graph shows a drastic increase in the contribution of the nitrate industry to national revenues following the War of the Pacific, which concluded in 1883. The percentage rises sharply from below 1 10%  in the late 1880s  to around 50% by the late 1890s .
II) The periods of decline and recovery as seen in the graph roughly correspond to the time of the Republic of Decadencece, which is more commonly referred to as the Parliamentary Republic in Chile. This period spanned from 1 119 years of the Republic, between 1898  and 1930 . The graph shows declines around the late 1910s  and some recovery into the 1920s .
III) The graph does not show a constant increase in the contribution of the nitrate industry to national revenues during the First World War. From 1914 (the outbreak of WWI) to about around 1915  there is an increase, but from 1915  to 1920  and beyond, the percentage shows a decline, with some minor rebounds later on.
Based on this analysis, the correct statement is: C: Solo I y II  Thus, the final answer is:  C

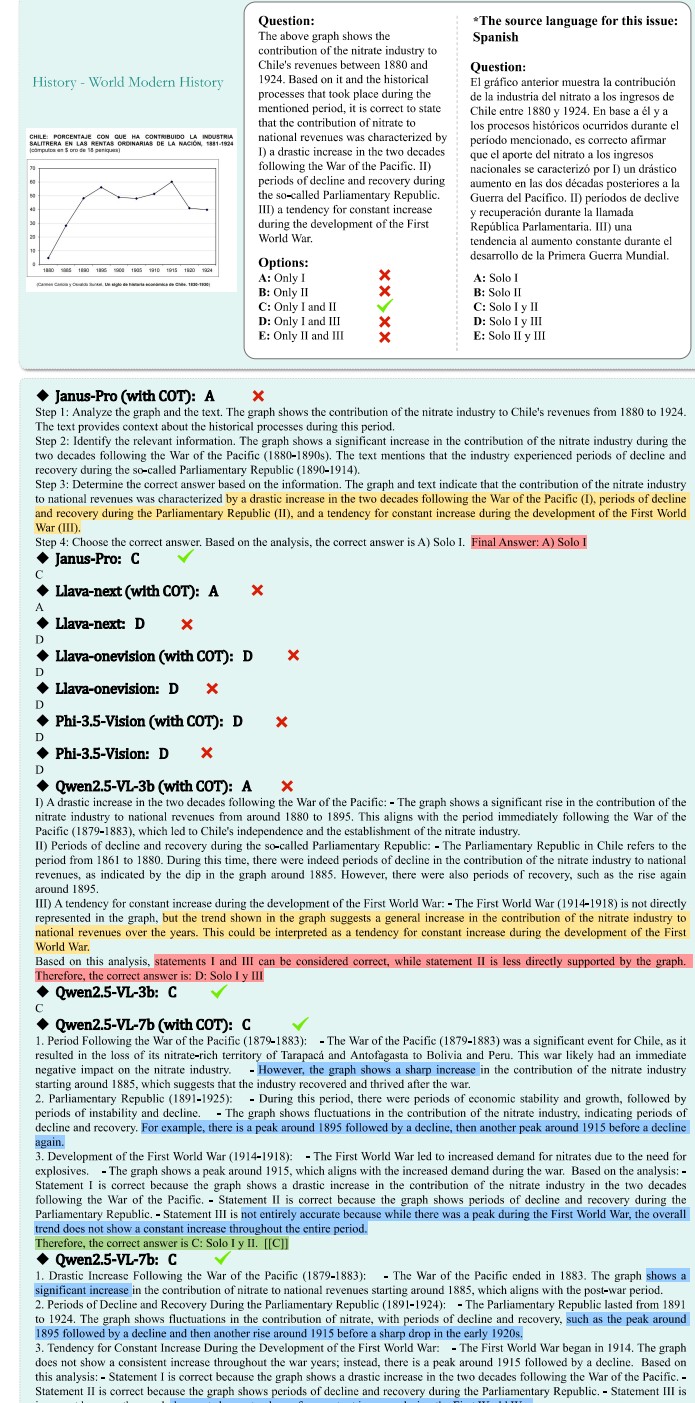

