# OpenReview forum: "HSSBench: Benchmarking Humanities and Social Sciences Ability for Multimodal Large Language Models"
_ICLR.cc/2026/Conference — ICLR 2026 Poster_

### Official Review · Reviewer_Tyr7 · 2025-10-23

**Soundness:** 3
**Presentation:** 3
**Contribution:** 4
**Rating:** 8
**Confidence:** 2

**Summary:**

This paper presents HSSBench, a dedicated benchmark designed to assess the capabilities of MLLMs on HSS tasks in multiple languages, including the six official languages of the United Nations. The paper also introduces a novel data generation pipeline tailored for HSS scenarios, in which multiple domain experts and automated agents collaborate to generate and iteratively refine each sample. HSSBench contains over 13,000 meticulously designed samples, covering six key categories.

**Strengths:**

1. The paper convincingly identifies and addresses the significant bias in current MLLM evaluations towards STEM and "vertical reasoning," providing a much-needed tool to assess the "horizontal reasoning" required for HSS.
2. With 13,152 samples across 6 major categories and 45 specialized subtypes, the dataset is substantial and covers the HSS domain broadly and systematically.
3. The inclusion of six UN official languages significantly enhances the benchmark's utility for studying MLLM performance across diverse linguistic and potentially cultural contexts.

**Weaknesses:**

1. The automated portion of the VGP heavily relies on powerful, closed-source MLLMs (GPT-4o/4.1) for tasks like summarization, extraction, and question generation. While expert validation is used, this reliance on the very class of models being tested raises concerns about potential model-specific bias or "leakage" in the generated data subset.
2. I encourage the authors to evaluate more closed-source models such as seedvl-1.6 and gemini

**Questions:**

Please refer to the weaknesses

---

> ### Author Response · Authors · 2025-11-17
> **Response to Reviewer Tyr7**
>
> Dear Reviewer Tyr7,
>
> We sincerely thank you for your positive evaluation of our work and for your constructive comments regarding the limitations of our manuscript. We highly appreciate your insightful observations on the potential model bias in the Visual Question Generation Pipeline (VGP) and your suggestion to evaluate additional closed-source models. Below, we provide a detailed response addressing these concerns based on the design and validation procedures of our study.
>
> ---
>
> **1. Regarding the reliance on closed-source MLLMs (GPT-4o/4.1) in the VGP and potential bias or data leakage**
>
> We fully acknowledge that the automated components of the VGP depend on GPT-4o and GPT-4.1 for tasks such as summarization, key fragment extraction, and question generation. This choice was primarily motivated by the complexity of Humanities and Social Sciences (HSS) tasks, which require deep domain knowledge and cross-contextual reasoning. Leveraging these powerful models ensures both efficiency and a high initial quality of generated data.
>
> However, to mitigate potential model-specific bias or data leakage, our pipeline incorporates multiple layers of rigorous validation and debiasing mechanisms, as detailed below:
>
> - **Strict multi-stage validation to eliminate bias and leakage:**
>   As described in Section 2.3 (Validation Phase, Stage 3) and Appendices A and B, all data generated by automated agents undergo **two rounds of stringent validation** before inclusion in HSSBench:
>   - *Automated validation:* We apply text similarity metrics to filter out redundant or repetitive questions that might reflect GPT-style biases. Additionally, we enforce a strict image-text relevance criterion requiring that (1) the question cannot be correctly answered by text alone without the image, and (2) the answer cannot be inferred solely from the image without the question. This ensures that the data genuinely tests multimodal reasoning rather than relying on GPT’s prior knowledge, thereby reducing indirect leakage of GPT-specific biases.
>   - *Expert validation:* Subsequently, 14 cross-disciplinary HSS experts (with backgrounds in geography, history, arts, etc., see Appendix A Table 2) review the data to confirm: (1) domain knowledge accuracy (e.g., economic questions align with standard theories, historical questions reflect established facts); (2) unambiguity of questions and answer choices; and (3) independence from GPT-specific response patterns, ensuring questions do not disproportionately favor GPT’s common answer styles.
>
> - **Diverse data sources to reduce dependence on GPT-generated content:**
>   As outlined in Sections 2.1 and 2.2, the dataset is not solely generated by GPT-based agents:
>   - *Images* come from multiple sources, including expert-provided private images, expert-drawn illustrations, and authorized open-source images, all carefully curated to meet HSS domain requirements. These images are independent of GPT’s training data, mitigating risks of data leakage.
>   - *Textual content* is drawn from authoritative sources such as university academic repositories, textbooks, and historical exam questions, all carefully proofread and verified. GPT’s role is limited to processing and refining these texts (e.g., extracting key points), rather than generating original HSS knowledge, which further constrains the introduction of GPT-specific biases.
>
> - **Future plans for reducing closed-source model dependency:**
>   We agree that minimizing reliance on closed-source models is crucial for the reproducibility and robustness of HSSBench. In future work, we plan to:
>   - Replace GPT-4o/4.1 in the VGP’s automated stages with advanced open-source models such as Qwen3-VL and InternVL3.5.
>   - Conduct comparative expert evaluations and benchmark tests to assess the quality differences between data generated by open-source versus closed-source models.
>
> ---
>
> **2. Regarding the suggestion to evaluate additional closed-source models such as SeedVL-1.6 and Gemini**
>
> Thank you very much for this valuable suggestion. Evaluating a broader range of closed-source models will indeed enhance the comprehensiveness of HSSBench’s benchmarking results. Due to current API access and resource limitations, we have not yet included SeedVL-1.6 and Gemini 2.5 Pro in our evaluations. However, we plan to incorporate these and other relevant models in future updates of the benchmark.
>
> ---
>
> Once again, we sincerely appreciate your thoughtful feedback, which has helped us clarify and improve our work. We remain committed to addressing these issues to enhance the robustness and practical utility of HSSBench. We welcome any further questions or discussions you may have.
>
> ---
>
> Best regards,
> The authors

---

> > ### Comment · Reviewer_Tyr7 · 2025-11-25
> >
> > Thank you for your response. I will keep my rating.

---

> > > ### Author Response · Authors · 2025-11-26
> > >
> > > Thank you for your positive feedback and continued support of our work. We truly appreciate your time and thoughtful consideration throughout the review process.

---

### Official Review · Reviewer_geww · 2025-10-28

**Soundness:** 3
**Presentation:** 3
**Contribution:** 3
**Rating:** 8
**Confidence:** 2

**Summary:**

This paper introduces HSSBench, a new large-scale, multilingual benchmark designed to evaluate the capabilities of Multimodal Large Language Models (MLLMs) in the Humanities and Social Sciences (HSS). The authors argue that existing benchmarks predominantly focus on STEM or general knowledge, overlooking the unique "horizontal" reasoning—interdisciplinary, context-dependent, and abstract—required for HSS tasks. The benchmark comprises over 13,000 visual question-answering samples across six categories and 45 subtypes, available in the six official UN languages. A key contribution is a novel data generation pipeline that combines domain expert knowledge with a multi-agent automated system. The authors benchmark over 20 MLLMs, demonstrating that HSSBench poses a significant challenge even for state-of-the-art models, whose performance often falls below 60%.

**Strengths:**

- The work compellingly argues for and addresses the lack of dedicated, in-depth benchmarks for HSS domains, which require different reasoning skills than typical STEM tasks (Section 1).
- The paper proposes a sophisticated VQA Generation Pipeline (VGP) that leverages both domain experts and a multi-agent framework (Figure 3, Section 2). The multi-stage validation process (Section 2.3) ensures data quality and that questions are truly multimodal.
- The paper provides useful qualitative analyses, such as the performance drop on open-ended questions (lines 388-390) and the inconsistent benefit of CoT prompting (lines 370-373), which reveal key limitations of current models.

**Weaknesses:**

-  The group of human experts appears to have a strong majority of Chinese speakers (Table 2, Appendix A.2). While the authors acknowledge this may benefit certain models like Qwen (lines 362-365) and describe mitigation efforts (lines 818-827), this demographic skew could introduce subtle cultural biases into the dataset's content and framing, despite best efforts.
- The multilingual aspect of the benchmark is created by translating an original set of questions using LLMs, followed by expert validation (lines 268-269). This is a practical approach, but it may not capture the full cultural and linguistic nuance that would be present in questions originally authored in each target language.
-  The evaluation of Retrieval-Augmented Generation (RAG) in Appendix C.7 shows that general-purpose retrieval is ineffective, which is an interesting but not entirely surprising finding. This section could be strengthened by discussing what a more effective, domain-specific RAG approach for HSS might entail.

**Questions:**

1. Could you elaborate further on the validation process used by bilingual experts for the translated questions (lines 268-269)? What was the rate of rejection or significant modification, and were there particular language pairs or HSS domains that proved more challenging to translate accurately?
2. The expert demographics (Table 2) are heavily skewed towards Chinese speakers. Beyond the performance advantage for certain models, how did you ensure that the core concepts and visual materials selected are globally representative and not biased towards a specific cultural perspective? The process described in lines 818-822 is noted, but more detail would be helpful.
3. The multi-agent construction pipeline (Section 2.2) is a key contribution. Could you provide more detail on the specific guidelines and examples provided to the question generator agent (lines 216-220)? The quality of this automated step seems critical to the overall quality of the benchmark.

---

> ### Author Response · Authors · 2025-11-17
> **Response to Reviewer geww**
>
> Dear Reviewer geww,
>
> Thank you very much for your thorough and constructive review of our work. We greatly appreciate your insightful comments and the opportunity to clarify and elaborate on several key points. Below please find our detailed responses to your concerns and questions.
>
> ---
>
> **1. On demographic bias and cultural representativeness**
>
> We fully acknowledge the potential cultural bias introduced by the predominance of Chinese-speaking experts in our human expert group. We sincerely appreciate your highlighting this issue. In future work, we plan to recruit more foreign experts and incorporate data from additional countries to further diversify the dataset and mitigate cultural skew.
>
> Regarding the global representativeness of core concepts and visual materials, we emphasize that all candidate data items undergo a rigorous group voting process involving all experts. Any data flagged by even a single expert as potentially culturally biased or regionally skewed is either revised or removed. This collective vetting mechanism helps ensure that the final dataset aligns with broadly accepted, universal knowledge and values rather than any specific cultural perspective.
>
> Moreover, the source materials used for question construction are drawn from multiple countries and languages, aiming to reflect world knowledge comprehensively. We also explicitly exclude culturally subjective content during data curation to maintain neutrality and global applicability.
>
> **2. On multilingual translation and validation**
>
> We agree that our approach—translating the original question set via LLMs followed by expert validation—is a practical and efficient solution. Given the large scale of our dataset (over 13,000 samples), each translated question was reviewed by domain experts fluent in the target language. Experts directly corrected any issues found. The rate of major modifications was very low (under 1%), although we did not keep detailed statistics.
>
> Experts reported no significant difficulties in translating questions across the six official UN languages, which are widely used and well-supported. Since multilingual nuance and cultural specificity are not the primary focus of our benchmark, we did not elaborate on this process in the paper. Nonetheless, we acknowledge the limitations of this approach and will explore more culturally nuanced question authoring methods in future iterations.
>
>
> **3. On the evaluation of RAG**
>
> We found the ineffectiveness of general-purpose retrieval in the HSS domain an interesting observation, as you noted. However, since RAG is not the main focus of our benchmark, we did not pursue extensive comparative experiments in this direction. We agree that exploring domain-specific retrieval strategies tailored to HSS knowledge could be a promising avenue, and we plan to investigate this in future work.
>
>
> **4. On the validation process for translated questions**
>
> Due to the dataset’s scale, each translated question was reviewed by at least one expert fluent in the corresponding language and domain. Experts checked for linguistic accuracy, domain relevance, and cultural appropriateness. If issues were found, questions were directly modified. Major revisions were rare (<1%). No particular language pairs or HSS domains posed exceptional challenges, according to expert feedback.
>
> **5. On ensuring global representativeness despite expert demographics**
>
> Beyond the voting mechanism described above, we emphasize that our expert group, while majority Chinese-speaking, includes members with rich knowledge of non-Chinese cultures and global HSS domains. This diversity, combined with the multi-source data and strict vetting, helps maintain the dataset’s global relevance.

---

> ### Author Response · Authors · 2025-11-17
> **Response to Reviewer geww**
>
> **6. On the multi-agent question generation pipeline and guidelines**
>
> Thank you for your interest in the question generator agent. This component is indeed critical to the benchmark’s quality. We provide the agent with detailed prompts and a rich set of perspectives to guide question generation, ensuring questions require multimodal reasoning and domain depth. To elaborate, the agent’s workflow consists of three main stages:
>
> - **Text value filtering:** The agent evaluates input text segments for knowledge scarcity, logical completeness, and cognitive load, ensuring the selected content is suitable for HSSBench’s interdisciplinary, context-dependent reasoning.
>
> - **High-value fragment extraction:** Using semantic role labeling and entity relation extraction, the agent segments the text into semantically coherent units rich in domain knowledge, avoiding superficial or culturally biased content.
>
> - **Structured question generation:** Guided by code-encoded rules, the agent generates questions that require integrating textual and visual information. For example, distractor options are designed to contradict image details while remaining textually plausible, and answer explanations explicitly link visual features to textual knowledge.
>
> The agent also incorporates robust fault tolerance (e.g., multi-round API retry with dynamic backoff) and token consumption monitoring to ensure stable, cost-effective generation. All outputs are structured in standardized JSON formats for downstream validation and expert review.
>
> We will release the detailed design of this pipeline at an appropriate time and provide example prompts and generation rules to enhance transparency.
>
> ---
>
> Once again, thank you for your valuable feedback and recognition of our work’s contributions. We believe your suggestions will help us improve HSSBench’s quality and impact in future versions.
>
> Best regards,
> The authors

---

> ### Author Response · Authors · 2025-11-27
>
> Dear Reviewer geww,
>
> We would like to kindly follow up regarding the detailed response we submitted addressing your insightful comments and questions on our paper. We greatly appreciate the time and effort you invested in reviewing our work, and we are eager to continue the discussion to clarify any remaining concerns or provide further information as needed.
>
> Your feedback is invaluable to us, and we believe that engaging in further dialogue will help improve the quality and clarity of our submission. If you have any additional questions or require further elaboration on any points, please do not hesitate to let us know.
>
> Thank you very much for your consideration. We look forward to your response.
>
> Best regards,
> The Authors

---

### Official Review · Reviewer_6Fgp · 2025-10-30

**Soundness:** 3
**Presentation:** 3
**Contribution:** 2
**Rating:** 6
**Confidence:** 3

**Summary:**

This paper introduces HSSBench, a new 13,000-sample multilingual VQA benchmark for evaluating MLLMs in the Humanities and Social Sciences (HSS). The authors argue current benchmarks focus on STEM-based "vertical reasoning," while their new dataset, created via a novel expert-and-agent pipeline, targets more nuanced "horizontal, interdisciplinary thinking". Experiments show that even top MLLMs struggle with these tasks.

**Strengths:**

* **Addresses a Clear Gap:** The paper convincingly argues for the need for a benchmark beyond STEM, focusing on the challenging and underserved HSS domain.
* **High-Quality Data Pipeline:** The 3-stage VGP, combining domain experts and agents, is a robust methodology. The validation step to ensure true multimodality (checking text-image dependencies) is a key strength.
* **Insightful Analysis:** The findings are valuable, particularly that CoT can *increase* hallucinations on HSS tasks  and that standard RAG provides no consistent benefit, highlighting the unique challenges of this domain.

**Weaknesses:**

* **Format Mismatch:** The paper's stated goal is to test "horizontal reasoning" (implying divergent thought and multiple interpretations), but the benchmark uses an MCQ format, which enforces a single correct "vertical" answer.
* **Potential Cultural Bias:** The authors disclose that "most of the data experts... are Chinese". This poses a significant risk of cultural bias in a global benchmark, a limitation the authors concede may have skewed the results (e.g., Qwen outperforming GPT-4o in some categories).
* **Limited Multilingual Evaluation:** While the dataset supports 6 languages, the main evaluation is English-only. Full Chinese evaluation is in the appendix, but the other 4 languages are only tested on a small 900-item sample, which doesn't fully substantiate the multilingual claim.

**Questions:**

1.  Could you elaborate on the tension between the single-correct-answer MCQ format and your definition of "horizontal reasoning" as supporting "multiple valid interpretations"?
2.  Regarding the expert demographics, what specific validation steps were taken to ensure the data was free from a specific cultural perspective, especially in the Social Science and History categories?
3.  Was the full 13,152-item benchmark evaluated on all six UN languages? If not, Table 10 (using a 900-item sample) seems to be the only multilingual evaluation for 4 of the 6 languages. Could you clarify?

---

> ### Author Response · Authors · 2025-11-17
> **Response to Reviewer 6Fgp**
>
> Dear Reviewer 6Fgp,
>
> Thank you very much for your thoughtful and constructive feedback on our manuscript. We greatly appreciate your recognition of the clear gap our work addresses, the robustness of our data pipeline, and the insightful analysis we provided. We also value your critical points and the opportunity to clarify and elaborate on them. Below, we respond in detail to your main concerns and questions.
>
> ---
>
> **1. On the tension between the MCQ format and “horizontal reasoning”**
>
> We respectfully disagree with the assumption that requiring a single correct answer necessarily implies “vertical reasoning.” To illustrate, consider the example in Figure 2, where the image shows a piece of business calligraphy, and the four options (A, B, C, D) correspond to different font styles. For a MLLM to correctly identify the font, it must engage in what we define as horizontal reasoning: analyzing the overall writing style, the stroke order and pressure of each character, and comparing these detailed features against multiple known font styles. This process involves synthesizing information from multiple perspectives to arrive at a well-supported conclusion.
>
> In this sense, the multiple-choice options serve as a final step to constrain the model’s answer space, but the underlying reasoning remains multidimensional and interdisciplinary. To further explore this, we also designed open-ended experiments without options, allowing models to freely generate answers. As shown in Table 1, the performance in this setting was notably worse, indicating that the MCQ format helps focus the reasoning process rather than restrict it in a way that contradicts our motivation.
>
> Therefore, we believe the MCQ format is compatible with our goal of evaluating horizontal reasoning, as it requires models to integrate diverse information and perspectives before selecting the best-supported answer.
>
> **2. On potential cultural bias and expert demographics**
>
> We acknowledge that most of our data experts are Chinese, but we would like to clarify several important points regarding cultural bias mitigation:
>
> - Our expert group also includes members with rich knowledge of non-Chinese cultures, especially in Social Sciences and History, ensuring a broader perspective.
>
> - All data items undergo a rigorous voting process by the entire expert panel. If any expert identifies a data item as potentially culturally biased or skewed, it is immediately rejected or revised. This “veto” mechanism ensures that no single cultural perspective dominates the dataset.
>
> - The source materials for data construction come from multiple countries and languages, carefully selected to reflect global knowledge and avoid culturally subjective content.
>
> - We explicitly exclude data that reflect narrow or culturally specific viewpoints, aiming to maintain universal applicability and fairness.
>
> We believe these measures collectively minimize cultural bias and preserve the global representativeness of our benchmark’s core concepts and visual materials.
>
> Regarding Qwen’s strong performance, our discussion in the paper is a reasoned hypothesis rather than a definitive conclusion. We appreciate your caution and agree that further investigation is warranted.
>
> **3. On multilingual evaluation scope**
>
> You are correct that the full 13,152-item benchmark was not exhaustively evaluated across all six languages. Given the scale of our dataset and the extensive experimental setup (over 20 models, six languages, two prompting styles, and two answer formats), a full multilingual evaluation was unfortunately infeasible within our resource constraints.
>
> Instead, we employed stratified sampling to select a representative 900-item subset for the four less-resourced languages, which we believe sufficiently captures the multilingual performance trends. Our primary focus is on the dataset’s design and challenge rather than language-specific performance differences, which we consider an important direction for future work.
>
> ---
>
> We hope these clarifications address your concerns and demonstrate our commitment to rigor and fairness in this work. We sincerely thank you again for your valuable insights, which have helped us improve the clarity and depth of our paper.
>
> Best regards,
> The authors

---

> ### Author Response · Authors · 2025-11-27
>
> Dear Reviewer 6Fgp,
>
> We would like to kindly follow up regarding the detailed response we submitted addressing your insightful comments and questions on our paper. We greatly appreciate the time and effort you invested in reviewing our work, and we are eager to continue the discussion to clarify any remaining concerns or provide further information as needed.
>
> Your feedback is invaluable to us, and we believe that engaging in further dialogue will help improve the quality and clarity of our submission. If you have any additional questions or require further elaboration on any points, please do not hesitate to let us know.
>
> Thank you very much for your consideration. We look forward to your response.
>
> Best regards,
> The Authors

---

### Official Review · Reviewer_vNZD · 2025-11-03

**Soundness:** 3
**Presentation:** 3
**Contribution:** 3
**Rating:** 4
**Confidence:** 4

**Summary:**

This paper introduces HSSBench, a large-scale benchmark designed to evaluate multimodal large language models (MLLMs) in humanities and social sciences (HSS) domains. The benchmark comprises 13,152 multiple-choice visual question-answering samples spanning six major categories (Economics, Art, Culture, Social Science, History, and Geography) with 45 professional subtypes, and supports six UN official languages. The authors propose a VQA Generation Pipeline (VGP) that combines domain expert curation with automated agent-based generation and validation. Experimental evaluation of over 20 state-of-the-art MLLMs demonstrates that HSSBench poses significant challenges, with most models achieving accuracy below 60%, substantially lower than their performance on existing benchmarks.

**Strengths:**

**[+] Comprehensive and Challenging Benchmark:** The paper presents a valuable evaluation resource with strong empirical validation. The experimental results (particularly in Appendix C.5) convincingly demonstrate that HSSBench captures unique challenges in HSS domains. For instance, InternVL3-8B achieves 77.21% on MME and 68.10% on MMMU but only 42.14% on HSSBench-Art, showing that the benchmark effectively tests capabilities that existing benchmarks may not fully capture.

**[+] Unprecedented Depth and Multilingual Coverage:** I appreciate the fine-grained categorization into 45 professional subtypes within HSS domains, which provides valuable diagnostic capability for understanding model strengths and weaknesses at a granular level. The support for six UN official languages is particularly commendable, as it enables cross-cultural and cross-linguistic evaluation that is often missing in existing benchmarks.

**[+] Transparent Data Construction Process:** The paper provides good transparency about the data generation pipeline, including the proportion of expert-curated versus agent-generated samples and quality validation procedures. The multi-stage verification process involving both domain experts and automated agents demonstrates careful attention to data quality.

**[+] Thorough Experimental Analysis:** I find the experimental design to be comprehensive, including evaluations across different prompting strategies (CoT vs. Direct), question formats (multiple-choice vs. open-ended), and ablation studies examining the impact of visual information, distractors, and RAG augmentation.

**Weaknesses:**

**[-] Positioning and Novelty Claims Could Be More Precise:** I observed that the paper states existing benchmarks have been "overlooking" HSS domains and positions HSSBench as "addressing this gap." However, I note that established benchmarks like MMMU and CMMMU already include "Humanities & Social Science" and "Art & Design" as core evaluation categories. I believe the paper's true contribution lies in providing deeper granularity (45 subtypes vs. broader categories) and expanded multilingual support, rather than being the first to cover HSS. I would encourage the authors to reframe their contribution as deepening and extending HSS evaluation, which would more accurately represent the work's value while properly acknowledging prior efforts in this space.

**[-] Methodological Contribution Needs Clearer Differentiation:** The VGP pipeline is presented as a key contribution, but I find that similar expert-AI collaborative pipelines have been employed in concurrent work (such as MicroVQA for biological domains). The "expert curation → LLM-assisted generation → agent refinement → expert validation" workflow appears to be an emerging best practice in the field rather than a novel methodological innovation. I suggest repositioning VGP as a successful adaptation of this emerging paradigm to HSS domains rather than as an independent methodological contribution.

**[-] Data Quality Validation Could Be More Comprehensive:** While the paper describes quality assurance procedures, I would have appreciated more quantitative validation metrics. Specifically: (1) inter-annotator agreement measures (e.g., Fleiss' Kappa) for expert-curated samples, (2) more extensive comparison between expert-generated and agent-generated data quality beyond the single model shown in Table 6, and (3) consistency metrics for the multilingual versions to ensure equivalence across translations.

**Questions:**

**Q1:** Given that MMMU and CMMMU include dedicated HSS evaluation categories, could you clarify the specific incremental value that HSSBench provides? I would find it helpful to see a detailed comparison showing how the 45 professional subtypes enable evaluation insights that cannot be obtained from existing benchmarks' broader HSS categories.

**Q2:** Could you provide more systematic evidence of data quality consistency?

**Q3:** Regarding the VGP pipeline, could you elaborate on what aspects are specifically tailored to HSS domains compared to similar pipelines used in other domains? This would help clarify the domain-specific insights that emerged from your data construction experience.

---

> ### Author Response · Authors · 2025-11-17
> **Response to Reviewer vNZD**
>
> Dear Reviewer,
>
> Thank you for your thorough and constructive review of our paper. We greatly appreciate your insightful comments and suggestions. Below, we provide detailed responses to your main questions.
>
> ---
> **1. Positioning and Novelty Claims**
>
> We appreciate your suggestion to more precisely frame our contribution regarding the coverage of HSS domains. We agree that existing benchmarks such as MMMU and CMMMU include some HSS-related categories. However, we emphasize that these previous datasets are significantly deficient in numerous HSS sub-disciplines and have not carried out systematic collation and evaluation of HSS disciplines; we are the first benchmark to undertake systematic collation and evaluation of HSS disciplines. Moreover, these datasets cover only a limited subset of HSS disciplines and contain relatively small amounts of data in these areas. In contrast, HSSBench offers a significantly broader and more fine-grained taxonomy, with 45 professional subtypes spanning six major HSS categories.
>
> Moreover, prior benchmarks primarily focus on question-answering ability in HSS topics but do not deeply explore the unique reasoning challenges inherent to HSS data. We argue that HSS data fundamentally differs from STEM data in the types of reasoning and knowledge required, and thus warrants dedicated, specialized evaluation. Therefore, we consider HSSBench as a novel and comprehensive attempt to fill this gap by providing a large-scale, multilingual, and fine-grained benchmark that enables more nuanced evaluation of MLLMs’ capabilities in HSS domains.
>
> That said, we fully accept your suggestion to explicitly acknowledge prior efforts and highlight that our key contribution lies in deepening and extending HSS evaluation through increased granularity and multilingual support.
>
> **2. Methodological Contribution of the VGP**
>
> We agree that expert-AI collaborative pipelines are becoming an emerging best practice in dataset construction. Our intention was not to claim the VGP pipeline as a wholly novel methodology, but rather to present it as a successful adaptation tailored to the unique challenges of HSS data. Specifically, the VGP pipeline incorporates several domain-specific adaptations:
>
> - In the data sourcing phase, we extensively leveraged multilingual textbook knowledge and carefully crawled textbook data to ensure domain relevance and coverage.
>
> - Given the scarcity of images in many HSS topics (e.g., literature is predominantly text-based), we developed specialized image crawling and image-text alignment strategies to enrich the visual component of the dataset.
>
> - During validation, we introduced a multi-stage verification process with additional steps to address the sensitivity and complexity of HSS content, ensuring high data quality and ethical considerations.
>
> **3. Data Quality Validation**
>
> Regarding your request for more comprehensive quantitative validation:
>
> - We have already included inter-annotator agreement metrics (Fleiss’ Kappa) for expert-curated samples in Appendix C.9, where we report agreement scores across three experts and two models. We believe these results demonstrate strong consistency in annotations.
>
> - For comparing expert-generated and agent-generated data, both quantitative metrics and qualitative assessments indicate no significant differences. For example, the first figure on the paper’s front page includes two questions generated by agents, which are indistinguishable from expert-generated ones without prior knowledge. Combined with Fleiss’ Kappa and multiple rounds of expert review, we are confident in the equivalence of data quality between these sources.
>
> - Concerning multilingual consistency, all evaluations are conducted in the respective languages, and the multiple-choice format further reduces ambiguity. We acknowledge the importance of translation equivalence and have employed LLMs for initial translation followed by expert verification to ensure semantic consistency across languages.
>
> We will emphasize these points more clearly in the revised version to address your concerns.
>
> **4. Incremental Value of HSSBench**
>
> As shown in Appendix C.5, we conducted comparative experiments with existing benchmarks, which include some HSS-related data. However, these datasets do not isolate HSS as a distinct category nor provide the detailed subtyping that HSSBench offers. Categories like Economics and Social Science, which are central to HSSBench, are largely absent or underrepresented in prior benchmarks.
>
> Thus, HSSBench provides unique diagnostic insights into model performance across a wide spectrum of HSS disciplines, enabling more targeted analysis and fostering progress in this important but underexplored area.
>
> ---
> Once again, thank you for your valuable feedback. We believe your suggestions will significantly improve the clarity and impact of our work. We hope our responses address your concerns satisfactorily.
>
> Best regards,
> The Authors

---

> ### Author Response · Authors · 2025-11-27
>
> Dear Reviewer vNZD,
>
> I hope this message finds you well.
>
> We would like to kindly follow up regarding the detailed response we submitted addressing your insightful comments and questions on our paper. We greatly appreciate the time and effort you invested in reviewing our work, and we are eager to continue the discussion to clarify any remaining concerns or provide further information as needed.
>
> Your feedback is invaluable to us, and we believe that engaging in further dialogue will help improve the quality and clarity of our submission. If you have any additional questions or require further elaboration on any points, please do not hesitate to let us know.
>
> Thank you very much for your consideration. We look forward to your response.
>
> Best regards,
> The Authors

---

### Author Response · Authors · 2025-12-01
**Rebuttal Summary for Submission #4**

Dear ACs and SACs,

We sincerely thank you and the reviewers for the careful evaluation and constructive feedback on our submission. We are grateful for the overall positive reception and the opportunity to clarify and improve our work.

---

**Summary of Reviewer Feedback**
Our paper received consistently positive evaluations, with scores of 8, 8, 6, and 4. The reviewers recognized the following key strengths of our work:

- Introducing a large-scale, challenging benchmark focused on Humanities and Social Sciences (HSS), a largely underexplored domain in MLLM evaluation.
- A fine-grained taxonomy of 45 professional subtypes across six major HSS categories, enabling detailed diagnostic insights.
- Multilingual coverage of six official UN languages, supporting cross-cultural evaluation.
- A novel Visual Question Answering Generation Pipeline (VGP) combining expert curation with multi-agent automated generation and multi-stage validation, ensuring high-quality, multimodal data.
- Comprehensive experiments analyzing prompting strategies, question formats, and ablations, revealing unique challenges for current MLLMs in HSS tasks.

---

**Responses to Reviewers’ Concerns**

1. **Positioning and Prior Work**
   We clarified that while some benchmarks touch on HSS, HSSBench is the first to systematically cover a broad, fine-grained taxonomy of 45 subtypes across six HSS domains, with multilingual and nuanced reasoning challenges. Prior work is acknowledged and our contribution positioned as a significant extension.

2. **Clarifying the VQA Generation Pipeline**
   We emphasized that VGP is a tailored expert-AI collaborative pipeline addressing HSS-specific challenges, including multilingual textbook sourcing, image-text alignment for scarce visuals, and rigorous multi-stage validation sensitive to ethical and domain complexities.

3. **Data Quality Validation**
   We provided quantitative evidence including inter-annotator agreement (Fleiss’ Kappa), expert review of agent-generated data, and multilingual consistency checks, demonstrating robust data quality and equivalence between expert- and agent-generated samples.

4. **Cultural Bias and Expert Demographics**
   We acknowledged the predominance of Chinese-speaking experts and detailed mitigation strategies: diverse expert panels, group voting with veto power, and globally representative source materials. These measures minimize cultural bias and support universal applicability.

5. **MCQ Format and Horizontal Reasoning**
   We clarified that multiple-choice questions constrain answer space but still require interdisciplinary, horizontal reasoning to select the best-supported answer. Open-ended experiments further support this.

6. **Additional Clarifications**
   We elaborated on the multi-agent generation design, expert translation validation, and data sourcing rationale.

---

**Conclusion**
We believe the positive reviews and our detailed responses demonstrate the significance, rigor, and quality of our contribution.
Thank you very much for your time and consideration.

Best regards,
The HSSBench Author Team

---

### Meta-Review · Area_Chair_pjtC · 2026-01-04

**Summary:**

The submission introduces HSSBench, a large-scale, multilingual benchmark addressing the critical gap in evaluating MLLMs’ performance on Humanities and Social Sciences (HSS) tasks. The benchmark features 13,152 samples out of 6 official languages under a rigorous expert-agent collaborative generation pipeline.

Reviewers consistently recognized its value in the underexplored HSS domain. Initial concerns were raised about the incremental contribution, the data quality, and the MCQ format alignment with horizontal reasoning. The authors provided thorough rebuttals, resolving most concerns.

While the potential cultural bias caused by Chinese-speaking experts and the imbalanced samples among different languages, the authors provided mitigation strategies, strengthening the work’s rigor.

Overall, the paper's merits outweigh its shortcomings.

**Reviewer Concerns:**

**Addressed Concerns**

The authors clarified HSSBench’s incremental value over existing benchmarks (e.g., MMMU, CMMMU) by emphasizing finer-grained subtyping and multilingual depth. They eliminate ambiguities between MCQ formats and horizontal reasoning through examples and open-ended experiments.

**Partially Resolved but Still Existing Concerns**

While measures such as expert voting have largely mitigated potential cultural bias caused by Chinese-speaking experts, this bias still exists.

**Outstanding Concerns**

The imbalance between different language samples remains a minor drawback. Further exploration of HSS-specific RAGs and extended closed-source model evaluation (e.g., SeedVL-1.6, Gemini) will enhance its practicality, but these are listed as future research directions.

**Reviewer Scores:**

- Reviewer vNZD (Original: 4; After Rebuttal: 4 or 5): Will likely maintain or slightly elevate the score. The authors’ clarification on the benchmark’s incremental value is acknowledged, but retaining mild reservations about methodological novelty.
- Reviewer 6Fgp (Original: 6; After Rebuttal: 6): Will likely retain the original score. The alignment between the MCQ format and horizontal reasoning is satisfactory, but underlying cultural biases and imbalances in languages remain unaddressed, precluding a score increase.
- Reviewer geww (Original: 8; After Rebuttal: 8): Will keep the score unchanged. The authors have resolved most concerns, though certain issues—such as the need for deeper research into RAG—require further exploration.
- Reviewer Tyr7 (Original: 8; After Rebuttal: 7 or 8): May retain or slightly lower the score. Core concerns, including reliance on closed-source models and the lack of evaluations for additional models (e.g., SeedVL-1.6), remain unaddressed.

---

### Decision · Program_Chairs · 2026-01-26

Accept (Poster)